# Condensates of synaptic vesicles and synapsin-1 mediate actin sequestering and polymerization

Akshita Chhabra [ID] [1,2,10], Christian Hoffmann [ID] [1,2,10], Gerard Aguilar Pérez [ID] [1,2,10],
Aleksandr A Korobeinikov [ID] [1], Jakob Rentsch [ID] [3], Nadja Hümpfer [ID] [3], Linda Kokwaro [ID] [1,4],
Luka Gnidovec[1], Arsen Petrović [ID] [5], Jaqulin N Wallace [ID] [6], Johannes Vincent Tromm [ID] [1,2,6],
Cristina Román-Vendrell [ID] [6], Emma C Johnson [ID] [6], Branislava Ranković [ID] [1,2], Eleonora Perego[7],
Tommaso Volpi [ID] [8], Rubén Fernández-Busnadiego [ID] [5], Sarah Köster [ID] [7], Silvio O Rizzoli [ID] [9],
Helge Ewers [ID] [3], Jennifer R Morgan [ID] [6] & Dragomir Milovanović [ID] [1,2,4]✉

## Abstract

**Neuronal communication relies on precisely maintained synaptic vesicle (SV) clusters, which assemble via liquid-liquid phase separation. This process requires synapsins, the major synaptic phosphoproteins, which are known to bind actin. Reorganization of SVs, synapsins, and actin is a hallmark of synaptic activity, but the molecular details of the interactions between these components remain unclear. Here, we combine in vitro reconstitution with expansion microscopy, super-resolution imaging, and cryo-electron tomography to dissect the roles of SV-synapsin-1 condensates in the organization of the presynaptic actin cytoskeleton. Our results indicate that condensation of synapsin-1 initiates actin polymerization. This process enables SV-synapsin-actin assemblies to facilitate the mesoscale organization of SV clusters along axons, which is similar to the native presynaptic organization observed at both lamprey and mammalian synapses. Understanding the relationship between the actin network and synapsin-synaptic vesicle condensates can help elucidate how coordinated neurotransmission along the axon enables circuit function and behavior.**

**Keywords** Actin; Synapsin; Presynapses; Synaptic Vesicles; Liquid-Liquid Phase Separation
**Subject Categories** Cell Adhesion, Polarity & Cytoskeleton; Neuroscience

## Introduction

Liquid-liquid phase separation (LLPS) is a major mechanism for organizing macromolecules, particularly proteins with intrinsically disordered regions, in compartments not limited by a membrane or a scaffold (Banani et al, 2017). However, it is currently unclear how such a complex concoction operates to allow for intracellular trafficking, signaling, and metabolic processes to occur with high spatio-temporal precision. We recently discovered that the cluster of synaptic vesicles (SVs) at nerve terminals represents a biomolecular condensate, which is packed with membrane-bound organelles, the synaptic vesicles (Milovanovic et al, 2018; Sansevrino et al, 2023; Alfken et al, 2024). While decades of research have pointed out that SVs are clustered by synapsins, a family of phosphoproteins highly abundant at nerve terminals (De Camilli et al, 1990; Rosahl et al, 1995; Pieribone et al, 1995), only recently the intrinsically disordered region (IDR) of synapsin was shown to be central for the assembly of SV condensates (Milovanovic et al, 2018; Hoffmann et al, 2023b). These data corroborate the analysis in living synapses, both upon acute and chronic disruption of synapsins. For example, the injection of anti-synapsin antibodies in the giant reticulospinal synapse of the lamprey results in the dispersion of SVs upon depolarization and at rest (Pieribone et al, 1995; Pechstein et al, 2020). Similarly, the chronic depletion of synapsins in mice—i.e., through synapsin gene deletions—results in less vesicles accumulating at the synaptic boutons; the SVs within the bouton are more dispersed than in wild-type synapses (Rosahl et al, 1995; Gitler et al, 2004). As a consequence, when synapsin is disrupted at synapses, there is a faster run-down of neurotransmission due to the lack of SVs (Pieribone et al, 1995; Hilfiker et al, 1998; Gitler et al, 2004; Song and Augustine, 2023).

[1]Laboratory of Molecular Neuroscience, German Center for Neurodegenerative Diseases (DZNE), 10117 Berlin, Germany. [2]Whitman Center, Marine Biological Laboratory, 02543 Woods Hole, MA, USA. [3]Institute of Chemistry and Biochemistry, Freie Universität Berlin, 14195 Berlin, Germany. [4]Institute of Biochemistry and Einstein Center for Neurosciences, Charité-Universitätsmedizin Berlin, Corporate Member of Freie Universität Berlin, Humboldt-Universität Berlin, and Berlin Institute of Health, 10117 Berlin, Germany. [5]Institute for Neuropathology, University Medical Center Göttingen, 37073 Göttingen, Germany. [6]The Eugene Bell Center for Regenerative Biology and Tissue Engineering, Marine Biological Laboratory, 02543 Woods Hole, MA, USA. [7]Institute for X-Ray Physics, University of Göttingen, Friedrich-Hund-Platz 1, 37077 Göttingen, Germany. [8]Department of Radiology and Biomedical Imaging, Yale University School of Medicine, New Haven, CT 06520, USA. [9]Institute for Neuro- and Sensory Physiology, University Medical Center Göttingen, 37073 Göttingen, Germany. [10]These authors contributed equally: Akshita Chhabra, Christian Hoffmann, Gerard Aguilar Pérez. ✉E-mail: dragomir.milovanovic@dzne.de; dragomir.milovanovic@charite.de

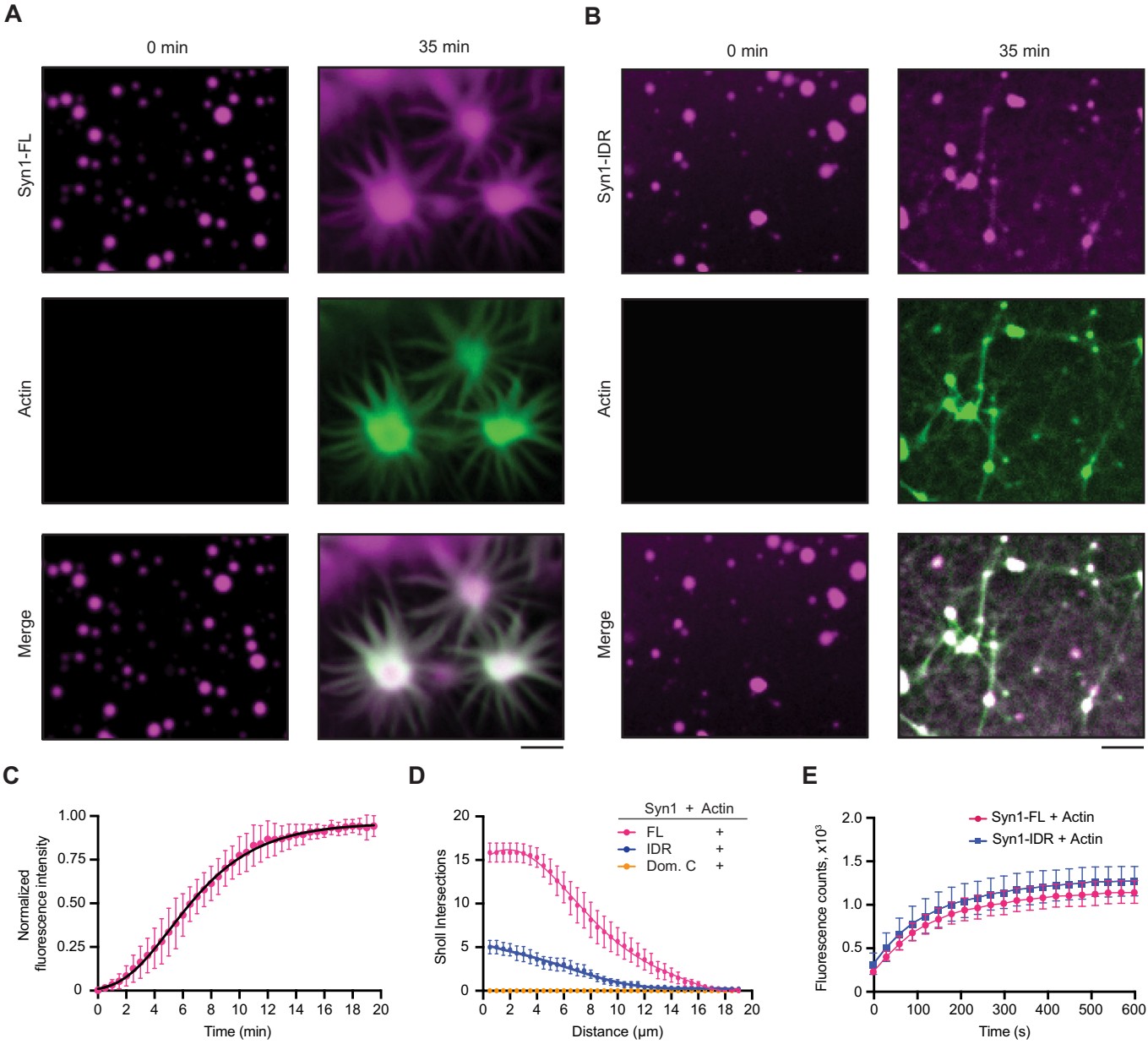

**Figure 1. Synapsin-1 condensates are sites for actin polymerization.**

(A) Representative confocal images from the in vitro reconstitution of EGFP-Syn1-FL (4 µM, 3% (w/v) PEG 8000) with ATTO647-labeled G-actin monomers (4 µM) at $t = 0$ (left) and 35 min (right). Buffer used in these experiments, hereafter referred to as 'reaction buffer', contained: 25 mM Tris-HCl (pH 7.4), 150 mM NaCl, 0.5 mM TCEP. Note the condensates growing over time and forming radial "asters." Scale bar, 5 µm. (B) Confocal images of the EGFP-Syn1-IDR (4 µM, 3% (w/v) PEG 8000) and ATTO647-labeled G-actin reconstitution (4 µM) in reaction buffer at $t = 0$ (left) and 35 min (right). Scale bar, 5 µm. (C) Plot for the quantification of normalized enrichment of actin within EGFP-Syn1-FL phases. Gompertz growth curve: $Y = YM*(Y0/YM)\hat{}(exp(-K*X))$, $YM = 0.9611, Y0 = 0.01032, K = 0.2958, 1/K = 3.381$, $R^2 = 0.9376$. Data shown here is quantified from three independent experiments and 30 analyzed condensates ($N = 3, n = 30$). Each dot represents the mean and error bars, SD. (D) Plot showing the number of Sholl intersections for polymerized actin in the presence of EGFP-Syn1-FL, EGFP-Syn1-IDR, and EGFP-Syn1-Dom. C condensates as a function of distance (in µm). The number of Sholl intersections was counted in 0.5 µm radius increments from the center of the condensates. Data from three independent replicates (For Syn1-FL: $N = 3; n = 21$; for Syn1-IDR: $N = 3; n = 10$; for Syn1-Dom.C $N = 3; n$ no condensates formed). Each dot represents the mean intersection count for each condition. Error bars represent SEM. (E) Plot depicting the quantification of pyrene-actin fluorescence intensity as a function of time. Pyrene-actin (10 µM) polymerization was assessed in the presence of EGFP-Syn1-FL or EGFP-Syn1-IDR (10 µM) in 1X actin polymerization buffer (10 mM Tris-HCl, pH = 7.5, 2 mM MgCl$_2$, 50 mM KCl, and 0.5 mM ATP). The data shown here is quantified from three independent experiments ($N = 3$). Each dot represents the mean and error bars, SD. Source data are available online for this figure.

 

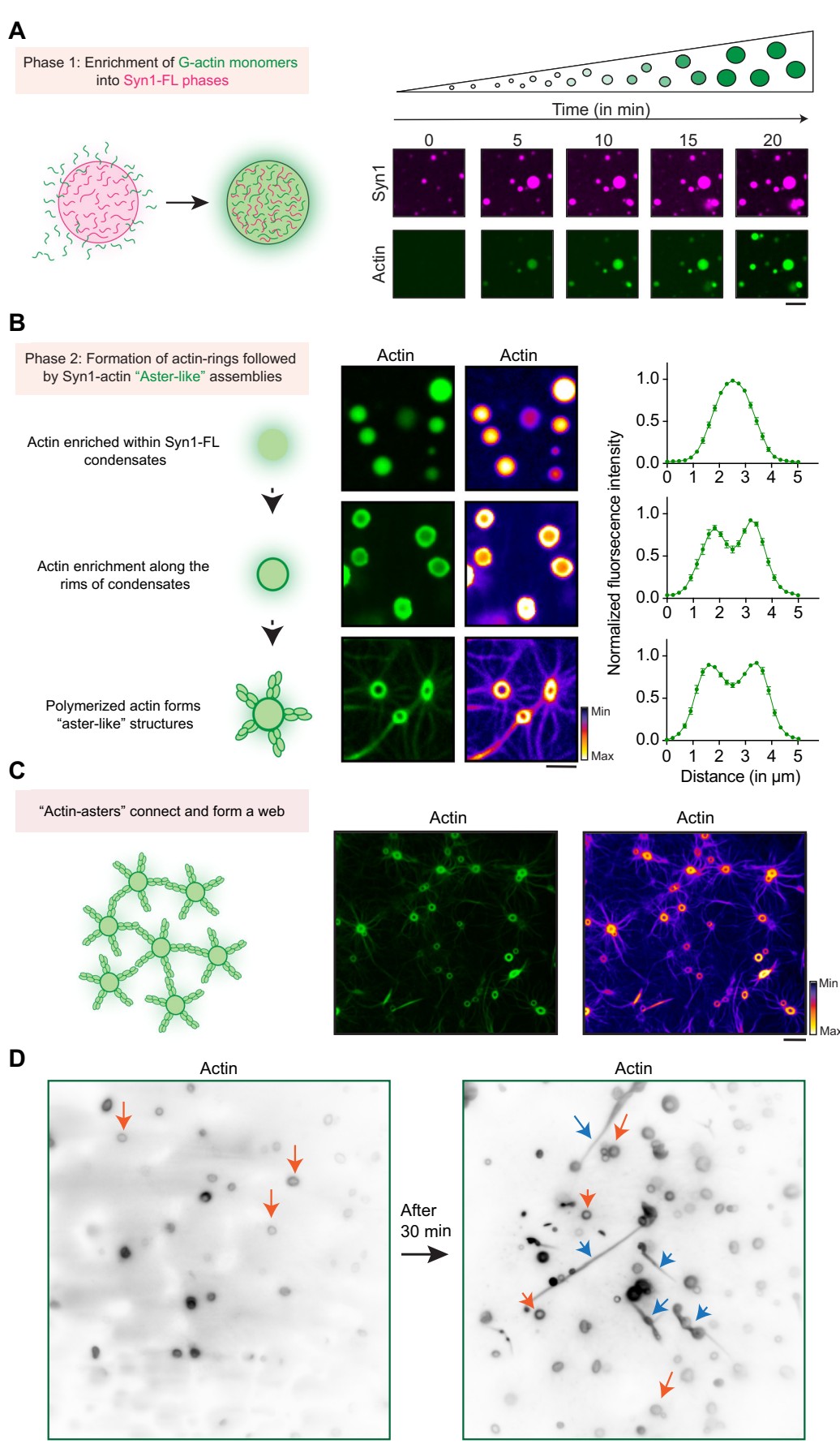

**Figure 2. Actin torus formation around Syn1-FL condensates precedes aster assembly.**

(A) Scheme and corresponding representative confocal images from the in vitro Syn1-actin reconstitution assay, showing actin enrichment over a period of 20 min within EGFP-Syn1-FL condensates. Images were acquired at excitation wavelengths 488 and 647 nm for EGFP-synapsin-1-FL and ATTO647 G-actin, respectively. Scale bar, 5 μm. (B) Scheme and corresponding representative confocal images depicting stages of "actin-aster" assembly from EGFP-Syn1-FL condensates. Scale bar, 5 μm. Here, the same datasets are shown with two different LUTs to highlight "torus" and "aster" structures. Right: line profiles depict the enrichment of fluorescently labeled actin within the condensate. Data obtained from three independent reconstitutions, 30 condensates analyzed for each stage. Error bars represent SEM. (C) Scheme of "actin-asters" connecting and forming "web-like" assembly and corresponding confocal image of polymerized "actin-asters" forming "web-like" structures, but with different LUTs to highlight "torus" and "aster" structures. Scale bar, 10 μm. (D) Representative images from the EGFP-Syn1-FL and actin reconstitution acquired using TIRF microscopy show "actin-toruses" and "actin-fibers" as indicated with orange and blue arrows, respectively. Scale bar, 5 μm. Source data are available online for this figure.

All synapsin proteins contain a well-conserved central region, so-called Domain C (Südhof et al, 1989; Benfenati et al, 1992), which was identified as an actin-binding site (Bähler and Greengard, 1987; Petrucci and Morrow, 1987; Petrucci et al, 1988; Bähler et al, 1989). Nucleation-dependent processes, such as actin polymerization, are particularly sensitive to changes in the concentration of G-actin and actin-regulatory proteins (Case et al, 2019). As synapsins both undergo LLPS and bind to actin, SV condensates are poised to act as molecular beacons for the formation of nucleated actin assemblies in an activity-dependent manner. According to the current model of SV cluster/actin interaction, synapsins at low concentrations nucleate actin and, together with SVs, lower the critical concentration of actin required for the assembly of filaments (Bähler and Greengard, 1987; Valtorta et al, 1992). Earlier functional studies led to a model in which actin forms corrals around SV clusters with filaments (so-called actin tracks) extending to both the active zone and endocytic sites (Shupliakov et al, 2002; Bloom et al, 2003; Sankaranarayanan et al, 2003). Synapsin was shown to closely associate with both vesicles in the cluster and actin networks involved in synaptic vesicle recycling (Bloom et al, 2003; Cingolani and Goda, 2008; Rust and Maritzen, 2015; Soykan et al, 2017). Hence, for many years it was thought that the synapsin-actin network occurred through protein-protein crosslinking to assist both vesicle clustering and vesicle recycling (Cesca et al, 2010).

However, this model was established before the LLPS properties of synapsin were known, and indeed, the synapsin-actin crosslinking model at synaptic boutons does not corroborate with several important findings. First, while the acute depletion of synapsins disrupts the clustering of SVs (Pieribone et al, 1995; Pechstein et al, 2020), conversely, disruptions of actin at lamprey and mouse hippocampal synapses do not disperse the distal SV cluster (Bourne et al, 2006; Bloom et al, 2003; Sankaranarayanan et al, 2003). Second, while synapsin colocalizes with the SV clusters, imaging data at lamprey and mouse synapses indicate the vast majority of actin is outside the SV cluster in a torus-like structure at the periactive zone, not precisely overlapping with synapsin signal (Shupliakov et al, 2002; Wilhelm et al, 2014; Ogunmowo et al, 2023). Third, quantitative mass spectrometry indicates that synapsins are present at three times higher concentration than actin at the synaptic bouton (Wilhelm et al, 2014), arguing against them being mere crosslinkers of actin filaments. Fourth, our recent work on synapsin phase separation implies that proteins, including actin, can be recruited to the synapsin-SV phase through both specific protein-protein interactions and transient associations (Milovanovic et al, 2018). In light of these new data, how synapsin-driven SV clustering coordinates with the actin network remains elusive.

Here, we set out to determine the cause–effect relationship between the formation of synapsin-SV condensates and actin organization at the synaptic boutons. Our data show that LLPS of synapsin condensates initiates actin polymerization, allowing for SV:synapsin:actin assemblies to be coupled with cortical actin and actin rings and to facilitate the mesoscale organization of SVs along axons.

## Results

### Synapsin-1 condensates sequester actin monomers and stimulate actin polymerization

We first sought to investigate the role of synapsin-1 condensation on actin polymerization (Fig. 1A–D). To address this question, we first purified EGFP-tagged full-length synapsin-1 (Syn1-FL) using a mammalian expression system (Fig. EV1). We next reconstituted Syn1-FL condensates (final concentration 4 μM) with the crowding agent polyethylene glycol [PEG 8000, final concentration 3% (w/v)] to mimic the crowded environment of the cytoplasm of synaptic boutons. The reaction was set up in a size-exclusion chromatography buffer (hereafter referred to as "reaction buffer", containing 25 mM Tris-HCl (pH 7.4), 150 mM NaCl, and 0.5 mM TCEP). We observed Syn1-FL condensates with diameters ranging from 2 to 4 μm on a glass-bottom dish using a spinning-disk confocal microscope. After 5 min, we added ATTO647-labeled G-actin monomers (final concentration 4 μM) to the preformed Syn1-FL condensates and followed actin polymerization for 35 min (Fig. EV1). We observed robust enrichment of G-actin monomers within Syn1-FL condensates as the fluorescence of actin within Syn1-FL regions increased rapidly over time and plateaued within 20 min (Fig. 1A,C; Movie EV1). After about 20 min, we witnessed polymerized actin within Syn1-FL clusters congregating into radial arrays, which we refer to as "asters" (Fig. 1A; Movie EV1). It is worth highlighting that actin polymerized within Syn1-FL condensates and not within the surrounding solution. Actin polymerization into asters is distinct for Syn1-FL condensates since actin alone did not polymerize into such actin arrays or fibrils in the presence of commercial polymerization buffer (Appendix Fig. S1A). In contrast to EGFP-synapsin-1, actin did not polymerize in the presence of EGFP alone (Appendix Fig. S1B).

At early stages, actin was often observed to be homogenously distributed within Syn1-FL condensates; however, it rearranged towards the rims, forming so-called "torus-like" structures at later stages, followed by dramatic rearrangement into "aster-like" radial assemblies (Fig. 2A,B). The actin-asters emerging from Syn1-FL

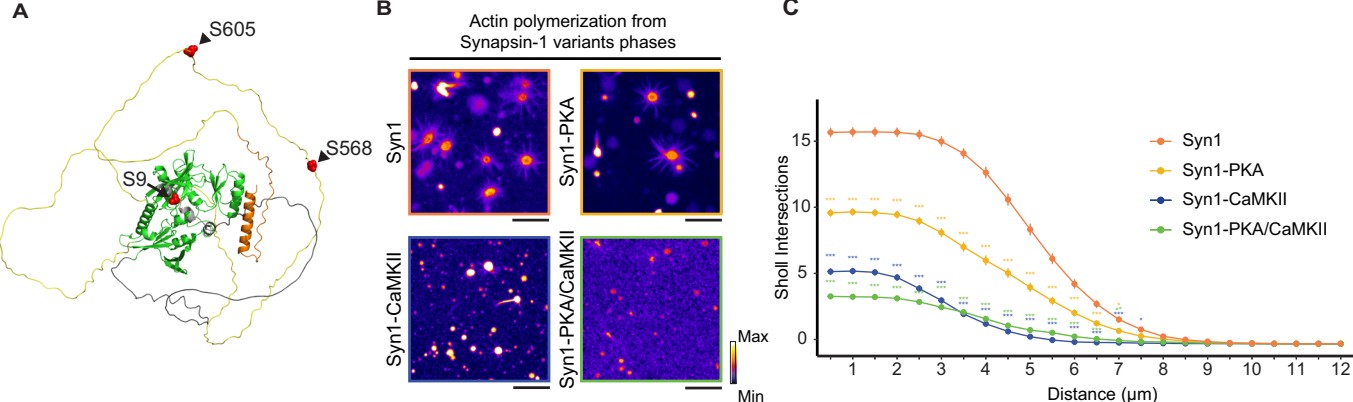

**Figure 3. The morphology of synapsin-1:actin assemblies as a function of synapsin 1 phosphorylation.**

(A) 3D synapsin-1 protein structure prediction obtained from the AlphaFold protein structure database, entry AF-O88935-F1-model_v4. Arrows point to the serine residues that are targeted by PKA (S9) and CaMKII (S568 and S605) in mouse synapsin 1. For the reconstituted protein variants used in this study, Syn1-PKA has serine 9 mutated to glutamate, Syn1-CaMKII has both serines 568 and 605 mutated, and EGFP-Syn1-PKA/CaMKII has all three serines mutated. (B) Representative images for actin polymerization reactions in the presence of purified EGFP-Syn1, EGFP-Syn1-PKA, EGFP-Syn1-CaMKII, and EGFP-Syn1-PKA/CaMKII, each with 3% PEG 8000 in reaction buffer. All images were taken after 40 min of incubation. Images were acquired using a spinning-disk confocal microscope, employing the 647 nm wavelength for actin (represented in green). Scale bars, 10 μm. (C) Graph showing the number of Sholl intersections for polymerized actin in the presence of EGFP-Syn1, EGFP-Syn1-PKA, EGFP-Syn1-CaMKII, and EGFP-Syn1-PKA/CaMKII condensates as a function of distance (in μm). The number of Sholl intersections was counted in 0.5 μm radius increments from the center of the condensates, with at least 100 synapsin:actin assemblies analyzed for each condition. Each dot represents the mean intersection count for each condition. Data from three independent reconstitutions were analyzed. Error bars represent the SEM. For statistical analysis, an R code for linear mixed effects model was applied as in (Jackson et al, 2024); The original code is available at https://zenodo.org/records/1158612 (Wilson et al, 2017). ***$p < 0.001$, **$p < 0.01$, *$p < 0.05$. Source data are available online for this figure.

condensates grew radially and often connected to form networks (Fig. 2C). Interestingly, actin enrichment within the rims persisted even after aster assembly. We found the torus-like distribution of actin within Syn1-FL clusters striking and could also reproduce the same morphology using a different imaging approach—total internal reflection fluorescence (TIRF) microscopy, which allows visualization of structures occurring within <100 nm of the coverslip surface. In addition to the above-mentioned morphologies of polymerized actin, we often observed actin-polymerized and elongated fibers frequently accumulating on the glass surface that were positive for both synapsin-1 and actin (Fig. 2D). Actin co-localized with synapsin condensates and had undergone continuous deformations throughout the assembly of these structures. Together, this implies that Syn1-FL condensates are a porous scaffold that permits continuous flux of G-actin monomers, thereby acting as a reaction center for actin polymerization.

In a series of control reconstitutions, we varied the reaction order by either adding G-actin monomers to Syn1-FL condensates or vice versa. Our data indicate that, irrespective of the reaction order, synapsin condensation results in the sequestering of G-actin monomers and triggers the polymerization of actin in the absence of any additional nucleation factor (Fig. EV2).

## Phase separation and actin binding of synapsin-1 both contribute to actin polymerization

We then set out to assess which domain of synapsin-1 is essential for actin polymerization – domain C that can homo- and hetero-dimerize (Hosaka and Südhof, 1999) or the phase separating IDR region (Milovanovic et al, 2018). We reconstituted actin with synapsin 1-IDR (Syn1-IDR) and synapsin-1-Domain C

(Syn1-Dom. C) in the same manner and followed actin polymerization for 35 min. We observed that the phase-separating Syn1-IDR suffices to polymerize actin. However, the architecture of polymerized actin from Syn1-IDR clusters was completely different from the ones of Syn1-FL condensates, as they appeared more as "networks" rather than "asters" (Fig. 1B; Movie EV2). In contrast, Syn1-Dom. C, which can homo- and hetero-oligomerize but does not phase separate, did not polymerize actin (Fig. EV3A). Given that there was a difference in the morphology of polymerized actin from Syn1-FL and Syn1-IDR phases, we conducted a Sholl analysis on polymerized actin associated with synapsin condensates and quantified the number of intersections of Sholl spheres with fibrils radiating from the synapsin condensates. Indeed, the count of intersections was higher in the case of actin polymerized from Syn1-FL phases as compared to Syn1-IDR (Fig. 1D). Thus, Syn1-IDR alone can accomplish some actin polymerization; however, other domains, including the actin-binding domain C, are needed to achieve the stereotypical aster-like morphology of the actin structures.

To discern whether any protein that can phase separate and enrich actin into its dense phase meshwork will suffice to trigger actin polymerization (Graham et al, 2023), we repeated these experiments using a stress granule protein G3BP2 (Yang et al, 2020). Unlike Syn1, G3BP2 lacks an actin-binding site (Jin et al, 2022). Adding actin to the preformed condensates of G3BP2 led to actin accumulation into condensates, as expected given that the dense phase is a dynamic meshwork (Alshareedah et al, 2021), yet it failed to trigger its polymerization (Fig. EV3B). This clearly indicates that both properties of synapsins—phase separation and the ability to bind actin—are needed to trigger actin polymerization.

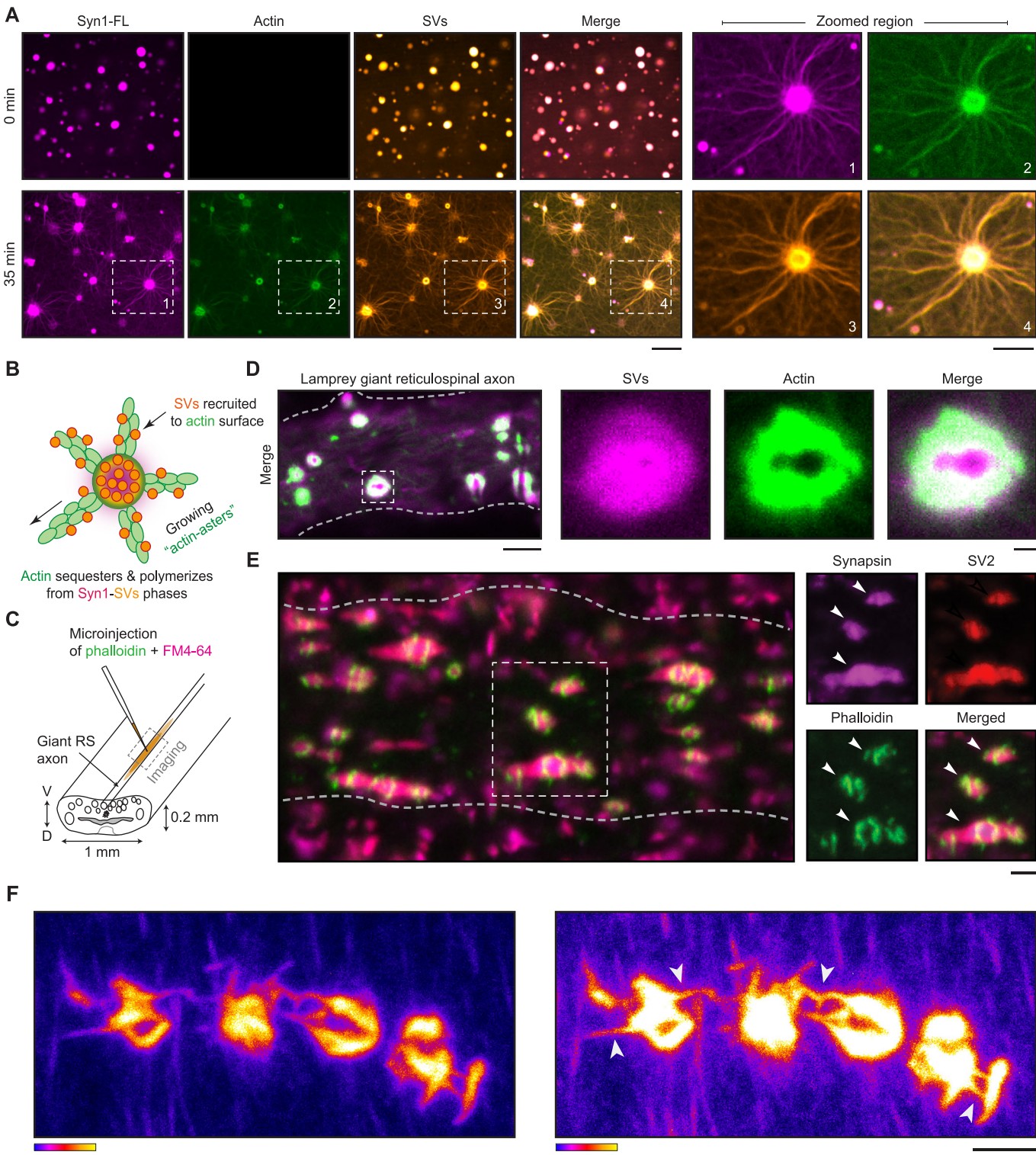

We next wanted to dissect whether the difference in the observed network morphology is due to the different polymerization kinetics of Syn1-FL versus Syn1-IDR. To assess this, we aimed to quantify differences in the kinetics of actin polymerization using a pyrene-actin polymerization assay. For this, we set up reactions of pyrene-labeled actin with Syn1-FL and Syn1-IDR [10 µM pyrene-labeled actin, 10 µM synapsin-1 (either Syn1-FL or Syn1-IDR), 1x

actin polymerization buffer supplemented with 0.5 mM ATP and 3% (w/v) PEG 8000)]. We observed no differences in the kinetics of actin polymerization from Syn1-IDR and Syn1-FL condensates (Fig. 1E), suggesting that the polymerization activity is preserved but that Domain C is crucial for the stereotypical aster-like morphology. Next, we exploited optical density as a measure to quantify the extent of actin polymerization by Syn1-FL. We set up a

**Figure 4. Actin forms toruses at the interface of SV condensates in vitro and in living synapses.**

(A) Left: Representative images of the reconstituted EGFP-Syn1-FL (4 µM, 3% (w/v) PEG 8000) and SVs (3 nM, labeled with FM4-64, 1.65 µM) condensates after adding ATTO647-labeled G-actin monomers (4 µM) at $t = 0$ (top) and 35 min (bottom). Images were acquired using a spinning-disk confocal microscope. Right: Magnified images of polymerized actin-asters formed at 35 min. Scale bar, 5 µm. (B) Scheme of SV recruitment along the actin fibers. (C) Cartoon illustrating the lamprey spinal cord with giant reticulospinal (RS) axons and microinjection strategy. V, ventral; D, dorsal. (D) Laser-scanning confocal microscopy (LSM) image of a live lamprey reticulospinal axon co-injected with FM4-64 and Alexa Fluor™ 488-phalloidin for characterizing SVs and actin, respectively. Excitation wavelengths were 488 nm for phalloidin-actin and 560 nm for SVs labeled with FM4-64. The Inset shows the clear assembly of actin as a torus around SVs. Scale bars, 5 µm (overview) and 1 µm (insets). (E) LSM images of immunolabeled lamprey synapses. Synapses were stained for endogenous synapsin 1, endogenous SV2 (an SV marker), and actin (Alexa Fluor™ 488-phalloidin). Scale bar, 1 µm. (F) High-resolution confocal microscopy maximum z-projection image showing individual synapses connected together by phalloidin-labeled "aster-like" structures (white arrows). The spinal cord was bathed in Ringer solution containing 0.03% DMSO, followed by injection of Alexa Fluor™ 488-phalloidin. The same image is depicted twice using the indicated minimum and maximum values (16-bit scale). Scale bar, 2 µm. Source data are available online for this figure.

reaction of Syn1-FL with actin in polymerizing conditions (4 µM Syn1-FL, 4 µM magnesium-exchanged ATP-actin, 3% (w/v) PEG 8000, 0.5 mM ATP, 140 mM NaCl, and 0.02% sodium azide) in a microtiter plate at 37 °C and followed the turbidity for 48 h at 405 nm. After recording measurements, we visualized the fluorescence of Syn1-FL by microscopy. We noted higher turbidity counts as actin polymerized in the presence of Syn1-FL, while in the absence of either one of them, turbidity stayed minimal (Fig. EV3C,D). Taken together, our data suggest that Syn1-FL condensates act as reaction centers for the recruitment, nucleation, and polymerization of actin into an elaborate network of interconnected actin-asters. Further, the phase separating IDR region of synapsin 1 is essential for actin polymerization.

## Phosphorylation shapes synapsin 1-actin assemblies

Phosphorylation at synapses plays a critical role in regulating synaptic vesicle clustering, with kinases and phosphatases influencing SV mobility via synapsin (Betz and Henkel, 1994; Chi et al, 2001; Chi et al, 2003). Synapsins are, in fact, a phosphorylation hub for multiple kinases and phosphatases. The phosphorylation pattern of synapsin-1 shifts with neuronal activity. During stimulation, calcium influx activates PKA and CaMKII, which phosphorylate synapsin-1, reducing its binding to SVs and causing cluster dispersion (Sihra et al, 1989; Jovanovic et al, 2001). To unravel how the pattern of synapsin-1 phosphorylation that mimics synaptic activity affects synapsin-1-actin assemblies, we cloned PKA (Ser 9), CaMKII (Ser 568 and Ser 605), and double PKA/CaMKII (Ser 9, Ser 568, and Ser 605) phosphomimetics by mutating the target serines to glutamate (Fig. 3A). All these proteins were purified using a mammalian expression system (Fig. EV4A–C). Each of these proteins formed condensates, although the CaMKII- and PKA/CaMKII phosphomimetics condensed less readily (Fig. EV4D,E), in line with the data that CaMKII-driven phosphorylation disrupts synapsin condensation (Milovanovic et al, 2018).

While actin could bind and be enriched in each of these condensates (Fig. 3B), the synapsin-1 phosphomimetics had striking effects on synapsin-1-actin assemblies. Sholl analyses of actin networks formed by these condensates of synapsin 1 indicate that PKA significantly reduces the connectivity and network properties of synapsin-1-actin assemblies. Yet, the CaMKII phosphomimetic, either alone or in combination with the PKA phosphomimetic, led to a dramatic decrease in actin polymerization (Fig. 3C), practically abolishing the ability of synapsin to trigger actin polymerization. These data suggest that phospho-regulation of synapsin-1, by enzymes triggered during synaptic activation, releases actin from synapsin condensates, presumably allowing for actin-driven remodelling processes at the plasma membrane and the periactive zone.

## Synapsin-SV condensates sequester actin both in reconstituted systems and in living synapses

We next examined how synapsin-SVs condensates correspond to actin networks at the synapse. To address this, we reconstituted phases of Syn1-FL (4 µM, 3% (w/v) PEG 8000) with SVs (3 nM, labeled with FM4-64, 1.65 µM) on a glass-bottom dish and imaged using a spinning-disk confocal microscope. After 5 min, we added ATTO647-labeled G-actin monomers (4 µM) to preformed synapsin-SVs condensates and followed actin enrichment within these phases for 35 min. Consistently, actin was sequestered within synapsin-SV condensates and assembled in actin arrays (Fig. 4A,B). The same observation was confirmed in a reconstitution where SVs were added to actin and synapsin, triggering the synapsin-SV condensation, which subsequently led to the sequestering of G-actin and polymerization (Appendix Fig. S2).

We also reconstituted actin with SVs as a control to test if any SV proteins, including the residual synapsins that are peripherally associated with the lipid bilayer of SVs, could promote actin polymerization. Yet, we did not observe actin polymerization in the presence of SVs alone (Fig. EV5A), indicating the central role of synapsins. Note that SV preparation is of high purity, and SVs contain a small fraction of synapsin-1 attached to them (Fig. EV5B), although this amount of synapsin alone was not sufficient to trigger actin polymerization. In line with previous studies (Benfenati et al, 1989; Benfenati et al, 1992), the presence of low concentrations of synapsin-1 enhanced the binding of actin to SVs, as indicated by fluorescence correlation spectroscopy (Fig. EV5C), but was insufficient to trigger polymerization. The actin-binding region is located within the central C domain of synapsin (Fig. EV1) (Petrucci and Morrow, 1987; Bähler et al, 1989). Given that actin polymerized into "network-like" structures from Syn1-IDR phases, we questioned whether we could reproduce actin sequestering and polymerization from Syn1-IDR clusters in the presence of SVs. Indeed, actin was sequestered within Syn1-IDR-SV condensates and polymerized into networks, which also included some aster-like structures (Fig. EV5D). This might be either due to synapsin molecules that remain associated with SVs or due to other putative actin binders present at the surface of SVs.

We then aimed to validate our in vitro observations in vivo using lamprey synapses as a model. Within their spinal cords, lampreys possess a subset of giant reticulospinal (RS) axons

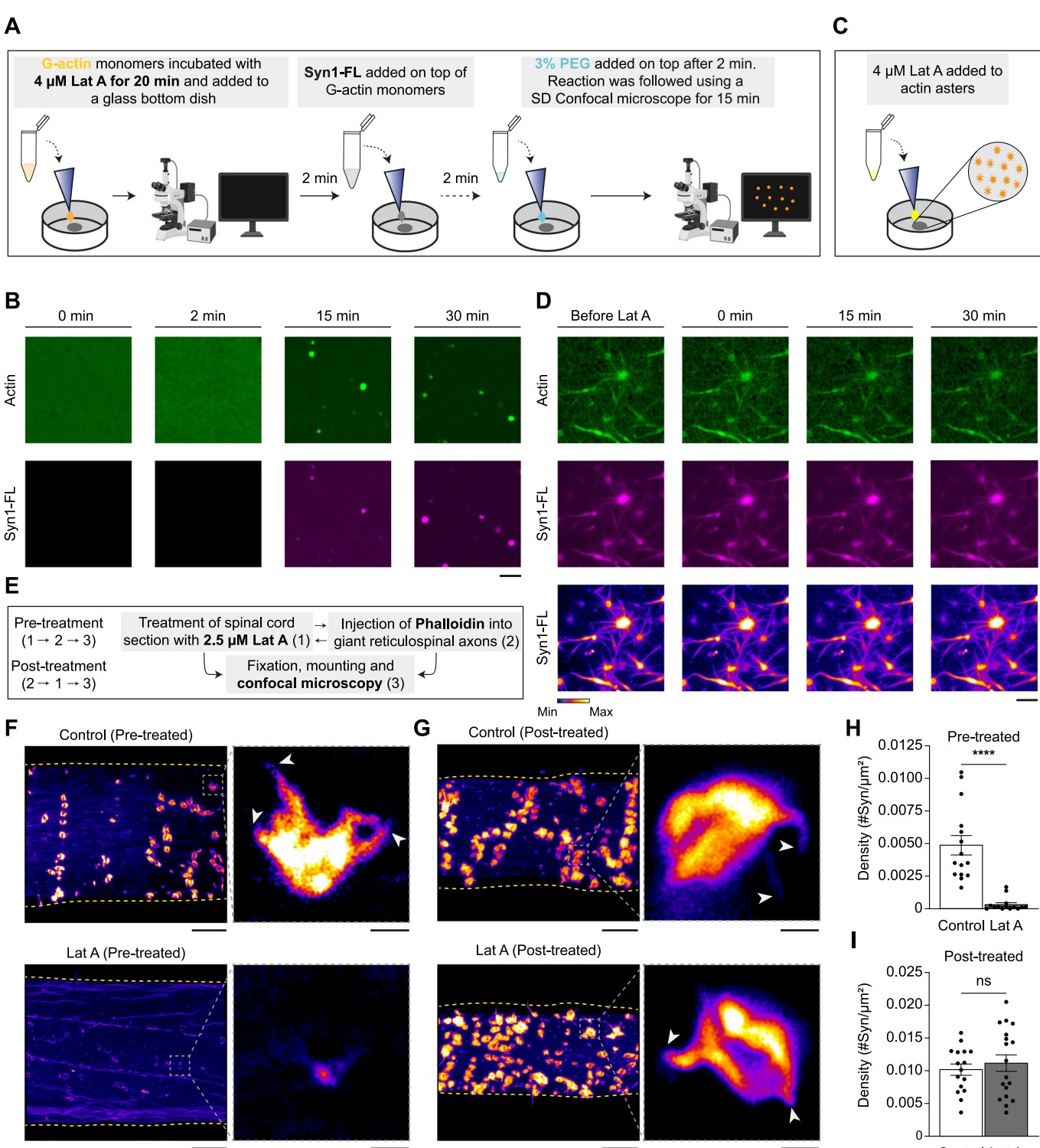

(20–80 μm diameter) with large *en passant* glutamatergic synapses (1–2 μm diameter; 1000–2000 SVs) along the perimeter of the axolemma, allowing experimental accessibility for exploring the mesoscale organization of the presynapse. Consequently, lamprey synapses serve as an excellent model for studying interactions between SVs and actin networks within axons (Shupliakov et al, 2002; Bloom et al, 2003; Bourne et al, 2006). To visualize the presynaptic actin assemblies and SVs inside the giant axons, we

co-microinjected Alexa Fluor™ 488-phalloidin (phalloidin) and FM4-64 for labeling polymerized actin and SVs, respectively. The axons were then imaged live using a laser-scanning confocal microscope within 10 min of the initial injection (Fig. 4C). We observed that actin assembled as "toruses" around the synaptic vesicle cluster (Fig. 4D), as previously demonstrated (Shupliakov et al, 2002; Bloom et al, 2003; Bourne et al, 2006). To confirm the mesoscale organization of the synapsin-SV condensates with actin,

◄

**Figure 5. The effects of the inhibition of actin polymerization on presynaptic actin assemblies in vitro and in living synapses.**

(A) Schematic illustrating the experimental outline for the reconstitution of G-actin monomers pretreated with Latrunculin A (Lat A) with the subsequent addition of synapsin-1 and 3% PEG 8000. (B) Representative maximum intensity projection from confocal microscopy images from the reconstitution of 4 µM Lat A-treated G-actin monomers with synapsin-1 condensates in reaction buffer at $t = 0, 2, 15,$ and 30 min. Images were acquired at 488 and 647 nm wavelengths for EGFP-synapsin 1 and ATTO647 actin, respectively. Scale bar, 3 µm. (C) Schematic illustrating the experimental outline for the addition of 4 µM Lat A on top of preformed "actin-asters" formed from synapsin-1 condensates. (D) Representative maximum intensity projection from confocal microscopy images from the treatment of reconstituted "actin-asters" with 4 µM Lat A. Images were acquired both before and after ($t = 0, 15,$ and 30 min) addition of 4 µM Lat A at 488 and 647 nm wavelengths for EGFP-synapsin-1 and ATTO647 actin, respectively. Scale bar, 3 µm. (E) Schematic illustrating the experimental outline for the Lat A-based experiments in lamprey reticulospinal (RS) axons ex vivo. (F) Treatment of lamprey RS synapses with Lat A before the injection of fluorescently conjugated phalloidin (Alexa Fluor 488-Phalloidin) leads to the disappearance of the presynaptic actin assemblies. Top: Representative maximum intensity z-projection image of presynaptic actin assemblies inside lamprey RS axons pretreated with 0.03% DMSO (Control), followed by microinjection with Alexa Fluor 488-Phalloidin. Bottom: Representative maximum intensity z-projection image showing the absence of pre-synaptic actin assemblies inside lamprey RS axons pretreated with 2.5 µM Lat A in 0.03% DMSO, followed by microinjection with Alexa Fluor 488-Phalloidin. Magnified regions (right) are illustrated in the overview section (left). White arrows point to aster formations. Scale bars = 10 µm (left) and 1 µm (right). (G) Treatment of lamprey RS synapses with Lat A after the injection of Alexa Fluor 488-Phalloidin does not affect pre-synaptic actin assemblies. Top: Representative maximum intensity z-projection image of pre-synaptic actin assemblies inside lamprey RS axons microinjected with Alexa Fluor 488-Phalloidin and post-treated with 0.03% DMSO (Control). Bottom: Representative maximum intensity z-projection image of pre-synaptic actin assemblies inside lamprey RS axons microinjected with Alexa Fluor 488-Phalloidin and post-treated with 2.5 µM Lat A in 0.03% DMSO. Magnified regions (right) are illustrated in the overview sections (left). White arrows point to aster formations. Scale bars = 10 µm (left) and 1 µm (right). (H) Bar chart showing the quantification of the density (number of objects per surface area of axon) of phalloidin-positive objects inside axons pretreated with Lat A (2.5 µM in 0.03% DMSO) or the corresponding control (0.03% DMSO). Each dot represents a z-stack image series through an axon. Error bars represent the standard error of the mean (SEM). Data were from $n = 14$ images; $n = 3$ axons from $N = 3$ animals. For statistical analysis, an unpaired $t$-test was performed in GraphPad PRISM. The $p$ value = $3.48 \times 10^{-6}$; ****$p < 0.0001$ (Pre-treated control-Lat-A). (I) Bar chart showing the quantification of the density of phalloidin-positive objects inside axons post-treated with Latrunculin A (2.5 µM in 0.03% DMSO) or the corresponding control (0.03% DMSO). Each dot represents a z-stack image series through an axon. Error bars represent the standard error of the mean (SEM). Data were from $n = 16$–18 images; $n = 3$–4 axons from $N = 3$–4 animals. For statistical analysis, an unpaired $t$-test was performed in GraphPad PRISM. The $p$ value = 0.53 (Post-treated Control-Lat-A). "ns" indicates not significant. Source data are available online for this figure.

we performed whole-mount immunofluorescence on phalloidin injected axons where synapses were subsequently labeled with antibodies against synapsin-1 and SV2 (a synaptic vesicle marker). We validated the recognition of synapsin-1 in lampreys by the antibody used (Appendix Fig. S3) and found that the synapsin-SV clusters were surrounded by actin toruses (Fig. 4E). Using advanced confocal imaging, we were able to capture high-resolution images of actin toruses around SV clusters and, excitingly, we observed asters emerging from these "actin-toruses" (Fig. 4F). Taken together, these observations imply a role for synapsin:SV condensates in the assembly of presynaptic actin networks.

Next, we set out to investigate the stability of polymerized actin within synapsin-1 condensates and at synapses. We first preincubated G-actin with Latrunculin A (Lat A), a compound well-known for its ability to sequester actin monomers and prevent polymerization (Coué et al, 1987). Subsequently, we added Syn1 and triggered condensation with 3% (w/v) PEG 8000. Indeed, the addition of actin monomers into synapsin condensates supplemented with Lat A prevented actin polymerization (Fig. 5A,B). On the contrary, preformed synapsin-actin assemblies remained stable upon the addition of Lat A (Fig. 5C,D), clearly indicating that synapsin-driven actin polymerization results in the formation of stable actin fibrils.

We also employed acute pharmacological perturbations of the actin cytoskeleton at a lamprey synapse as previously described (Bourne et al, 2006). We treated RS synapses with 2.5 µM Lat A either before or after acute microinjections of phalloidin to label any F-actin present (Fig. 5E). Along the lines of our in vitro experiments, pre-treating the synapses with Lat A disrupted actin polymerization, as shown by a significant decrease in the number of phalloidin-positive presynaptic actin structures as compared to control (Fig. 5F,H). Interestingly, Lat A depleted both the previously described aster-like structures as well as the actin-toruses. In contrast, Lat A was unable to disrupt F-actin structures that were already stabilized with phalloidin (Fig. 5G,I).

Altogether, these results consistently imply that synapsin-SV condensates can sequester G-actin monomers and nucleate actin polymerization in both a minimal reconstitution system and in living lamprey synapses.

## The surface of lipid membranes shapes actin polymerization from synapsin condensates

The synaptic bouton is a densely packed environment with numerous membrane-bound organelles and the presynaptic plasma membrane in the vicinity of the SV clusters (Rizzoli and Betz, 2004; Wu et al, 2017). We next employed Giant Unilamellar Vesicles (GUV) as a model system to characterize the effect of membranes on actin polymerization from synapsin condensates. We reconstituted synapsin condensates as previously described and added them to neutrally charged DOPC GUVs (final lipid concentration: 1.5 µM). After 5 min, when condensates settled on the surfaces of GUVs, we added G-actin monomers spiked with ATTO647 G-actin (final total actin monomer concentration 4 µM) and 0.5 mM ATP to the reaction mix. Similar to our previous observations, actin enriched over time within synapsin condensates forming "aster-like" assemblies (Fig. 6A). Interestingly, we observed that many synapsin-1-actin condensates were inter-connected with each other by actin filaments, and the length of these connecting filaments was several times longer than the filaments going from a condensate to the solution without reaching another condensate (Fig. 6B; Movie EV3).

Interconnected condensates formed elaborate networks, which could span over territories of more than 100 µm² (Fig. 6C). The lipid surface of GUVs acted as a template for actin polymerization stemming from the condensates wetting the GUV membrane (Fig. 6D–G). This suggests that synapsin condensates wetting neutrally charged membranes do not inhibit actin polymerization, and membranes may serve as a guiding surface for (synapsin-bound) actin filaments. Analyzing the similarities between nodes of

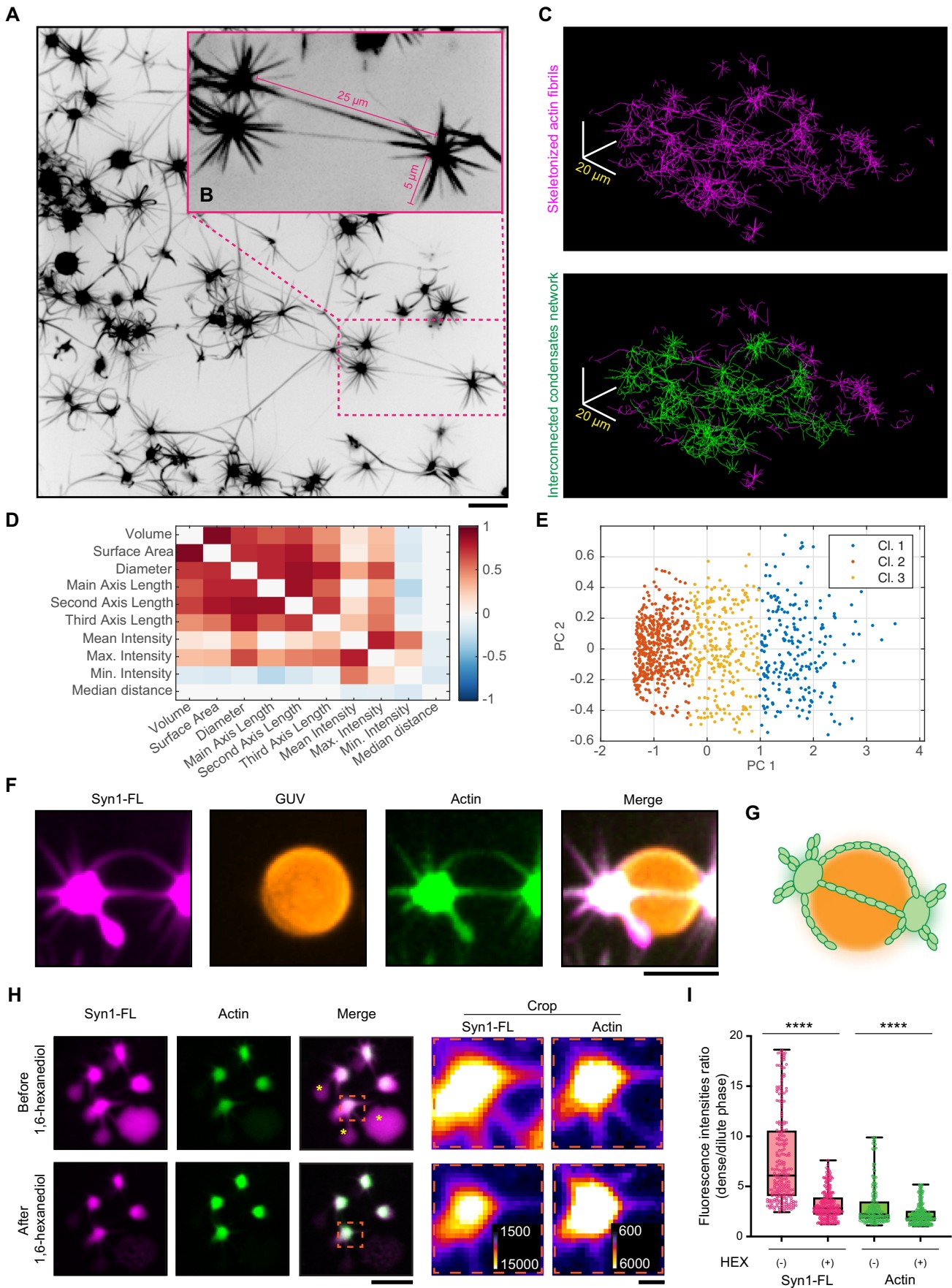

**Figure 6. Actin aster are stable structures connecting neighboring condensates.**

(A) Representative confocal image from the reconstituted EGFP-synapsin-1:actin:GUV assemblies. Projection of the maximal intensity, 30 min post-incubation. Scale bar, 10 μm. (B) Magnified region from (A), indicating the long actin fibers connecting the adjacent condensates. (C) Top: Skeletonized actin fibrils from (A). Bottom: A single continuous network of synapsin condensates interconnected with a continuous actin network is highlighted in green. Scale bar, 20 μm. (D) Heatmap of co-dependencies (Pearson's correlation) for synapsin1:actin assemblies' node properties: (1) volume, (2) surface area, (3) diameter, (4) main-, (5) second-, (6) third- axis length, (7) mean-, (8) maximum, (9) minimum- intensity, (10) median Euclidean distance from other nodes). Data obtained from three independent reconstitutions. (E) Principal component (PC) analysis of synapsin-1:actin assemblies: scatter plot of PC1 vs. PC2 scores (three clusters identified by k-means clustering highlighted in different colors). Data obtained from three independent reconstitutions. (F) Representative confocal image from the reconstituted synapsin-1:actin assemblies spreading over the surface of a GUVs. Projection of the maximal intensity, 30 min post-incubation. Scale bar, 3 μm. (G) Schematic representation of the synapsin:actin:GUV association. (H) Representative confocal images of synapsin:actin assemblies before (top) and after (bottom) the addition of 1,6-hexanediol (10% w/v). The treatment of the synapsin:actin assemblies with 1,6-hexanediol partially disperses synapsin condensates without affecting the preformed actin-asters. Asterisks indicate synapsin condensates without actin, which do disperse upon the addition of 1,6-hexanediol, as expected. Scale bars: 10 μm (overview), 1 μm (magnified). (I) Quantification of the synapsin 1 and actin partitioning before and after treatment with 1,6-hexanediol. Data from three independent reconstitutions, >1500 condensates analyzed for each condition. The box stretches from the 25th to 75th percentile, the dots represent individual data points, the central line shows the median, and the whiskers represent the min and max values. The $p$ values are $1.25 \times 10^{-50}$ for Syn1-FL before vs. after hexanediol and $2.74 \times 10^{-5}$ for actin before vs. after hexanediol. ****$p < 0.0001$; Mann–Whitney $U$-test (two-tailed, non-parametric). Source data are available online for this figure.

the actin network and their relationships, we observed that larger nodes exhibit the highest range of intensities and are the most connected, whereas smaller nodes show the opposite trend (Fig. 6D,E). The complex nature of condensate-actin networks and the significant difference in length between "connecting" and "free" actin filaments may indicate a role of synapsin condensates as a "homing beacon/amplifier" for actin polymerization.

## Polymerized actin network stabilizes synapsin condensates

To characterize the architecture of the polymerized actin assemblies in terms of stability and dependence on the persistent existence of synapsin condensates, we further aimed to disperse polymerized "actin-asters" with 10% 1,6-hexanediol (final concentration), an aliphatic alcohol known to disrupt low-affinity hydrophobic interactions (Ribbeck and Görlich, 2002; Kroschwald et al, 2017). To address this aim, we again reconstituted synapsin condensates with DOPC GUVs and G-actin monomers (spiked with ATTO647 G-actin, as previously described). After 30 min, when "actin-asters" were formed, we added 10% 1,6-hexanediol (final concentration) to the sample. Upon 1,6-hexanediol treatment, synapsin-1-actin condensates that participated in the aster formation had Syn1 partially dispersed, while actin remained unaffected (Fig. 6H). Another aspect of this phenotype is that synapsin remaining after 1,6-hexanediol treatment is attached to the actin core, that is, stabilized. This was confirmed by partitioning analysis, which showed significantly lowered partitioning of synapsin to dense phase, but no difference in actin partitioning (Fig. 6I). Interestingly, synapsin 1-actin condensates without apparent actin polymerization (i.e., no asters) were completely disrupted (Fig. EV6). Thus, where actin is polymerized, synapsin-1:actin assemblies remain undisrupted with Syn1 enrichment within them, whereas there is a complete dispersion of synapsin 1-actin condensates in the absence of actin polymerization, indicating the stabilizing effect of actin polymerization.

## Synapsin condensates are essential for the actin accumulation in synaptic boutons

Several studies indicated that actin accumulates both at the pre- and post-synapse (Hirokawa et al, 1989; Fernández-Busnadiego

et al, 2010; Wilhelm et al, 2014; Helm et al, 2021). At the presynapse, activity-dependent actin remodeling was shown to play a central role in regulating synaptic vesicle recycling (Morales et al, 2000; Bleckert et al, 2012; Soykan et al, 2017; Rust and Maritzen, 2015). To visualize the presynaptic actin, we turned to advanced cryo-electron tomography of hippocampal neurons in culture. In line with previous studies (Fernandez-Busnadiego et al, 2010), cryo-electron tomography showed the presence of filamentous actin in the presynaptic terminal (Fig. 7A; Movie EV4). Indeed, we could observe actin filaments associated with SV clusters (Fig. 7B–D) as well as the filaments in the periactive zone (Fig. 7E). Only a few filaments were detected at the presynapse, likely because the dense presynaptic environment and generally short actin filaments are challenging to visualize in cryo-electron tomograms. Furthermore, at rest, presynaptic actin is largely monomeric, making it elusive in cryo-ET analyses. The development of machine-learning approaches that will require larger datasets and enable more robust analysis will be needed to address these challenges. We further performed expansion microscopy, followed by confocal imaging, and quantified that the presynaptic actin is significantly reduced at the presynapses from synapsin triple knockout (TKO) mice (Fig. 7F,H). To account for synapse heterogeneity, we only analyzed the synaptic structures where both pre- and post-synapses were clearly resolved upon expansion. This effect was specific to the presynaptic compartment as the levels of actin in the postsynapses remained unchanged (Fig. 7G,I).

Previous studies have suggested that perturbation of synapsin function at the active synapse not only led to the drastic reduction of actin at the periphery of the SV cluster but also a corresponding reduction in the number of SVs in the cluster (Pieribone et al, 1995; Bloom et al, 2003); however, neither of the studies examined the effects of synapsin disruptions on synaptic actin. Conversely, synaptic activity and SV recycling critically rely on the local polymerization of actin (Sankaranarayanan et al, 2003; Ogunmowo et al, 2023). We questioned whether synapsin-driven SV condensation was important for actin accumulation at the synaptic boutons. To achieve this, we used mouse hippocampal neurons as a model to assess the role of synapsin phase separation in the context of synapsin-SV interactions with actin. We stained primary hippocampal neurons from WT and synapsin TKO mice for synaptophysin, a well-known integral membrane protein of SVs, and phalloidin-StarRed, which allowed the visualization of remarkable

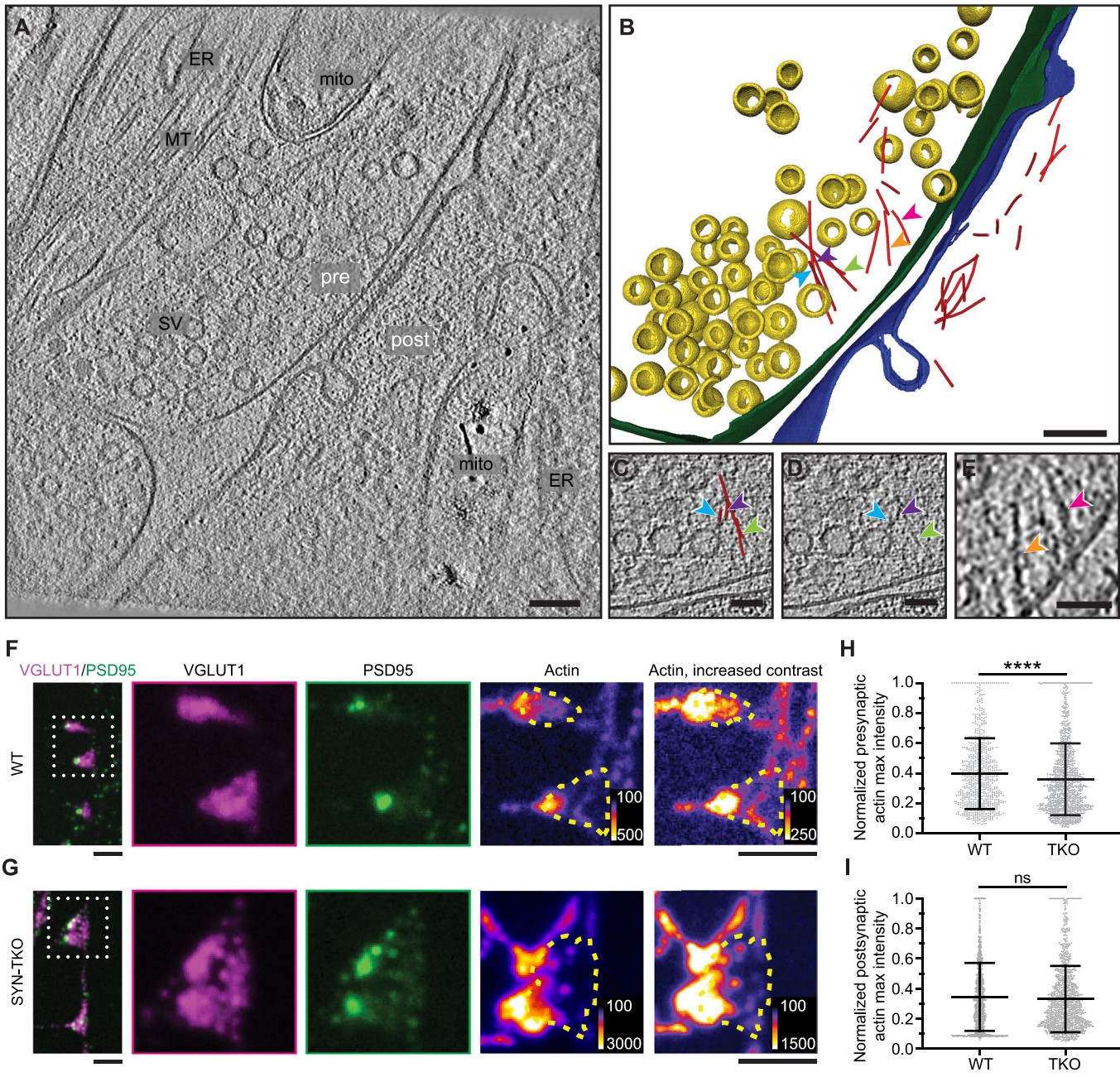

**Figure 7. Visualizing actin at pre- and post-synapses in murine neurons.**

(A) Cryo-electron tomogram of a wild-type synapse. Scale bar, 100 nm. (B) 3D rendering of structures identified in (A). Annotation: yellow, synaptic vesicles (SVs); green, pre-synaptic plasma membrane; blue, post-synaptic plasma membrane; red, actin filament; colored arrowheads point to actin filaments at the presynapse. Scale bar, 100 nm. (C, D) Exemplary region highlighting actin filaments associated with the SV cluster from (A). Note the traced actin filaments (red in C) and original density (arrowheads in D) juxtaposed to SVs. Scale bar, 50 nm. (E) Exemplary region highlighting actin filaments in the periactive zone associated with the plasma membrane. Scale bar, 50 nm. (F) Expansion microscopy images of wild-type hippocampal neurons in culture (14 DIV) immunostained for VGLUT1 (an integral SV protein), PSD95 (post-synaptic protein), and actin. The inset (white dotted line) focuses on resolved pre- and post-synaptic regions. Note the presence of actin in both compartments (yellow line outlines the presynapse). Expansion factor 4.3; scale bar, 1 μm (G). The same as in (F) but for neurons from synapsin triple knockout animals (SynTKO). Expansion factor 4.3; scale bar, 1 μm. (H) Quantification of the actin signal present in VGLUT1-positive regions in synapses from WT and SynTKO neurons. Data from three independent neuronal preparations, 731/1148 synapses analyzed for WT/TKO, respectively. Bars represent mean ± SD; Mann–Whitney $U$-test (two-tailed, non-parametric); $p$ value $= 2.73 \times 10^{-5}$ (WT-TKO). ****$p < 0.0001$. (I) Quantification of the actin signal present in PSD95-positive regions in synapses from WT and SynTKO neurons. Data from three independent neuronal preparations, 1036/1215 synapses analyzed for WT/TKO, respectively. Bars represent mean ± SD; Mann–Whitney $U$-test (two-tailed, non-parametric); $p$ value $= 0.138$ (WT-TKO); n.s., non-significant. Source data are available online for this figure.

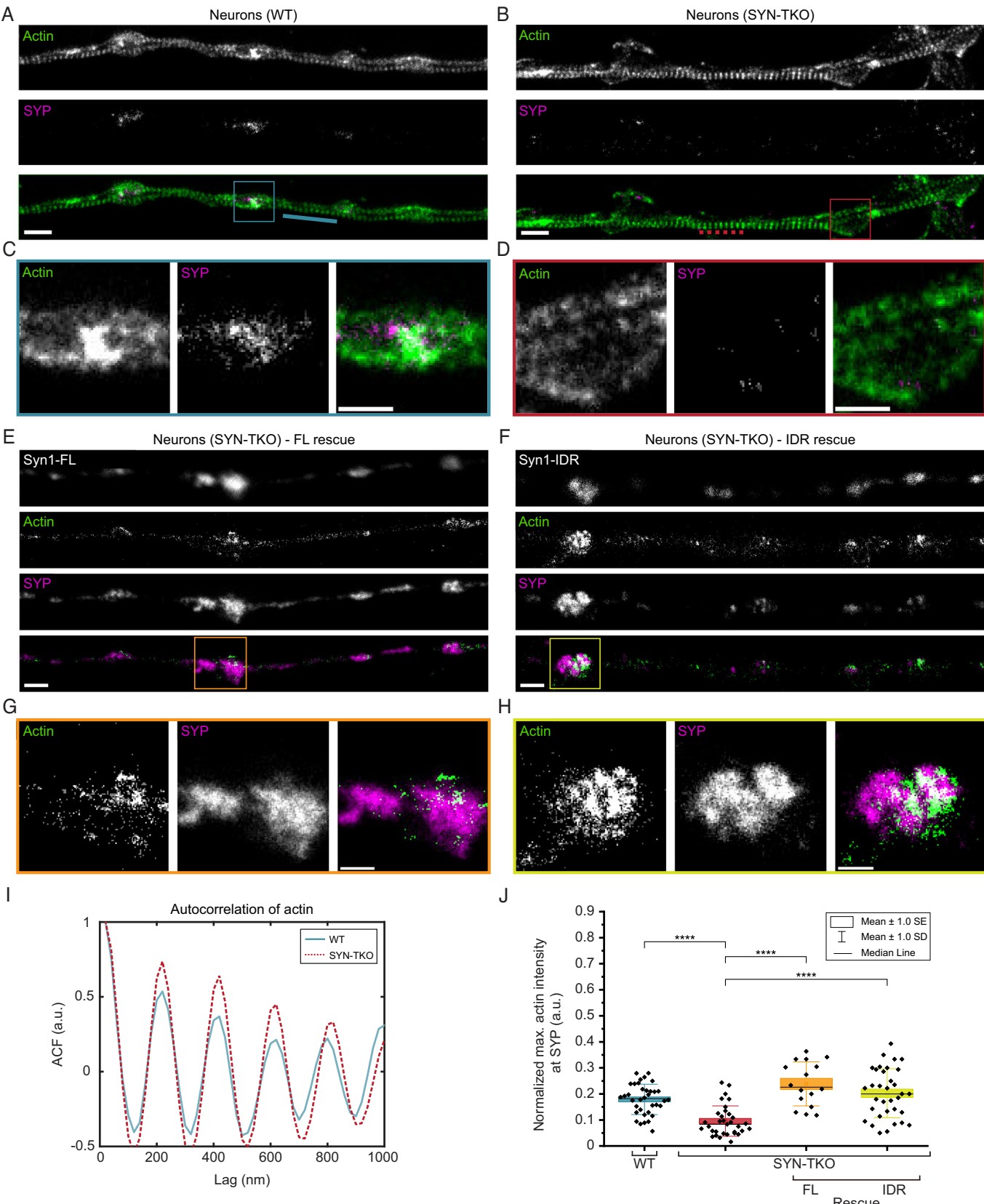

**Figure 8. Synapsin-SV condensates are necessary to sequester actin in living synapses.**

(A, B) Two-color super-resolution STED images of hippocampal neurons (14 DIV) from (A) wild-type and (B) synapsin triple knockout neurons stained with StarRed-phalloidin and anti-synaptophysin (Star580). Scale bars, 1 μm. (C) Magnified region from (A) showing a cohort of synaptic vesicles (anti-synaptophysin) colocalizing with actin (StarRed-phalloidin). Scale bar, 500 nm. (D) Magnified region from (B) indicating a more dispersed signal of SVs (anti-synaptophysin) lacking the colocalization with actin (StarRed-phalloidin). Scale bar, 500 nm. (E, F). Representative images of synapsin triple knockout neurons rescued with (E) full-length EGFP-synapsin-1 or (F) EGFP-synapsin-1 IDR and stained with StarRed-phalloidin and anti-synaptophysin (Star580). Scale bars, 1 μm. (G, H). Magnified regions from (E, F), respectively, showing colocalization of SVs with actin. Scale bars, 500 nm. (I) Autocorrelation of the actin signal along the axons (StarRed-phalloidin; see lines in the images in A, B) indicates that actin rings remain unaltered in the absence of synapsins. (J) Actin enrichment in synaptic vesicle cohorts is defined as a normalized intensity signal of actin channel within regions positive for synaptophysin (wild-type and synapsin triple knockout) or both synaptophysin and EGFP (in rescue experiments). Data from at least three independent neuronal preparations (for WT: $N = 3$; $n = 36$; for SynTKO: $N = 3$; $n = 33$; for SynTKO+FL: $N = 4$; $n = 16$; for SynTKO+IDR: $N = 3$; $n = 34$). Data points present median maximum actin intensity per image; the box represents mean $\pm 1.0$ SE and whiskers represent mean $\pm 1.0$ SD. Mann–Whitney $U$-test (two-tailed, non-parametric); WT-SynTKO ($p = 9.82 \times 10^{-7}$); SynTKO-SynTKO+FL ($p = 2.21 \times 10^{-6}$); SynTKO-SynTKO+IDR ($p = 3.92 \times 10^{-6}$) are all $p < 0.0001$ (****). Source data are available online for this figure.

axonal actin rings using a two-color stimulated depletion emission (STED) microscope (Xu et al, 2013). Furthermore, STED images showed the accumulation of actin within synaptic boutons in WT hippocampal neurons as indicated by a strong actin fluorescence signal within SV clusters (Figs. 8A,C and EV7A). In contrast, actin failed to accumulate at the synaptic boutons of synapsin TKO hippocampal neurons, which also exhibited dispersed SVs due to the absence of synapsin (Figs. 8B,D,J and EV7A). This phenotype was rescued upon transfection with either the full-length EGFP-synapsin-1 or EGFP-synapsin-1 IDR (Figs. 8E–H,J and EV7A,B), which is sufficient to rescue SV clustering (Hoffmann et al, 2023b). Interestingly, the actin rings along the axons remain stable in the absence of synapsins (i.e., in the neurons from synapsin TKO animals; Fig. 8I), suggesting that synapsin/SV condensates are central for the local organization of the actin network in synaptic boutons.

## Discussion

Our results show that the presynaptic actin network can be reconstituted with a surprisingly small number of components, indicating that synapsin-SV condensates act as reaction centers for organizing actin at the synaptic bouton. Actin monomers are first recruited into synapsin-driven condensates; upon reaching a critical concentration within condensates, actin accumulates at the interfaces and polymerizes, without a need for a nucleation factor (Figs. 1 and 2). Similarly to the asters formed by centrosomes for microtubule organization (Woodruff et al, 2015), the synapsin:actin phase directs SVs along the actin fibers, spanning long-range distances of tens of microns (Figs. 4 and 6). The IDR region of synapsin, responsible for its phase separation (Milovanovic et al, 2018), suffices to concentrate actin and trigger its polymerization but does not recapitulate the aster-like morphology and long-range fibers. Our in vitro results faithfully recapitulate the actin morphologies observed in vertebrate synapses (Shupliakov et al, 2002; Bloom et al, 2003; Bourne et al, 2006) (Fig. 4), which demonstrate that actin forms a torus around cohorts of synaptic vesicles with actin fibers radiating from these synapsin:SV:actin assemblies.

Condensation as a mechanism for SV organization relies on the surrounding membrane-bound organelles to provide a template for the actin network assembly. Given the long distance and varying topology of the actin network within synaptic boutons (Bingham et al, 2023; Ogunmowo et al, 2023), our model of synapsin:actin assemblies explains that SV trafficking can occur independently of

the concentration gradient of SVs. Indeed, our super-resolution STED imaging in murine hippocampal neurons indicates that there is an absence of actin enrichment in the vicinity of SVs in the absence of synapsins (Fig. 8).

These new data show that synapsin-SV condensates nucleate and amplify several pools of the presynaptic actin: (1) within the synapsin-SV condensate; (2) in the torus around the SVs (periactive zone); (3) in the asters that extend from the synapsin-SV condensate, some of which extend to other condensates/synapses. This is in line with prior reports showing that there are several functional pools of presynaptic actin regulating exo/endocytosis (Bloom et al, 2003; Bleckert et al, 2012). These different pools of synapsin-actin may help fine-tune neurotransmission at different synapse types or upon different activity levels (Rust and Maritzen, 2015). Our biochemical analysis of the phase separation properties of synapsin-2 suggested a similar phase behavior (Milovanovic et al, 2018). Similarly, synapsins -1 and -2 share sequence characteristics and biophysical properties such as "charge blockiness" and the enrichment with proline and polar residues (Hoffmann et al, 2025). It is important to note that our results were obtained from glutamatergic synapses and that it will be important to explore how this plays out at GABAergic synapses and in different activity states which we did not yet examine and where synapsin-2, another family member of synapsins that undergoes phase separation, plays a major role (Song and Augustine, 2023; Longfield et al, 2024). It is also important to emphasize that the amino acid sequences of synapsin-1, the chemical signatures of synapsin 1-IDRs and their physiological functions at synapses are all highly conserved across vertebrates and invertebrates (Kao et al, 1999; Pieribone et al, 1995; Bloom et al, 2003; Hoffmann et al, 2025), suggesting that these findings are likely to be generalizable across species.

Current thinking regarding the actin involvement in the SV cycle envisions that actin and actin motors would form a range of fixed, distinct structures within a presynaptic compartment, each of which would facilitate a distinct step of the cycle (Cingolani and Goda, 2008; Rust and Maritzen, 2015; Nelson et al, 2013). We propose an alternative, although not mutually exclusive, possibility in which synapsin-driven SV condensates may act as molecular beacons for locally sequestering actin and directing its polymerization. Together, the minimal machinery for recapitulating synapsin:SV:actin assemblies described here explains the role of actin in maintaining the clusters of SVs and directing the traffic of vesicles both within and between boutons during activity. Understanding the coupling of neighboring SV condensates along the axons is central to understanding the circuit function and the coordination of behavior.

# Methods

### Reagents and tools table

| Reagent/resource | Reference or source | Identifier or catalog number |
| --- | --- | --- |
| **Experimental models** | | |
| WT rats (*R. norvegicus*) | Takamori et al, 2006 | |
| C57BL/6 mice (*M. musculus*) | Hoffmann et al, 2023b | |
| Synapsin triple-knockout (SYN-TKO: B6; 129-Syn2tm1Pggd Syn3tm1Pggd Syn1tm1Pggd/Mmjax) mice (*M. musculus*) | Gitler et al, 2004 | |
| Sea lamprey (*P. marinus*) | Bourne et al, 2006 | |
| **Recombinant DNA** | | |
| 6xHis-EGFP-rnSynapsin 1 | Milovanovic et al, 2018 | |
| 6xHis-EGFP(A206K)-hsSynapsin 1 Domain C (a.a. 112–420) | This study | Materials and Methods section |
| 6xHis-EGFP(A206K)-hsSynapsin 1 IDR (a.a. 416–705) | Hoffmann et al, 2023b | |
| 14xHis-SUMO-EGFP-mmSynapsin 1 | This study | Synapsin 1 |
| x14His-SUMO-EGFP-mmSynapsin 1 S9E | This study | Synapsin 1 S9E (PKA) |
| x14His-SUMO-EGFP-mmSynapsin 1 S568/605E | This study | Synapsin 1 S568/605E (CaMKII) |
| x14His-SUMO-EGFP-mmSynapsin 1 S9/568/605E | This study | Synapsin 1 S9E (PKA) and S568/605E (CaMKII) |
| 14xHis-TEV-SENP1_EuB | Addgene | Cat. #149333 |
| **Antibodies** | | |
| Biotinylated anti-synaptotagmin monoclonal antibodies | Synaptic Systems | Cat. #105 011BT |
| Anti-vGLUT1-STAR635P single-domain antibodies | Nanotag | Cat. #N1605 |
| Mouse anti-synaptophysin | Synaptic Systems | Cat. #101011 |
| Goat-anti-mouse-STAR580 | Abberior | Cat. #ST580-1001-500UG |
| Anti-VGLUT1 | Synaptic Systems | Cat. #106008 |
| Anti-PSD95 | Synaptic Systems | Cat. #124308 |
| Anti-β-actin | Proteintech | Cat. #66009-1-Ig |
| Goat-anti-guinea pig AlexaFluor555 | Life Technologies | Cat. #A-21435 |
| Donkey-anti-mouse AlexaFluor488 | Invitrogen | Cat. #A-21202 |
| Donkey-anti-rabbit AlexaFluor647 | Invitrogen | Cat. #A-31573 |
| Mouse monoclonal anti-synaptic vesicle glycoprotein 2 (SV2) | Developmental Hybridoma Studies Bank | AB_2315387 |
| Rabbit monoclonal anti-synapsin 1a/b IDR | Synaptic Systems | Cat. #106 008 |
| Alexa Fluor™647 goat anti-mouse IgG | Thermo Fisher | Cat. #A32728 |
| Alexa Fluor™594 goat anti-rabbit IgG | Thermo Fisher | Cat. # A-11005 |
| **Primer sequences for plasmid construction inserted into DNA sequences of 6xHis-EGFP-BglII...SacI-SV40 expression cassette encoding for human Syn1-Dom.C and Syn1-IDR inserts.** | | |
| #DML0022 + 0023 (A206K) | **FW:** CACCCAGTCC**AAG**CTGAGCAAAGACCCC **RV:** CTTTGCTCAG**CTT**GGACTGGGTGCTCAGG | |

| Reagent/resource | Reference or source | Identifier or catalog number |
| --- | --- | --- |
| **Primer sequences for performing single point mutations (serine to glutamate) on the mouse open reading frame of Syn1 (Uniprot #O88935) to obtain Syn1-phosphomimetics.** | | |
| **Point mutation(s)** | **Forward primer(s)** | **Reverse primer(s)** |
| Synapsin 1 S9E (PKA) | 5'-CGGCGCCGCCTG**GAA**GACAGCAACTTCATGG-3' | 5'-AGTTGCTGTC**TTC**CAGGCGGCGCCGCAG-3' |
| Synapsin 1 S568/605E (CaMKII) | 5'-CCGTCAGGCA**GAA**ATCTCTGGTCC-3' for S568 & 5'-ACTCGTCAGGCC**GAA**CAGGCAGGTCCC-3' for S605 | 5'-CAGAGAT**TTC**TGCCTGACGGGTAG-3' for S568 & 5'-CCTGCCTG**TTC**GGCCTGACGAGTGGG-3' for S605 |
| **Oligonucleotides and other sequence-based reagents** | | |
| Forward primer for EGFP A206K point mutation | This study | #DML0022 |
| Reverse primer for EGFP A206K point mutation | This study | #DML0023 |
| Forward primer for mmSynapsin 1 S9E point mutation | This study | Synapsin 1 S9E (PKA) |
| Reverse primer for mmSynapsin 1 S9E point mutation | This study | Synapsin 1 S9E (PKA) |
| Forward primer for mmSynapsin 1 S568E point mutation | This study | Synapsin 1 S568/605E (CaMKII) |
| Reverse primer for mmSynapsin 1 S568E point mutation | This study | Synapsin 1 S568/605E (CaMKII) |
| Forward primer for mmSynapsin 1 S605E point mutation | This study | Synapsin 1 S568/605E (CaMKII) |
| Reverse primer for mmSynapsin 1 S605E point mutation | This study | Synapsin 1 S568/605E (CaMKII) |
| **Chemicals, Enzymes and other reagents** | | |
| Adenosine triphosphate (ATP) | Roth | Cat. # HN35.1 |
| Chloroform | Roth | Cat. # Y015.1 |
| Dimethyl sulfoxide (DMSO) | Sigma-Aldrich | Cat. # 276855 |
| EDTA-free protease inhibitors | Roche | Cat. # 11836170001 |
| Hydrochloric acid (HCl) | Roth | Cat. # X942.9 |
| HEPES | Roth | Cat. # 6763.2 |
| 1,6-Hexanediol | Sigma-Aldrich | Cat. # 240117-50 G |
| Imidazole | PanReac AppliChem | Cat. # A1073,0500 |
| Magnesium chloride (MgCl$_2$) | Roth | Cat. # KK36.2 |
| Formaldehyde (FA) 16% | PierceTM | Cat. # 28908 |
| Polyethylene glycol (PEG) 8000 | Roth | Cat. # 263.1 |
| Potassium chloride (KCl) | Roth | Cat. # 6781.1 |
| Sodium dodecyl sulfate (SDS) | Roth | Cat. # 2326.2 |
| Sodium Chloride (NaCl) | Roth | Cat. # 3957.3 |
| Potassium hydroxide (KOH) | Roth | Cat. #1310-58-3 |
| TEMED | Roth | Cat. # 2367.3 |
| Tris (2-carboxyethylphosphine-hydrochloride (TCEP) | Sigma | Cat. # C4706-2G |
| Tris | Roth | Cat. # 4855.3 |
| ROTIPHORESE ® Gel 30 (37.5:1) | Roth | Cat. # 3029.1 |

 

| Reagent/resource | Reference or source | Identifier or catalog number |
|---|---|---|
| Expi293™ Expression Medium | Gibco™ | Cat. # A1435101 |
| ExpiFectamine™ 293 Transfection Kit | Gibco™ | Cat. # A14524 |
| Expi293F™ cells | Thermo Fisher | Cat. # A14527 |
| Rosetta (DE3) cells | Novagen | Cat. # 70954 |
| Unlabeled actin | Hypermol | Cat. # 8101-01 |
| ATTO647-labeled G-actin monomers | Hypermol | Cat. # 8158-02 |
| Actin-Toolkit TIRFM | Hypermol | Cat. # 8097-01 |
| Mono-mix buffer | Hypermol | Cat. # 5120-01 |
| ME buffer | Hypermol | Cat. # 5111-01 |
| FM™ 4-64 Dye | Invitrogen™ | Cat. # T13320 |
| 1,2-dioleoyl-sn-glycero-3-phosphocholine (DOPC) | Avanti Polar Lipids | Cat. # 850375 |
| 1,2-dioleoyl-sn-glycero-3-phosphoethanolamine (DOPE) | Avanti Polar Lipids | Cat. # 850725 |
| D-Sorbitol | MP Biomedicals | Cat. # 200-061-5 |
| Commercial actin polymerization buffer FluMaXx | Hypermol | Cat. # 5161-01 |
| Pyrene-labeled actin | Cytoskeleton | Cat. # AKL99 |
| Neutravidin | Thermo Fisher Scientific | Cat. #31055 |
| Alexa Fluor 488-labeled actin | Invitrogen | Cat. # |
| Atto-488 dye | AttoTech | Cat. #AD 488 |
| Latrunculin A (Lat A) | Merck | Cat. # 428026 |
| Hank's balanced salt solution (HBSS) | Gibco | Cat. # 14170112 |
| HEPES buffer | Gibco | Cat. # 15630049 |
| Penicillin/streptomycin | Gibco | Cat. # 15140122 |
| Sodium pyruvate | Gibco | Cat. # 11360039 |
| Magnesium chloride solution | Roth | Cat. # KK36.2 |
| Papaine | Sigma | Cat. # P3125 |
| DNAseI | PanReac AppliChem | Cat. # A3778 |
| Neurobasal Medium A (NB-A) | Gibco | Cat. # 10888022 |
| FBS | Sigma | Cat. # 12106 C |
| B27 | Gibco | Cat. # 17504044 |
| Glutamax | Gibco | Cat. # 35050038 |
| Poly-L-lysine | Sigma | Cat. # P6282 |
| Kynurenic acid | Sigma | Cat. # K3375 |
| Image-iT solution | Thermo Fisher | Cat. #R37602 |
| Phalloidin-STARRED | Abberior | Cat. # STRED-0100-20UG |
| Phalloidin | Thermo Fisher | Cat. #P3457 |
| Alexa Fluor™ 488-conjugated phalloidin | Thermo Fisher | Cat. #A12379 |
| Tricaine methanesulfonate (Syncaine) | Syndel | Cat. #NC0872873 |
| ProLong™ Gold Antifade Mountant | Thermo Fisher Scientific | Cat. #P10144 |
| 6xHis-EGFP-rnSynapsin 1 | This study | Materials and Methods Section |
| 6xHis-EGFP(A206K)-hsSynapsin1-IDR | This study | Materials and Methods Section |
| 6xHis-EGFP(A206K)-hsSynapsin1-Domain C | This study | Materials and Methods Section |

| Reagent/resource | Reference or source | Identifier or catalog number |
|---|---|---|
| EGFP-mMSynapsin1a | This study | Materials and Methods Section |
| EGFP- mMSynapsin1a (S9E) | This study | Materials and Methods Section |
| EGFP- mMSynapsin1a (S568/605E) | This study | Materials and Methods Section |
| EGFP- mMSynapsin1a (S9/568/605E) | This study | Materials and Methods Section |
| His14-TEV-SENP1_EuB | Hoffmann et al, 2023b | |
| **Software** | | |
| Fiji/ImageJ | Rueden et al, 2017 | |
| Prism | GraphPad Software | |
| R (RStudio) | R Core Team | |
| AlphaFold | Jumper et al, 2021 | |
| PyMOL | Schrödinger Inc. | |
| Python | Python Software Foundation | |
| StackRegJ plugin | Slaughter et al, 2013 | |
| Create spectrum jru v1 plugin | Slaughter et al, 2013 | |
| Skeletonize (2D/3D) plugin | | |
| SNT plugin | Arshadi et al, 2021 | |
| MATLAB | The MathWorks | v. R2022b and R2023a |
| OriginPro | OriginLab | |
| **Other** | | |
| HisTrap™HP column | Cytiva | Cat. #17524801 |
| Superdex™ 200 Increase 10/300 column | GE Healthcare | Cat. #28990944 |
| Poly-prep® Chromatography column | Bio-Rad | Cat. #7311550 |
| cOmplete™ His-Tag Purification resin | Roche | Cat. #5893682001 |
| 30 K MWCO Protein Concentrators PES | Pierce™, ThermoScientific™ | Cat. #88502 |
| Expi cell culture flasks (125 ml) | Thermo Fisher Scientific™ | Cat. #4115-0125 |
| 35 mm Glass-bottom dishes | Cellvis | Cat. #D35-20-1.5-N |
| 15-well glass slide | Ibidi | Cat. # 81506 |
| 384-well microtiter plate | Greiner Bio-One | Cat. # 781906 |
| Synergy H1 Hybrid Multi-Mode Microplate Reader | BioTek Instruments | |
| Vesicle Prep Pro device | Nanion Technologies | |
| ITO-coated glass slide | Nanion Technologies | |
| 16-mm O-ring | Nanion Technologies | |
| Glass coverslips (#1 thickness) | Thermo Scientific Technologies | Cat. #3317 |
| 35 mm glass-bottom μ-dishes | Ibidi | Cat. # 81158 |

## Cloning

Plasmid encoding for 6xHis-EGFP-rnSynapsin-1 fusion protein (Syn1-FL) is described previously (Milovanovic et al, 2018). For EGFP-synapsin-1 Domain C and IDR expression plasmids, first, an EGFP A206K mutation was introduced by mutagenesis PCR (#DML0022 and #DML0023) into the Syn1-FL plasmid. Second,

the Syn1-FL sequence was exchanged for human synapsin-1 Domain C (Syn1-Dom.C, a.a. 112–420) or human synapsin-1 IDR (Syn1-IDR, a.a. 416–705) sequences using restriction sites BglII and SacI (Hoffmann et al, 2023b). For details of sequences, please see the Reagents and Tools Table.

Plasmid encoding for 14xHis-SUMO-EGFP-mmSynapsin-1 fusion protein was created by inserting the mouse synapsin-1 open reading frame sequence (Uniprot #O88935) into a 14xHis-SUMO-EGFP-MCS vector by using BglII and SalI restriction sites (Genscript).

To obtain the plasmids encoding for x14His-SUMO-EGFP-mmSynapsin 1 S9E, x14His-SUMO-EGFP-mmSynapsin-1 S568/605E, and 14xHis-SUMO-EGFP-mmSynapsin-1 S9/568/605E, three different phosphomimetic constructs of mmSynapsin 1 (mmSynapsin-1 S9E, mmSynapsin-1 S568/605E, and mmSynapsin-1 S9/568/605E) were created from the mouse synapsin-1 open reading frame sequence (Uniprot #O88935). For details on the primers, please see the Reagents and Tools Table. The obtained constructs were then also inserted into a 14xHis-SUMO-EGFP-MCS vector by using BglII and SalI restriction sites (Genscript). Plasmid encoding for 14xHis-TEV-SENP1_EuB (SUMO protease, pAV0286) was from Addgene #149333.

## Protein expression and purification

*His6-EGFP-rnSynapsin-1 (Syn1-FL), His6-EGFP(A206K)-hsSynapsin-1-Domain C* (112–420), and *His6-EGFP(A206K)-hsSynapsin-1-IDR (421–705) (Syn1-IDR)* were expressed in Expi293F™ cells (ThermoFisher) for 3 days following induction. Cell lysis was performed by three cycles of freezing in liquid nitrogen and thawing at 37 °C in a buffer containing 25 mM Tris-HCl (pH 7.4), 300 mM NaCl, 25 mM imidazole, 0.5 mM TCEP (buffer A), supplemented with EDTA-free Roche Complete protease inhibitors, 10 µg/mL DNase I, and 1 mM MgCl₂. The purification process was conducted at 4 °C as previously described (Hoffmann et al, 2023b; Hoffmann et al, 2021). The lysate was further subjected to centrifugation for 1 h at 20,000 × g, followed by two-step purification. The first step involved affinity purification of soluble supernatant on a Ni-NTA column (HisTrap™HP, Cytiva, ÄKTA pure 25 M). Washing steps were performed with buffer A containing 40 mM imidazole, and elution was performed with buffer A containing 400 mM imidazole. Elution fractions were concentrated (30 K MWCO protein concentrator, Pierce) and subjected to size-exclusion chromatography (Superdex™ 200 Increase 10/300, GE Healthcare, ÄKTA pure 25 M) in 25 mM Tris-HCl (pH 7.4), 150 mM NaCl, and 0.5 mM TCEP. Elution fractions were analyzed by SDS-PAGE. All purified proteins were snap-frozen in liquid nitrogen and stored at −80 °C until use.

*His14-SUMO-EGFP-mmSynapsin-1, 14xHis-SUMO-EGFP-mmSynapsin-1 S9E, 14xHis-SUMO-EGFP-mmSynapsin-1 S568/605E,* and *14xHis-SUMO-EGFP-mmSynapsin 1 S9/568/605E* constructs were also expressed in Expi293F™ cells (Thermo Fisher Scientific) following the manufacturer's instructions for three days after induction and purified as described in the previous section, adding a couple of additional steps. Eluted fractions from size-exclusion chromatography were digested overnight at 4 °C with His-TEV-SENP1 (SUMO protease, protease to protein ratio of 1:20) to cleave the His14-SUMO tag. After overnight digestion, the cleaved tag and the protease were removed by passing three times the protein through a polyprep gravity flow column (Bio-Rad) packed with

complete His-Tag Purification resin (Roche) equilibrated in a buffer consisting of 300 mM NaCl, 25 mM Tris-HCl (pH 7.4), 0.5 mM TCEP, and 15 mM Imidazole. NaCl and Imidazole concentrations in the digested protein fractions were compensated in order to match the buffer composition. The flow-through was dialyzed overnight against 25 mM Tris-HCl (pH 7.4), 150 mM NaCl, 0.5 mM TCEP buffer, and analyzed by SDS-PAGE. All purified proteins were aliquoted, snap-frozen in liquid nitrogen, and stored at −80 °C until further use.

*His14-TEV-SENP1_EuB* (SUMO protease) was expressed in Rosetta (DE3) competent *E. coli* cells and recombinantly purified as described previously (Hoffmann et al, 2023b).

## Synaptic vesicle preparation

All isolations were approved by the Institutional Animal Welfare Committees of the State of Lower Saxony, Germany, and the Max Planck Institute for Multidisciplinary Sciences (Göttingen, DE). Native synaptic vesicles (SVs) from rat brain were prepared following the previously published procedure (Takamori et al, 2006; Nagy et al, 1976; Huttner et al, 1983). All isolation steps were performed at 4 °C. In brief, brains were isolated from 20 adult rats and homogenized in ice-cold sucrose buffer (5 mM HEPES-KOH pH 7.4, 320 mM sucrose supplemented with 0.2 mM PMSF and 1 mg/mL pepstatin A). The homogenate was subjected to centrifugation (10 min at 900g$_{Av}$), and the supernatant was further centrifuged for 10 min at 12,000g$_{Av}$. The pellet was washed once with sucrose buffer and subjected to centrifugation for 15 min at 14,500g$_{Av}$. Synaptosomes were lysed by hypo-osmotic shock, and the lysate was subjected to centrifugation for 20 min at 20,000g$_{Av}$. Free SVs were collected from the resulting supernatant, which was further ultracentrifuged for 2 h (230,000 g$_{Av}$) to obtain a crude SV pellet. Synaptic vesicles were further purified by loading the resuspended pellet (40 mM sucrose) on a continuous sucrose density gradient (50–800 mM sucrose) and further centrifugation for 3 h at 110,880g$_{Av}$. Separation of SVs from residual larger membrane contaminants was achieved by size-exclusion chromatography on controlled pore glass beads (300 nm diameter), equilibrated in glycine buffer (5 mM HEPES-KOH pH 7.40, 300 mM glycine). SV-containing fractions were combined, and synaptic vesicles were collected by centrifugation for 2 h at 230,000g$_{Av}$ and subsequently aliquoted into single-use fractions and snap-frozen in liquid nitrogen.

## In vitro reconstitution experiments

All in vitro reconstitutions had a final concentration of 4 µM EGFP-Syn1-variants (FL, IDR, Dom. C), 3% (w/v) PEG 8000, 3 nM SVs (for reaction setups that required SV assessment), and 4 µM ATTO647 actin. Size-exclusion chromatography buffer (hereafter referred to as "reaction buffer", containing 25 mM Tris-HCl (pH 7.4), 150 mM NaCl, 0.5 mM TCEP) served as a base for all in vitro reconstitutions. Low-binding pipette tips were used for handling protein and SVs. ATTO647-labeled G-actin monomers from rabbit skeletal muscle (Hypermol, 8158-02) were used for assessing G-actin enrichment and polymerization over time within the liquid phases of Syn1-variants. The actin mix was freshly prepared on the day of the experiment following the manufacturer's protocol (Hypermol, 8097-01). Briefly, 10 µM ATTO647-labeled G-actin

monomers were ATP and magnesium-exchanged using monomix (Hypermol, 5120-01) and ME buffer (Hypermol, 5111-01). The final concentration of actin and ATP in all reaction mixtures was 4 µM and 0.5 mM, respectively.

### Reconstitutions of Syn1-variants (Syn1-FL, Syn1-IDR, Syn1-Dom. C, and Syn1-phosphomimetics) with G-actin

For assessing actin enrichment within and polymerization from synapsin 1 phases, 6 µM EGFP-Syn1-variants (FL, IDR, Dom. C, and Syn1-phosphomimetics) were co-incubated with 3% (w/v) PEG 8000 (final concentration) on a glass-bottom dish (Cellvis D35-20-1.5-N). When the condensates became ~2–4 µm in size (approx. after 5 min), ATP and magnesium-exchanged ATTO647-labeled G-actin monomers were added into these preformed condensates. The final reaction mix had 4 µM Syn1-variants, 4 µM actin, and 0.5 mM ATP. Actin enrichment and polymerization were followed for 35 min.

### Reconstitution of synapsin-1-synaptic vesicle condensates with G-actin

The effect of SVs on actin enrichment and polymerization was assessed by co-incubating 6 µM EGFP-Syn1-variants (Syn1-FL and Syn1-IDR) with 3% (w/v) PEG 8000 and 3 nM SVs (final concentration, labeled with 1.65 µM FM4-64) in a tube and placing the mixture on a glass-bottom dish (Cellvis D35-20-1.5-N). ATP and magnesium-exchanged ATTO647-labeled G-actin monomers were added to these preformed Syn1/SVs condensates approximately after 5 min. The final reaction mix had 4 µM Syn1-FL, 4 µM actin, and 0.5 mM ATP. Enrichment of actin within Syn1/SVs phases was followed for 35 min.

### Reconstitution of Syn1-FL condensates and giant unilamellar vesicles (GUVs) with G-actin

For assessing the effect of actin polymerization from synapsin-1 phases in the presence of membranes, synapsin-1 condensates were prepared by co-incubating 10 µM Syn1-FL with 7.5% (w/v) PEG 8000 in the reaction buffer (pH 7.4) for 15 min. Subsequently, preformed synapsin-1 condensates were added from the top onto GUVs pre-loaded on a 15-well glass slide (ibidi, 81506). Condensates were allowed to settle on the surface of GUVs for ~20 min. Eventually, ATP and magnesium-exchanged actin (here, unlabeled actin was spiked with ATTO647-labeled actin; Hypermol, 8101-01 and 8158-02) were added from the top onto the condensate-GUV mix. Thus, the final reaction mix had 3% (w/v) PEG 8000, 4 µM Syn1-FL, 4 µM actin, 74.5 mM NaCl, 0.5 mM ATP, 150 mM sorbitol, and 1.5 mM lipids in the form of GUVs. Actin polymerization from synapsin-1 condensates in the presence of GUV was followed after 30 min. Reactions involving 1,6-hexanediol were conducted on a glass-bottom dish (Cellvis D35-20-1.5-N). Briefly, after actin polymerization (~30 min), 1,6-hexanediol (10% final concentration) was added to the reaction mixture, followed by a gentle mix and 5 min of incubation. Images were acquired both before and after the addition of 1,6-hexanediol at the same locations.

### Validation of actin polymerization
We performed four control experiments.

First, G-actin polymerization was validated in the presence of the final purification buffer (i.e., negative control) and in the commercial polymerization buffer, FluMaXx (Hypermol, 5161-01; positive control). In both of the conditions, ATP and magnesium-exchanged ATTO647-labeled G-actin monomers (final concentration 4 µM) were allowed to polymerize on a glass-bottom dish in respective buffers. Both reactions were followed for 45 min.

Second, we tested actin polymerization in the presence of 6x-His-EGFP (4 µM) for 45 min to quantify whether the tag affects actin polymerization. Briefly, we prepared a pre-mix of 6x-His-EGFP (final concentration 4 µM) with 3% (w/v) PEG 8000. ATP and magnesium-exchanged ATTO647-labeled G-actin monomers (final concentration 4 µM) were added after 5 min to the reaction mixture, and the reaction was followed for 45 min.

Third, we tested actin polymerization in the presence of SVs in the reaction buffer (25 mM Tris-HCl (pH 7.4), 150 mM NaCl, and 0.5 mM TCEP) for 45 min to validate if any SV-bound proteins could direct actin polymerization. Here, we prepared a pre-mix of 3 nM SVs (labeled with 1.65 µM FM4-64) and 3% (w/v) PEG 8000. Subsequently, ATP and magnesium-exchanged ATTO647-labeled G-actin monomers were added after 5 min, and the reaction was followed for 45 min.

Fourth, actin polymerization was also examined in the presence of G3BP2 condensates. Here, G3BP2 condensates were formed by mixing G3BP2 (4 µM) with 3% (w/v) PEG 8000. ATP and magnesium-exchanged ATTO647-labeled G-actin monomers (final concentration 4 µM) were added to the preformed condensates, and the reaction was followed for 45 min.

## Pyrene-actin assay

To deduce the effect of synapsin-1 condensation on the kinetics of actin polymerization, we set up reactions of pyrene-labeled actin (Cytoskeleton, AKL99) with EGFP-Syn 1 variants (Syn1-FL and Syn1-IDR) in the absence of a crowding agent. Briefly, reaction mixtures containing 10 µM pyrene-labeled actin, 10 µM synapsin-1 variants, 1X actin polymerization buffer (10 mM Tris-HCl, pH = 7.5, 2 mM MgCl$_2$, 50 mM KCl, and 0.5 mM ATP) were set up in a 384-well microtiter plate (Greiner Bio-One, 781906). For all reaction conditions, 1x Monomix buffer was used as a base for adjusting the reaction volume to 30 µL. Pyrene fluorescence was measured at an emission wavelength of 410 nm (excitation wavelength 365 nm) every 30 s for at least 15 min at 37 °C using a Synergy H1 Hybrid Multi-Mode Microplate Reader (BioTek Instruments). Measurements obtained from the pyrene-actin assay for each condition were baseline corrected using the average fluorescence value of pyrene-actin in 1x Monomix buffer. Further, data points were plotted using GraphPad Prism software.

## Turbidity measurements

To examine the effect of synapsin-1 condensation on actin polymerization, we set up reactions of actin (Hypermol, 8101-01) with EGFP-Syn1-FL. The changes in turbidity reflect the rate and extent of condensate formation (Hoffmann et al, 2021). Here, we used the further changes in optical density as a proxy of actin polymerization. Reaction setups for polymerizing conditions comprised of 4 µM EGFP-Syn1-FL, 4 µM ATP and magnesium-exchanged-actin, 3% (w/v) PEG 8000, 0.5 mM ATP, 140 mM NaCl, and 0.02% sodium azide. The following control reactions were also undertaken: (i) EGFP-Syn1-FL in 1X Monomix and 3% (w/v) PEG

8000 and (ii) Actin and 3% (w/v) PEG 8000. The reaction buffer (pH 7.4) was used as a base for adjusting the reaction volume to 30 μL for all reactions. After mixing, all reactions were transferred into a 384-well microtiter plate (Greiner Bio-One, 781906) and turbidity was measured every 10 min for each condition at 405 nm and 37 °C in shaking mode for at least 48 h using a Synergy H1 Hybrid Multi-Mode Microplate Reader (BioTek Instruments). Measurements from the microplate reader were baseline corrected (baseline was determined as the turbidity of the reaction buffer), and plots were prepared using GraphPad Prism software. After measurements, all conditions were imaged using a Nikon Eclipse TS2-FL microscope, capturing widefield signal and GFP-epifluorescence.

## Preparation of giant unilamellar vesicles (GUVs)

Lipids used for GUV preparation were purchased from Avanti Polar Lipids. Neutrally charged 1,2-dioleoyl-*sn*-glycero-3-phosphocholine (DOPC) GUVs were prepared by electroformation in the Vesicle Prep Pro device (Nanion Technologies GmbH) on the conductive surface of indium tin oxide (ITO)-coated glass slides, as described previously (Hoffmann et al, 2023a). The total lipid concentration was 5 mM. For labeling, lipids were spiked with 1,2-dioleoyl-sn-glycero-3-phosphoethanolamine (DOPE)-Rhodamine Red (10 μM final concentration). Lipids were first mixed with chloroform for a 25-μL total volume and applied in a thin, even film of circular shape on a conductive surface of the ITO-coated glass slides. This film was left at room temperature for drying for 20 min. Subsequently, a 16-mm O-ring (Nanion Technologies GmbH) was placed around the dried film, and the entire assembly was placed inside the Vesicle Prep Pro device. Further, 250 μL of an aqueous solution containing 500 mM D-Sorbitol (MP Biomedicals) was pipetted inside the O-ring. This O-ring was covered with a second ITO-coated glass slide in such a way that the O-ring and solution inside it were sandwiched between conductive surfaces. A program was run on the Vesicle Prep Pro device (amplitude = 3 V; frequency = 5 Hz; temperature = 36 °C; time = 2 h; ramps = 5 min). Afterwards, the upper glass slide was removed from the assembly, and the solution inside the O-ring was transferred to a 1.5 ml reaction tube. The vesicle solutions were stored at 4 °C and subsequently used for experiments within 1 week after preparation.

## Fluorescence correlation spectroscopy

Fluorescence correlation spectroscopy (FCS) was conducted using a confocal microscope (Perego et al, 2020). The setup was based on an inverted microscope (Olympus IX73, Olympus Europa SE & CO. KG, Hamburg, Germany) and equipped with two pulsed diode lasers (Cobolt Samba-532 100 mW and Cobolt Calypso-491 25 mW, Cobolt AB, Solna, Sweden), a 60× water immersion objective (UPlanApo, NA = 1.2, Olympus), and two avalanche photodiodes (tau-SPAD, Picoquant GmbH, Berlin, Germany). Additionally, the microscope was equipped for epifluorescence microscopy. The SPADs were connected to a digital correlator card (ALV-7004 USB, ALV-Laser Vertriebsgesellschaft mbH, Langen, Germany), and the resulting correlation measurements were analyzed in a post-processing phase using a custom Python script (Python Software Foundation, https://www.python.org).

Synaptic vesicles were patterned as described previously (Perego et al, 2020). Briefly, the vesicles were immobilized on glass coverslips (#1 thickness, Thermo Scientific Technologies Inc., Wilmington, USA) by adding neutravidin (0.05 g/L, Thermo Fisher Scientific, Waltham, MA, USA) and biotinylated anti-synaptotagmin monoclonal antibodies (0.01 g/L, Synaptic Systems GmbH, Göttingen, Germany). The vesicles were labeled with anti-vGLUT1-STAR635P single-domain antibodies (0.05 g/L, Nanotag, Göttingen, Germany) for visualization. For the FCS measurement, approximately 250 μL of either only Alexa Fluor 488-labeled actin (1 mg/L, actin from rabbit muscle, Invitrogen, Carlsbad, CA, USA) or in combination with synapsin (0.1 mg/L), were applied to the patterned vesicle sample. Fluorescence signals were collected using the confocal modality for 15 min, divided into 1-min portions, and averaged to generate the final fluorescence correlation curve. Data portions exhibiting the passage of aggregates or artifacts were excluded from the analysis. The observation volume was calibrated with Atto-488 (AttoTech GmbH, Siegen, Germany) before the fluorescence correlation experiment, resulting in an observation volume of $(280 \pm 10)$ nm. All correlation curves were fitted using a Levenberg–Marquardt nonlinear least-square routine.

## Microscopy

### Spinning-disk confocal microscopy

Imaging of the in vitro reconstitution experiments was performed on a Nikon spinning-disk confocal CSU-X (SDC CSU-X) microscope equipped with two EMCCD cameras (iXon3 DU-888 Ultra), Andor Revolution SD System (CSU-X), and a PL APO163 60x/1.4 NA, oil immersion objective. Excitation wavelengths were: 488 nm for EGFP-fusion proteins (synapsin-1 variants, G3BP2 and EGFP-control), 561 nm for GUVs and SVs (GUVs: (DOPE)-Rhodamine Red, SV: FM4-64), and 647 nm for ATTO647-labeled actin.

Imaging of the reconstitutions involving phosphomimetics assessment, Latrunculin treatment, and assessment of reaction orders was done using a Nikon spinning-disk confocal CSU-X (SDC CSU-X) microscope with pco.edge 4.2 bi sCMOS camera detector and a PL APO163 60x/1.4 NA, oil immersion objective.

### Total internal reflection fluorescence (TIRF) microscopy

TIRF microscopy was performed on a Nikon Ti-E TIRF/STORM setup equipped with a sCMOS camera (Photometrics Prime 95B (95% QE, 1200 × 1200 pixels, 11 μm pixel size, 41 fps full frame, 82 fps@12-bit), and a Plan Apo VC 100x/1.49 NA DIC H N2 TIRF oil immersion objective. The excitation wavelength was 647 nm for ATTO647-labeled actin.

### Confocal laser-scanning microscopy and Airyscan

Images of fixed lamprey samples were acquired using a Zeiss LSM780 confocal microscope (Plan-Apochromat 40x/1.4 objective).

### Two-color stimulated emission depletion (STED)

Two-color STED microscopy was performed on an Abberior STED system with a 100×/1.4 NA oil UPlanSApo Olympus objective. A single focal plane in the center of neuronal processes was imaged. Pixel size was set to 20 nm and pixel dwell time to 10 μs.

### Spinning-disk confocal expansion microscopy

Expanded hydrogels of murine hippocampal neurons were mounted in 35 mm glass-bottom µ-dishes (ibidi, 81158) with 2% agarose. Representative images were acquired on an inverted Olympus IX71 microscope equipped with a CSU-X1 spinning disk unit (Yokogawa) and an ORCA Flash 4.0LT sCMOS camera (Hamamatsu). An Olympus 60×/1.42 NA oil immersion objective was used. Excitation wavelengths were 491 nm for β-actin, 561 nm for PSD95, and 635 nm for VGLUT1. To calculate the scale, the pixel size of the camera was divided by the measured gel expansion factor (~4.3).

## Primary hippocampal neurons

### Preparation of hippocampal neurons

All animal experiments were approved by the Institutional Animal Welfare Committees of the State of Berlin, Germany, and Charité University Clinic (Berlin, DE). Hippocampal neurons were prepared following an established procedure (Hoffmann et al, 2023b). Briefly, brains of P0/1 mice of WT (C57BL/6) and synapsin triple-knockout (SYN-TKO: B6; 129-Syn2tm1Pggd Syn3tm1Pggd Syn1tm1Pggd/Mmjax (Gitler et al, 2004) background were dissected, followed by hippocampi extraction. The hippocampi were washed with cold supplemented Hank's balanced salt solution [HBSS-supp, Gibco, 14170112; supplemented with 10 mM HEPES (Gibco, 15630049), 50 units/ml penicillin/streptomycin (Gibco, 15140122), 1 mM sodium pyruvate (Gibco, 11360039), and 6 mM $MgCl_2$ (Roth, KK36.2)]. Subsequently, hippocampi were digested with Papaine (Sigma, P3125) and DNAse I (PanReac AppliChem, A3778) in HBSS-supp for 20 min at 37 °C. The enzymatic reaction was inactivated by washing the cells with plating medium [Neurobasal Medium A (NB-A Gibco; 10888022), supplemented with 5% FBS (Sigma, 12106 C), 1% B27 (Gibco, 17504044), 1% Glutamax (Gibco, 35050038), and 50 units/ml penicillin/strepto-mycin]. The digested tissue was subjected to gentle mechanical dissociation using a P1000 pipette and cells were seeded in a plating medium on PLL-coated glass coverslips (0.1 mg/mL poly-L-lysine, Sigma, P6282). Hippocampal neuronal cultures were maintained at 37 °C temperature and 5% $CO_2$ in growth medium (NB-A supplemented with 1% B27, 1% Glutamax, and 50 units/ml penicillin/streptomycin).

### Transfection of hippocampal neurons

Calcium phosphate transfection protocol for neurons was adapted from (Jiang and Chen, 2006). Briefly, a coverslip containing DIV4-6 hippocampal neurons was transferred to a fresh petri dish containing 1 ml of growth medium supplemented with 4 mM kynurenic acid (Sigma, K3375, 20 mM stock solution in NB-A, freshly prepared). DNA mix was composed of 2 µg of the plasmid of interest in 1x TE (10 mM Tris-HCl (pH 7.3) and 1 mM EDTA) supplemented with $CaCl_2$ to a final concentration of 250 mM (stock: 10 mM HEPES (pH 7.2), 2.5 M $CaCl_2$). To prepare the transfection mix, one-eighth of the DNA mix was added stepwise to 2xHEBS (42 mM HEPES (pH 7.2), 274 mM NaCl, 10 mM KCl, 1.4 mM $Na_2HPO_4$, and 10 mM glucose) with slow vortexing (~60 rpm) for 2–3 s between each addition and incubated for 20 min at room temperature. The transfection mix was added to neurons, and dishes were incubated at 37 °C, 5% $CO_2$ for 1.5 h. Subsequently, the medium was replaced by 1 mL of NB-A with 4 mM kynurenic acid

supplemented with 2.5 mM HCl, followed by an incubation at 37 °C, 5% $CO_2$ for 15 min. After the wash step, the coverslip was placed back in the primary feeding culture dish with its own conditioned growth medium.

### Sample preparation for two-color stimulated emission depletion (STED)

Samples of mouse hippocampal neurons (DIV14) were fixed, immunostained, and imaged as previously described (Rentsch et al, 2024). Cells from at least three independent preparations per condition (WT, SYN-TKO, FL-rescue, IDR-rescue) were fixed in 4% PFA (v/v) in PBS for 15 min. Samples were quenched in 50 mM NH4Cl/PBS for 30 min, followed by three brief washes with PBS. A permeabilization/blocking step was performed with 1% BSA (w/v), 0.05% Saponin (w/v), and 4% Horse serum (v/v) in PBS for 45 min. Before antibody staining, the samples were incubated in a drop of Image-iT (Thermo Fisher, #R37602) for 30 min. Samples were stained with mouse anti-synaptophysin (Synaptic Systems 101 011, 1:50) overnight at 4 °C in 1% BSA (w/v), 0.05% Saponin (w/v), and 4% Horse serum (v/v) in PBS. Samples were washed three times with 1% BSA (w/v), 0.05% Saponin (w/v) in PBS for 5 min with agitation. Subsequently, samples were incubated with goat anti-mouse-STAR580 (Abberior, ST580-1001-500UG, 1:50) and phalloidin-STARRED (Abberior, STRED-0100-20UG, 1 g/L, 1:200) for 1 h in 1% BSA (w/v), 0.05% Saponin (w/v), and 4% Horse serum (v/v) in PBS. After secondary antibody incubation, the samples were washed three times with PBS and analyzed by two-color STED microscopy.

### Ultrastructure expansion microscopy (U-ExM) of murine hippocampal neurons

Samples were transformed into expandable hydrogels following the U-ExM protocol (Gambarotto et al, 2019; Gambarotto et al, 2020) (w/w) acrylamide (AA), 0.1% (wt/wt) N,N'-methylenebisacrylamide (BIS) in PBS, was prepared 1 day before the gelation and stored at −20 °C until use. Mouse hippocampal neurons grown on 18 mm coverslips (DIV14) were fixed in 4% PFA (v/v)/PBS for 15 min at room temperature and quenched in 50 mM NH4Cl/PBS for 30 min, followed by three brief washes with PBS. Coverslips were then incubated with 1 mL of anchoring solution (0.7% formaldehyde (FA), 1% AA in PBS) at RT for 16 h. For gel polymerization, 90 µL of monomer solution were mixed with 5 µL of 10% N,N,N',N'-tetramethylethylenediamine (TEMED) and 5 µL of 10% ammonium persulfate (APS) (0.5% APS and TEMED in final monomer solution). The coverslips were placed on 80 µL drops of monomer solution in a humid chamber and incubated on ice for 5 min, followed by incubation at 37 °C for 1 h. Gels (still on coverslips) were placed in 2 mL denaturing buffer (200 mM SDS, 200 mM NaCl, 50 mM Tris, and pH 9) in a six-well plate for 15 min at RT with agitation. For denaturation, gels were transferred into 15 mL centrifuge tubes filled with 3 mL of denaturation buffer and incubated at 95 °C for 1 h. Subsequently, the gels were washed several times with MilliQ water and then expanded in MilliQ water at 4 °C overnight.

The gel diameter was measured the following day to calculate the expansion factor. Gels were cut into 2 cm squares, and the squares were incubated twice in PBS for 15 min. The shrunken gel pieces were stained with 300 µL antibody solution diluted in 2% BSA/PBS. Subsequently, the gels were stained with antibodies against VGLUT1 (Synaptic Systems, 106008, 1:400), PSD95

(Synaptic Systems, 124308, 1:400), and β-actin (Proteintech, 66009-1-Ig, 1:400) for 3 h at 37 °C while slightly agitated at 80 rpm. Gels were then washed three times for 20 min with PBS-T (PBS supplemented with 0.1% Tween 20) in a 6-well plate with gentle agitation. Secondary antibody (goat-anti-guinea pig AlexaFluor555 [Life Technologies, A-21435, 1:250], donkey-anti-mouse Alexa-Fluor488 [Invitrogen, A-21202, 1:250], donkey-anti-rabbit Alexa-Fluor647 [Invitrogen, A-31573, 1:250]) incubation was executed as indicated above for 2.5 h, followed by a second series of washing with PBS-T. For the final expansion, the gels were placed in MilliQ water, washed several times, and left for full expansion at 4 °C overnight.

## Lamprey tissue preparation

### Axonal microinjections

All lamprey animal procedures were approved by the Institutional Animal Care and Use Committee at the Marine Biological Laboratory in Woods Hole, MA (USA), in accordance with standards set by the National Institutes of Health. The sea lamprey (*Petromyzon marinus*) spinal cord was prepared as described (Bourne et al, 2006; Walsh et al, 2018). Briefly, late-stage larval lampreys (10–13 cm; M/F) were anesthetized in 0.1 g/L tricaine methanesulfonate (Syncaine; Syndel). Next, 2–3 cm sections of the lamprey spinal cord were dissected and pinned ventral side up in a Sylgard-lined dish (Ellsworth Adhesives). Dissected spinal cords were submerged in fresh, oxygenated lamprey Ringer solution (100 mM NaCl, 2.1 mM KCl, 1.8 mM MgCl$_2$, 4 mM sucrose, 2 mM HEPES, 0.5 mM L-glutamine, 2.6 mM CaCl$_2$, pH 7.4). To label presynaptic actin, giant reticulospinal (RS) axons were microinjected for 3–5 min with Alexa Fluor™ 488-conjugated phalloidin (Thermo Fisher) diluted in lamprey internal solution to 100 μM (10 mM HEPES-KOH, 180 mM KCl, and pH 7.4). For the live imaging experiments, RS axons were co-injected with FM4-64, a synaptic vesicle label, and phalloidin (Thermo Fisher) and then imaged 10–15 min later using a Zeiss LSM780 laser-scanning confocal microscope (Plan-Apochromat 40x/1.4 objective, digital zoom 5.2).

## Whole-mount immunofluorescence

After microinjection of phalloidin, spinal cord preparations were fixed overnight in 4% paraformaldehyde in 0.1 M PBS, pH 7.4, and subsequently processed for whole-mount immunofluorescence as previously described (Wallace et al, 2024; Roman-Vendrell et al, 2024). The primary antibody used for labeling SV clusters was a mouse monoclonal made against synaptic vesicle glycoprotein 2 (SV2; Developmental Hybridoma Studies Bank; 1:100), which was previously characterized in lamprey nervous tissues (Busch and Morgan, 2012; Banks et al, 2020; Wallace et al, 2024). Synapsin condensates were co-labeled using a rabbit monoclonal antibody made against the IDR of synapsin 1a/b (Synaptic Systems 106 008; 1:100), which was characterized in Appendix Fig. S3. Secondary antibodies used were Alexa Fluor™ plus 647 goat anti-mouse IgG (H + L) and Alexa Fluor™ plus 594 goat anti-rabbit IgG (H + L), respectively (Thermo Fisher; 1:300).

## Pharmacological perturbations of the actin cytoskeleton

Latrunculin A (Lat A, Millipore Sigma) was dissolved in 100% dimethyl sulfoxide (DMSO) and subsequently diluted in lamprey

ringer solution for a final concentration of 2.5 μM Lat A and 0.03% DMSO. Correspondingly, controls were 0.03% DMSO in lamprey Ringer solution. Dissected spinal cord were either (1) pretreated in either Lat A (2.5 μM and 0.03% DMSO) or no compound (0.03% DMSO) ringer solution for 30 min at room temperature (RT) followed by microinjection of phalloidin or (2) RS axons were microinjected with phalloidin followed by post-treatment in either Lat A (2.5 μM and 0.03% DMSO) or no compound (0.03% DMSO) ringer solution for 30 min. In both cases, the spinal cord tissue was fixed immediately in 4% paraformaldehyde in 0.1 M PBS, pH 7.4, no longer than 70 min after the beginning of the treatment, for >3 h at RT and then overnight at 4 °C. The fixed tissue was mounted on glass slides with No 1.5 glass coverslip and sealed using ProLong™ Gold Antifade Mountant (Thermo Fisher Scientific). Lamprey RS axons of interest were identified by the Alexa-Fluor™ 488 signal. Then, axons were imaged on a Leica STELLARIS DMi8 inverted confocal microscope (Plan-Apochromat 100x/1.4 objective, 2x digital zoom). Image z-stacks through the axon ($n = 14$–18) were collected from $N = 3$–4 animals from $n = 3$–4 axons per experimental condition.

## Cryo-electron tomography of primary murine neurons in culture

Primary rat hippocampal neurons were cultured on EM grids (R2/2 SiO2 film on Au mesh, Quantifoil) coated with poly-L-lysine. Cells (DIV 14–18) were incubated for ~5 min in a Tyrode solution containing 5% (v/v) glycerol prior to plunge-freezing. Cryo-focused ion beam (FIB) milling (Aquilos 2 Cryo-FIB, Thermo Fisher Scientific) was used to produce electron-transparent lamellae. Tomograms were collected with a 300 kV Krios G4 Cryo TEM microscope (Thermo Fisher Scientific) at the nominal pixel size of 2.94 Å. Tomograms were reconstructed using a weighted-back projection algorithm implemented in the IMOD software. Tomograms were deconvolved using a Wiener-like filter implemented in the TOM toolbox (Nickell et al, 2005). Tomograms were segmented using MemBrain-Seg (Lamm et al, 2024) and manually curated in Amira (Thermo Fisher Scientific). Actin filaments were traced using the XTracing module implemented in Amira (Rigort et al, 2012).

## Image processing, data analysis, and statistics

ImageJ software was used for creating representative movies and images. All image processing steps, such as cropping, brightness adjustments, merging channels, scale bars, time, and making montages from movies, were carried out using ImageJ software. Intensities for preparing representative line profiles were noted using ImageJ software. The data obtained from post-image analysis was further analyzed using the tools available in the GraphPad Prism software.

All experiments were independently reproduced by several experimenters, in several different laboratories of collaborating PIs on the study. No data were excluded by intention. Prior to statistical analysis, all data sets were tested for normality. Only data of a corresponding variance have been statistically compared.

### Actin enrichment with Syn1-FL phases

For determining actin enrichment within Syn1-FL phases, at first x-y shifts during the acquisition of the movie were corrected for

individual condensates using the StackRegJ plugin (Slaughter et al, 2013). Further, an ROI was created on condensates and their change in intensity over time was computed using the create spectrum jru v1 plugin (Slaughter et al, 2013).

### Actin network reconstructions

For creating actin network reconstructions, images from the actin channel were preprocessed by the difference of Gaussians (Sigmas used were 2.0 and 1.0) to improve fibril recognition. The resulting image was combined with the thresholded initial image to restore severed fibril-condensate connections. An iterative median filter (two iterations; radius 2.0 pixels) was applied to reduce the background salt-and-pepper noise and followed by thresholding to discard it. Finally, the image was skeletonized using the Skeletonize (2D/3D) plugin. The skeletonized image was traced (automatic tracing; discarding segments shorter than three pixels) and 3D-reconstructed using the SNTplugin (Arshadi et al, 2021).

### Sholl analysis

The SNT plugin was also used for performing Sholl analysis on polymerized actin networks from synapsin-1 phases. Images of actin-asters were semi-automatically traced using the plugin, and traces were used for Sholl analysis. A series of concentrical spheres (in 0.5-μm radius increments) with centers in synapsin condensates were created, and intersections of actin fibrils with these spheres were counted. An R code for linear mixed effect model (https://zenodo.org/records/1158612) was used for running statistical analysis of the Sholl data (Jackson et al, 2024; Wilson et al, 2017).

### Partitioning analysis

The Delta G analysis was conducted by adapting a previously published pipeline (Baggett DW et al, 2022). Briefly, condensates were segmented using the Otsu algorithm after Gaussian convolution, and the mean intensity was determined within and outside the condensates. The partition coefficient (Kp) was calculated as the ratio of mean intensity inside condensates to the mean intensity outside of condensates (see equation 1). $K_p$ was further used for computing Delta G (see equation 2).

$$K_p = \frac{mean\ intensity\ inside\ the\ condensate}{mean\ intensity\ outside\ the\ condensate} \tag{1}$$

$$\triangle G = -RT\ ln\left(K_p\right) \tag{2}$$

Here, R is the gas constant, its value is 1.9872 and is expressed in units of kcal/mol, and $T$ is temperature expressed in Kelvins (Baggett et al, 2022). Outliers were removed using the ROUT test. Normality was tested for all datasets. If normality was rejected, the Mann–Whitney test was used to assess significance. Otherwise, a two-sample $t$-test (equal variance not assumed) was used to assess significance.

### Quantification of phalloidin-positive clusters in lamprey RS axons

To understand the effects of Lat A may have on synaptic actin, a density analysis was conducted on RS axons where phalloidin-positive clusters were segmented, thresholded (>4000 intensity counts, >10 pixel area), and counted per axonal surface area. This

analysis was written and performed in MATLAB (MathWorks, R2022b). Data were graphed and statistical analysis conducted using GraphPad Prism Software.

### Quantification of two-color STED images

Analysis was performed using custom FIJI macros and Python scripts (see Data availability for a link to the GitHub repository). A FIJI macro was used that measures actin intensity inside synaptophysin-positive spots: The macro converted the synaptophysin (SV marker) channel into a mask using the built-in FIJI find_mask() function in the WT and SYN-TKO conditions. In the rescue conditions (rescue-FL, rescue-IDR) a shared mask of the GFP (synapsin/IDR) and the synaptophysin channel was created. Thereby, it was ensured that further analysis in the rescue conditions was only performed in GFP-expressing cells. The mask was then converted into ROIs using the FIJI built-in "Analyze particles" tool. The maximum intensity of the actin channel and the area is then measured for each ROI. A separate FIJI macro was used to measure the maximum intensity of each full actin image. A custom Python script was used to normalize the actin intensity per ROI to the actin intensity of the overall image and to extract the area of each ROI. The Python script took the outputs of the FIJI macros and normalized the maximum intensity per ROI of actin to the maximum intensity of the whole actin image. These results were used to generate medians of normalized actin intensity of synaptophysin-positive regions (ROIs) per image, per replicate, and per condition. Furthermore, the area of each ROI was extracted from the outputs of the FIJI macros. All conditions (WT, SYN-TKO, FL-rescue, and IDR-Rescue) were analyzed using the same pipeline. Graphs were generated using OriginPro. Outliers were removed using Grubb's test. Normality was tested for all datasets. If normality was rejected, the Mann–Whitney test was used to assess significance. Otherwise, a two-sample $t$-test (equal variance not assumed) was used to assess significance.

### Quantification of expanded synapses

Analysis was performed analogously to the STED images. In addition to the above described pipeline, a minimum intensity threshold was added for the VGLUT1 and PSD95 channels to avoid background inclusion during mask creation. Furthermore, generated ROIs were separated into smaller subregions by their circularity to avoid the merging of adjacent boutons. The detailed code for data processing is available as open source on GitHub public repository: https://github.com/jakobrentsch/actin_in_bouton. Expansion microscopy data and quantifications in primary hippocampal neurons are done in a blinded fashion.

## Cluster and principal component analysis of synapsin:actin assemblies

Analyses were performed in MATLAB (v. R2023a, The Math-Works). The connected components of each reconstructed actin network were identified as individual elements/nodes of the actin network. The largest nodes were split using the watershed algorithm based on the distance transform (Beucher 1992), and the smallest ones (<100 pixels) were removed. Ten properties were extracted for each node: (1) volume (μm³), (2) surface area (μm²), (3) diameter (μm), (4) main axis length (μm), (5) second axis length (μm), (6) third axis length (μm), (7) mean intensity, (8) maximum

intensity, (9) minimum intensity, and (10) median Euclidean distance from all other nodes ($\mu m^2$). The node properties from three datasets were concatenated (intensity values were rescaled for comparability). The similarity between these properties across nodes was assessed as Pearson's correlation. Parametric k-means cluster analysis was used to find a natural separation in the node-by-property matrix (log-transformed to account for scale differences and non-Gaussianity). The silhouette plot was used to determine the optimal number of clusters. A principal component (PC) analysis was applied to the node-by-property matrix, and PC1 and PC2 scores were plotted against each other to evaluate k-means results in a low-dimensional space (Bishop 2006).

## Data availability

All data generated or analyzed for this study are available within the paper and its associated supplementary information files. All other data presented are available upon reasonable request from the corresponding author. The detailed code for data processing of expansion microscopy data is available as open source on the GitHub public repository: https://github.com/jakobrentsch/actin_in_bouton. All codes used for this study are available upon reasonable request from the corresponding author.

The source data of this paper are collected in the following database record: biostudies:S-SCDT-10_1038-S44318-025-00516-y.

## Peer review information

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

## Acknowledgements

We thank the Advanced Medical Bioimaging Core Facility at Charité and the Central Microscopy Facility at MBL for the support. We thank Dr. Reinhard Jahn and his entire team (Max Planck Institute for Multidisciplinary Sciences, Göttingen, Germany) for the generous support to use the resources needed for the preparation of native synaptic vesicles; to Manfred Trimborn for technical and logistic support, to Anna Siegert for the assistance with cryo-electron tomography. The work is supported by the start-up funds from DZNE, the grants from the German Research Foundation (MI 2104 and SFB1286/B10) and the ERC Grant MemLessInterface (101078172) to DM; the grants from the National Institutes of Health NIA 2RF1 NS078165-12 to JRM and NINDS K99 NS126575-01 to CRV; the SFB1286/B02 to SK and SOR. CH is supported by a fellowship of the Innovative Minds Program of the German Dementia Association, and JVT by the German Academic Scholarship Foundation.

## Author contributions

**Akshita Chhabra**: Data curation; Formal analysis; Validation; Investigation; Visualization; Methodology; Writing—original draft; Writing—review and editing. **Christian Hoffmann**: Data curation; Formal analysis; Validation; Investigation; Visualization; Methodology; Writing—original draft; Writing—review and editing. **Gerard Aguilar Pérez**: Resources; Data curation; Formal analysis; Validation; Investigation; Visualization; Methodology; Writing—review and editing. **Aleksandr A Korobeinikov**: Formal analysis; Validation; Investigation; Visualization; Methodology; Writing—review and editing. **Jakob Rentsch**: Software; Formal analysis; Validation; Investigation; Visualization; Methodology; Writing—review and editing. **Nadja Hümpfer**: Data curation; Software; Formal analysis; Validation; Investigation; Visualization; Methodology; Writing—review and editing. **Linda Kokwaro**: Data curation; Formal analysis; Investigation; Methodology; Writing—review and editing. **Luka Gnidovec**: Data curation; Formal analysis; Investigation; Methodology; Writing—review and editing. **Arsen Petrović**: Data curation; Software; Formal analysis; Validation; Investigation; Visualization; Methodology; Writing—review and editing. **Jaqulin N Wallace**: Data curation; Software; Formal analysis; Validation; Investigation; Visualization; Methodology; Writing—review and editing. **Johannes Vincent Tromm**: Data curation; Formal analysis; Validation; Investigation; Visualization; Methodology; Writing—review and editing. **Cristina Román-Vendrell**: Data curation; Formal analysis; Investigation; Methodology; Writing—review and editing. **Emma C Johnson**: Data curation; Formal analysis; Investigation; Methodology; Writing—review and editing. **Branislava Ranković**: Data curation; Formal analysis; Investigation; Methodology; Writing—review and editing. **Eleonora Perego**: Data curation; Formal analysis; Investigation; Methodology; Writing—review and editing. **Tommaso Volpi**: Data curation; Software; Formal analysis; Validation; Investigation; Visualization; Methodology; Writing—review and editing. **Rubén Fernández-Busnadiego**: Resources; Software; Supervision; Investigation; Methodology; Writing—review and editing. **Sarah Köster**: Resources; Supervision; Investigation; Methodology; Writing—review and editing. **Silvio O Rizzoli**: Resources; Supervision; Investigation; Methodology; Writing—review and editing. **Helge Ewers**: Resources; Software; Supervision; Investigation; Methodology; Writing—review and editing. **Jennifer R Morgan**: Resources; Data curation; Formal analysis; Supervision; Funding acquisition; Validation; Investigation; Visualization; Methodology; Writing—review and editing. **Dragomir Milovanović**: Conceptualization; Resources; Data curation; Software; Formal analysis; Supervision; Funding acquisition; Validation; Investigation; Visualization; Methodology; Writing—original draft; Project administration; Writing—review and editing.

Source data underlying figure panels in this paper may have individual authorship assigned. Where available, figure panel/source data authorship is listed in the following database record: biostudies:S-SCDT-10_1038-S44318-025-00516-y.

## Funding

## Disclosure and competing interests statement

The authors declare no competing interests.

# Expanded View Figures

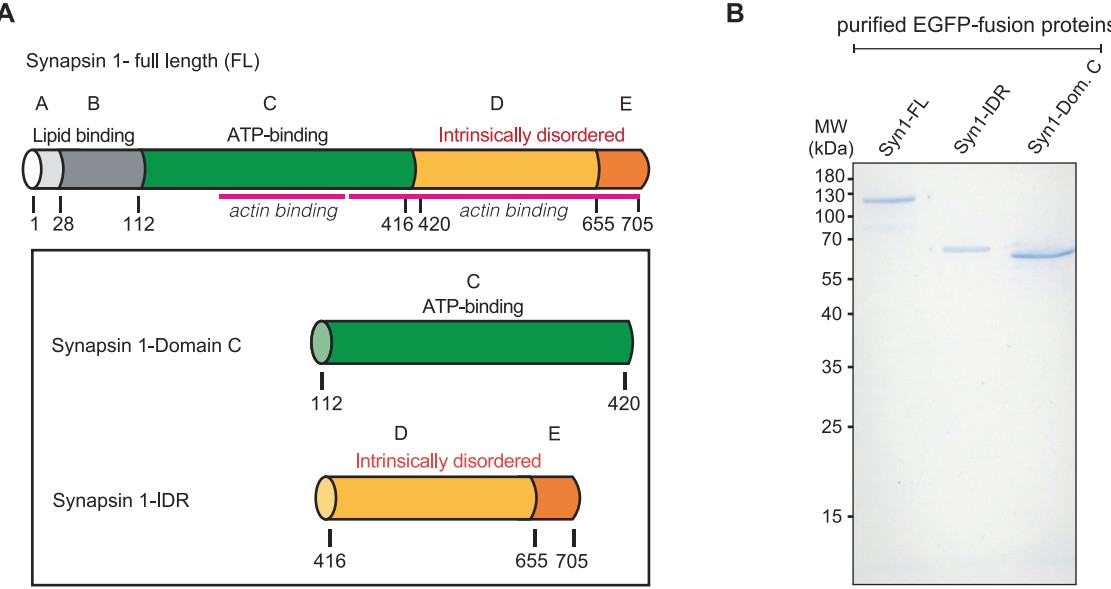

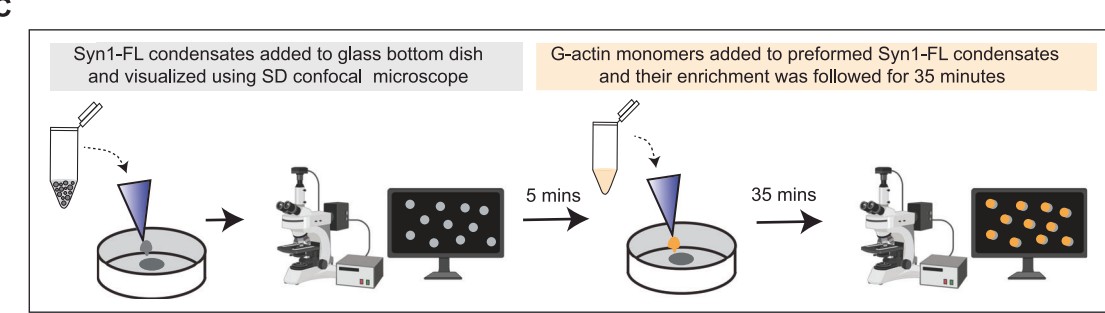

**Figure EV1.  Recombinant EGFP-synapsin-1 and domains used for in vitro reconstitutions.**

(A) Schematic cartoon representing the domain organization of synapsin-1 (Syn1). Syn1 full length (FL), Domain C (Dom. C), and intrinsically disordered region (IDR) are depicted. Actin-binding regions are depicted with magenta (a.a. 222–370 and 371–705). (B) SDS-PAGE gel of the purified proteins employed for in vitro reconstitutions in this study: EGFP-Syn1-FL (102.235 kDa), EGFP-Syn1-IDR (57.621 kDa), and EGFP-Syn1-Dom. C (63.116 kDa). (C) Schematic illustration of the Syn1-actin reconstitution assay flow. Actin polymerization from Syn1 phases was examined by first preforming 6 μM EGFP-Syn1-FL condensates with 3% (w/v) PEG 8000 on a glass-bottom dish. After incubating for 5 min, when EGFP-Syn1-FL condensates became 3–4 μm in size, ATTO647-labeled G-actin monomers were added from the top into these preformed EGFP-Syn1-FL condensates such that the final concentration of actin and EGFP-Syn1-FL in the final reaction mix was 4 μM for both components. Source data are available online for this figure.

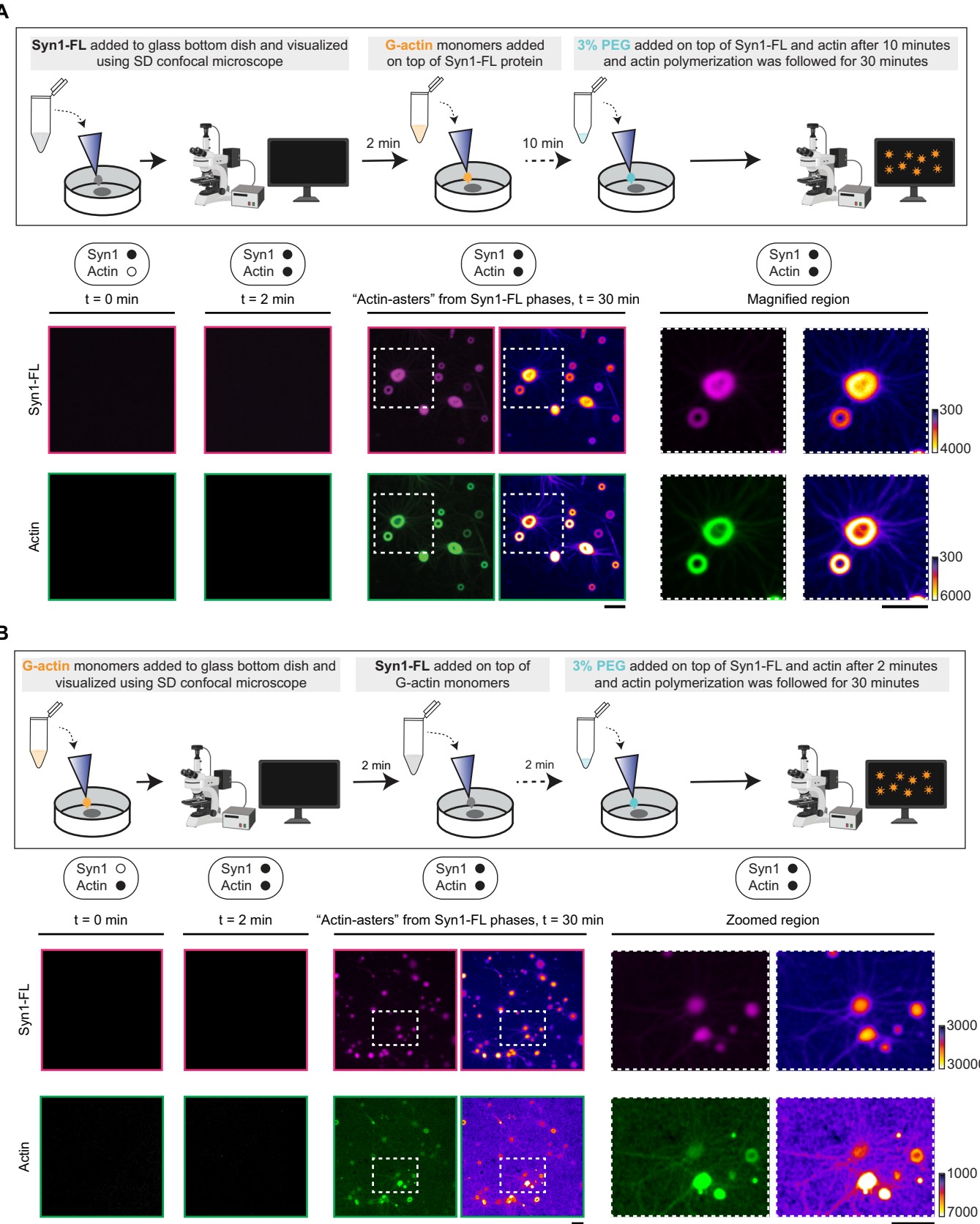

◀

**Figure EV2.  Reconstitution of actin with synapsin-1 in two distinct reaction orders.**

(A) Top: Schematic illustration showing the order-1 of the reconstitution assay. Actin polymerization from synapsin-1 phases was examined by first adding 4 μM synapsin-1 to a glass-bottom dish, followed by the addition of 4 μM ATTO647 G-actin monomers 2 min later. Subsequently, 3% (w/v) PEG 8000 was added on top of the reaction mix after 10 min, and actin polymerization was followed for 30 min. Bottom: Representative SD confocal microscopy images from the reconstitution of actin with synapsin 1 liquid phases in reaction buffer at $t = 0, 2$, and 30 min. Zoomed-in regions towards the right side show actin-asters. Images were acquired at 488 and 647 nm wavelengths for EGFP-Syn1-FL and actin, respectively. Scale bar, 5 μm. (B) Top: Schematic illustration showing the order-2 of the reconstitution assay. Actin polymerization from synapsin-1 phases was assessed by first adding 4 μM ATTO647 G-actin monomers to a glass-bottom dish. Two minutes later, 4 μM synapsin 1 was added on top of the reaction mix. Subsequently, 3% (w/v) PEG 8000 was added on top of the reaction mix after 2 min, and actin polymerization was followed for 30 min. Bottom: Representative confocal microscopy images from the reconstitution of actin with synapsin-1 liquid phases in reaction buffer at $t = 0, 2$, and 30 min. Zoomed-in regions towards the right side show actin-asters. Images were acquired at 488 and 647 nm wavelengths for EGFP-Syn1-FL and actin, respectively. Scale bar, 5 μm. Source data are available online for this figure.

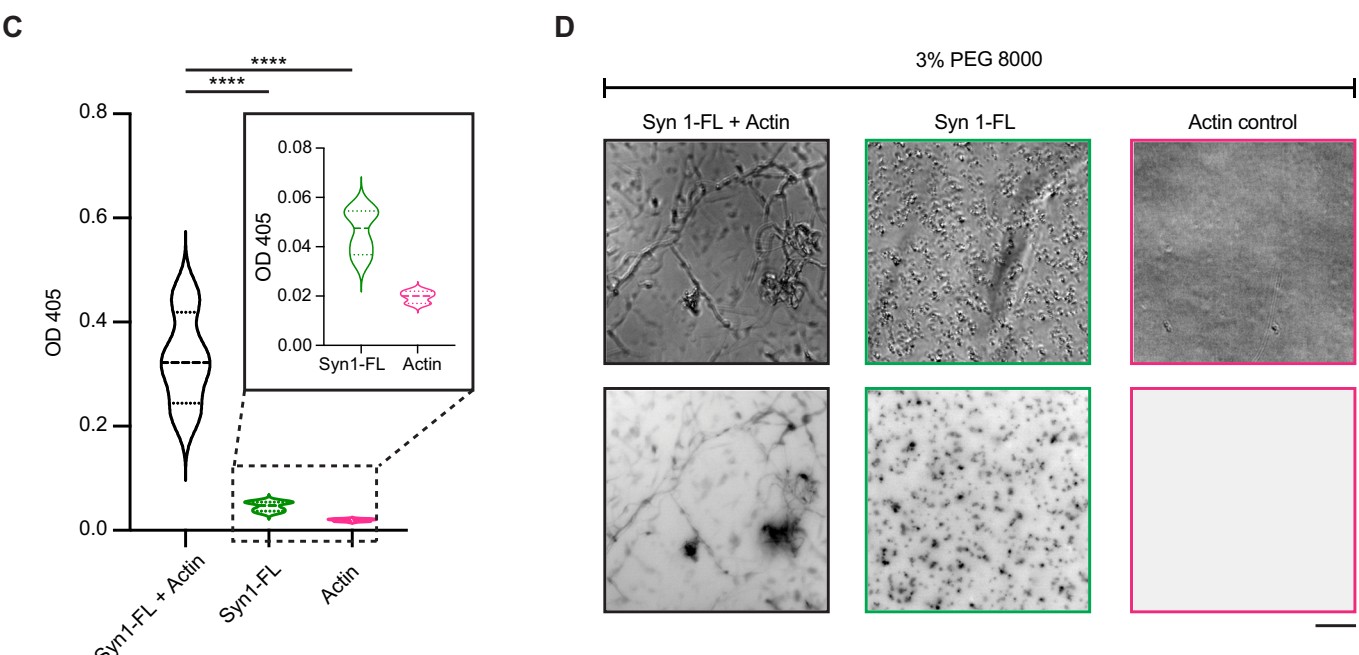

**A**

| | 0 min | 35 min |

Syn1-Dom. C

Actin

Merge

Syn1-Dom. C - no condensates

Actin is not enriched; no polymerization

**B**

| | 0 min | 35 min |

G3BP2

Actin

Merge

G3BP2 forms condensates

Actin enriches within condensates but does not polymerize

**C**

**D**

3% PEG 8000

Syn 1-FL + Actin    Syn 1-FL    Actin control

◀ **Figure EV3.** **In vitro reconstitutions and turbidity measurements of Syn1 variants with actin.**

(A) Representative confocal images from the in vitro reconstitution of EGFP-Syn1-Dom. C (4 μM, 3% (w/v) PEG 8000) with ATTO647-labeled G-actin monomers (4 μM) at $t = 0$ (left) and $t = 35$ min (right). Image acquisition for EGFP-Syn1-Dom. C and ATTO647 G-actin was carried out at excitation wavelengths 488 and 647 nm, respectively. Scale bar, 5 μm. (B) Representative confocal images from the in vitro reconstitution of EGFP-G3BP2 (4 μM, 3% PEG 8000) with ATTO647-labeled G-actin monomers (4 μM) at $t = 0$ (left) and $t = 35$ min (right). Images were acquired at excitation wavelengths 488 and 647 nm for EGFP-G3BP2 and ATTO647 G-actin, respectively. Scale bar, 5 μm. (C) Quantification of the turbidity assay. Plot comparing the turbidity measurements for EGFP-Syn1-FL with actin, EGFP-Syn1-FL alone, and actin alone in the presence of 3% (w/v) PEG 8000. Actin polymerization was assessed as an increase in optical density after a 48 h incubation period. Actin used for the assay was $Mg^{2+}$-exchanged and supplemented with 0.5 mM ATP. Data shown here is quantified from four independent experiments ($N = 4$). The $p$ values are: $8.53 \times 10^{-4}$ for Syn1-FL+Actin Vs Syn1-FL and $1.13 \times 10^{-4}$ for Syn1-FL+Actin Vs Syn1-FL Vs Actin; ****$p < 0.0001$; one-way ANOVA test. (D) Top: representative brightfield images of reaction mixes after turbidity assay from (A). Bottom: epifluorescence images of the same regions at 488 nm excitation wavelength. Scale bar, 50 μm. Source data are available online for this figure.

**A**

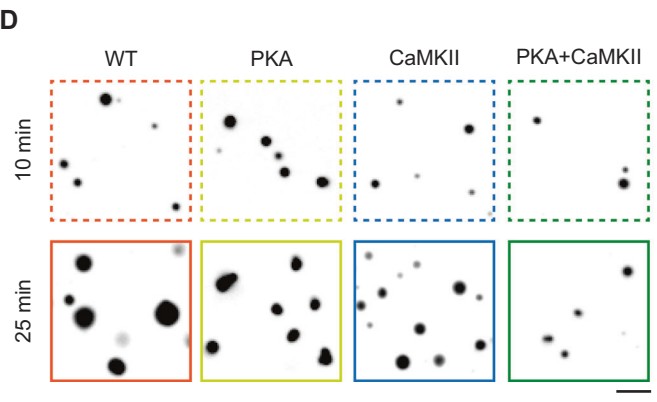

Transfected Expi293™ cells → Lyse cells → 1 Lysate → Clarify by centrifugation → 2 Supernatant → Filter supernatant Load into HisTrap column → 3 Flow-through → Wash with 40 mM Imidazole → 4 Wash → Elute with 400 mM Imidazole → 5 Elution/Pre-SEC → Load into S200 column Collect fractions → 6 Post-SEC/Pre-digestion → Digest fractions with SENP1 (SUMO protease) → 7 Post-digestion/Pre-ReHis → Run through Ni-NTA beads → 8 Post-ReHis → Dialyze against SEC buffer (150 mM NaCl, 25 mM Tris-HCl pH 7.4, 0.5 mM TCEP) → Purified protein

**B**

His-SUMO-EGFP-Synapsin-1 purification steps

Lanes: M, 1, 2, 3, 4, 5, 6, 7, 8

kDa: 180, 130, 100, 70, 55, 40, 35, 25, 15, 10

**C**

purified EGFP-fusion proteins

Syn1-WT, Syn1-PKA (S9E), Syn1-CaMKII (S568/605E), Syn1-PKA+CaMKII (S9/568/605E)

kDa: 180, 130, 100, 70, 55, 40, 35, 25, 15, 10

**D**

WT, PKA, CaMKII, PKA+CaMKII

10 min

25 min

**E**

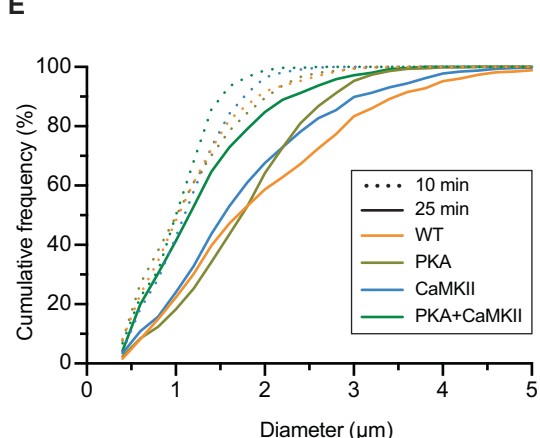

Cumulative frequency (%) vs Diameter (μm)

···· 10 min
— 25 min
WT
PKA
CaMKII
PKA+CaMKII

◀ **Figure EV4.** **Recombinant synapsin 1 phosphomimetics used for in vitro reconstitutions.**

(A) Schematic cartoon depicting the purification steps followed to obtain synapsin 1 (Syn1) from transfected Expi293™ cells expressing His-SUMO-EGFP-Syn1. This same pipeline is used for all Syn1 variants. (B) Exemplary SDS-PAGE gel for His-SUMO-EGFP-Syn1-CaMKII (S568/605E) construct, showing the fractions from each purification step from 1 to 8 in (A). (C) SDS-PAGE gel of the final purified EGFP-Syn1 versions employed in vitro reconstitutions in this study: EGFP-Syn1 WT, EGFP-Syn1-PKA (S9E), EGFP-Syn1-CaMKII (S568/605E), and EGFP-Syn1-PKA+CaMKII (S9/568/605E). All proteins weigh 102.235 kDa. (D) Representative images of EGFP-Syn1 WT, EGFP-Syn1-PKA (S9E), EGFP-Syn1-CaMKII (S568/605E), and EGFP-Syn1-PKA+CaMKII (S9/568/605E) when reconstituted in SEC buffer and in the presence of 3% PEG 8000 at $t = 10$ and $t = 25$ min. Images were acquired using a spinning-disk confocal microscope, employing the 488 nm wavelength for EGFP-Syn1. Scale bar, 5 μm. (E) Cumulative frequency indicating the size distribution of at $t = 0$ (full line) and $t = 45$ min (dotted line); color code as in (D). Data from three independent reconstitutions for each condition. Source data are available online for this figure.

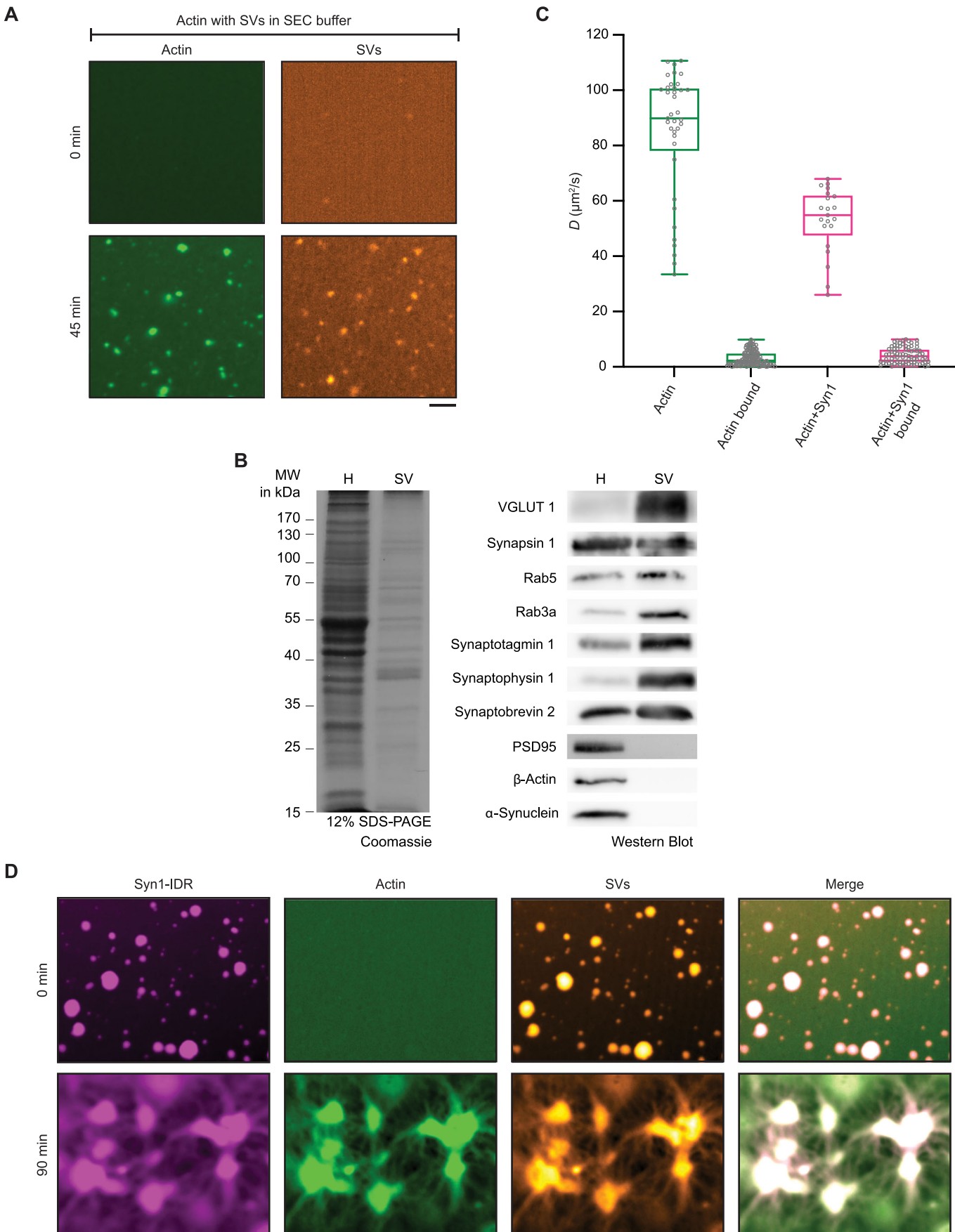

◀ **Figure EV5.  Reconstitution of actin with synapsin-1 and native SVs.**

(A) Representative images for actin reconstitution in the presence of natively purified 3 nM SVs labeled with FM4-64 (1.65 µM final concentration), 3% PEG 8000 in reaction buffer at $t = 0$ and $t = 45$ min. Images were acquired using a spinning-disk confocal microscope at 561 and 647 nm for SVs and actin, respectively. Scale bar, 5 µm. (B) Quality control of synaptic vesicles. Left: Coomassie-stained 12% SDS-PAGE of fractions, brain homogenate (H) and final synaptic vesicle fraction (SV), from the synaptic vesicle isolation procedure from native rat brain (5 µg per lane). Right: Western blot of brain homogenate and final SV fraction detecting classical protein markers of SV enrichment and absence of proteins regarded as contaminants. (C) Diffusion coefficients of actin bound to SVs immobilized on a functional surface as described in Perego et al, 2020. Data from three independent replicates ($N = 3$). The box stretches from the 25th to the 75th percentile, the dots represent individual data points, the central line shows the median, and the whiskers represent the min and max values. Green boxes, diffusion of actin alone; magenta boxes, diffusion of actin in the presence of EGFP-synapsin 1. (D) Reconstitution of actin with synapsin-1 IDR-SV phases and actin-asters. Representative confocal images of the reconstituted EGFP-Syn1-IDR (4 µM, 3% PEG 8000) and SVs (3 nM, labeled with FM4-64, 1.65 µM) condensates after adding ATTO647-labeled G-actin monomers (4 µM) at $t = 0$ (top) and 90 min (bottom). Excitation at 488 nm for EGFP-synapsin 1-IDR, 560 nm for SVs labeled with FM4-64 and 647 nm for ATTO647 G-actin. Scale bar, 5 µm. Source data are available online for this figure.

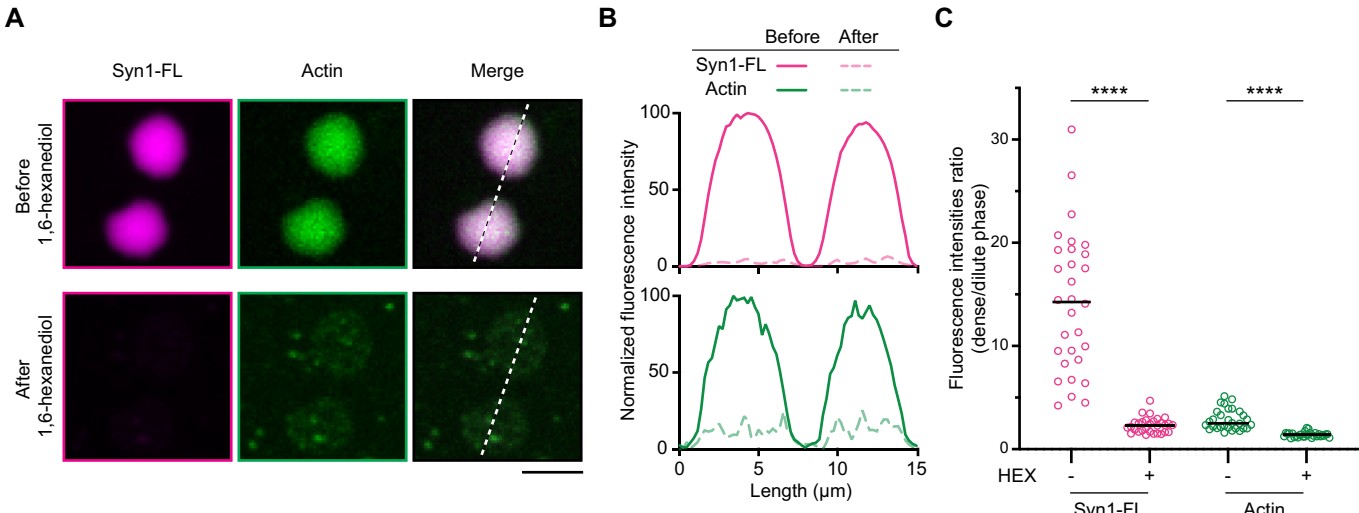

**Figure EV6. 1,6-Hexanediol disperses EGFP-synapsin 1:actin condensates lacking an apparent actin polymerization.**

(A) Representative confocal images from the in vitro reconstituted EGFP-synapsin-1:actin:GUV assemblies before and after 1,6-hexanediol treatment. The dashed line represents the line used for plotting the intensity profile. Scale bar, 5 μm. (B) Fluorescence intensity profiles of EGFP-synapsin-1 and actin along the dashed line from (A). Solid lines, fluorescence intensity before, and dashed lines, after treatment with 1,6-hexanediol. (C) Quantification of synapsin-1 and actin partitioning in EGFP-synapsin-1: actin:GUV assemblies before and after 1,6-hexanediol treatment. Data from three independent reconstitutions, 30 condensates analyzed for each condition. The $p$ values are $6.76 \times 10^{-17}$ for Syn1-FL before vs. after treatment and $6.31 \times 10^{-15}$ for actin before vs after treatment. ****$p < 0.0001$; Mann–Whitney $U$-test (two-tailed, non-parametric). Source data are available online for this figure.

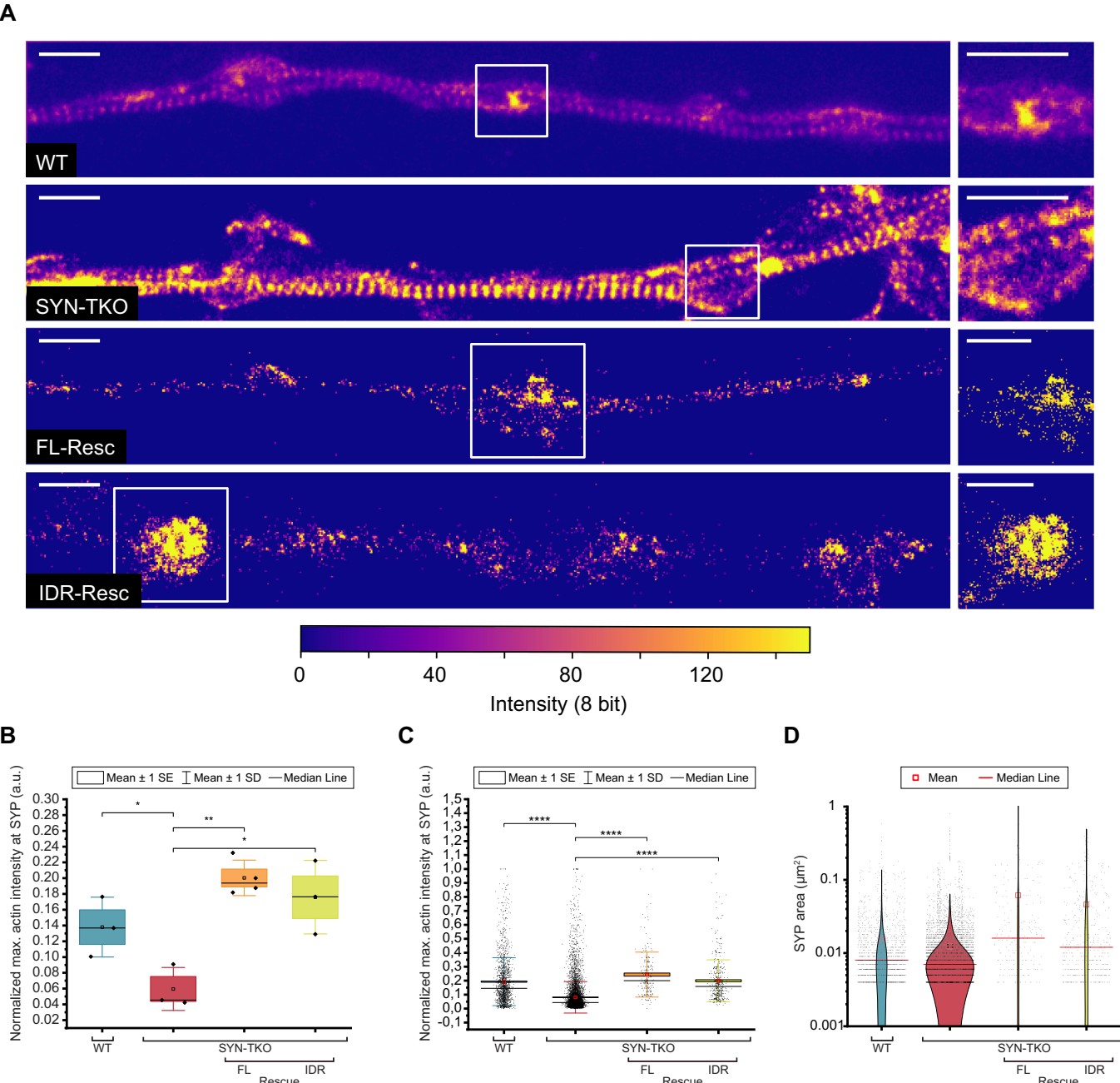

**Figure EV7. Synaptic vesicle condensates are necessary for concentrating actin at the presynaptic boutons.**

(A) Primary hippocampal neurons in culture (DIV 14) stained for actin (phalloidin-AberriorStarRed); images as in Fig. 8, but here shown with the same intensity scaling. Scale bars, 1 μm. (B) Actin enrichment in synaptic vesicle cohorts is defined as a normalized intensity signal of actin channel within regions positive for synaptophysin (wild-type and synapsin triple knockout) or both synaptophysin and EGFP (in rescue experiments). Data from at least three independent neuronal preparations (for WT: $N = 3$; $n = 3$; for SynTKO: $N = 3$; $n = 3$; for SynTKO+FL: $N = 4$; $n = 4$; for SynTKO+IDR: $N = 3$; $n = 3$); data points represent median maximum actin intensity per neuronal preparation; the box represents mean ± 1.0 SE, whiskers represent mean ± 1.0 SD, and the central line represent median two sample $t$-test (equal variance not assumed). WT-SynTKO $p = 0.0049$, *; SynTKO-SynTKO+FL $p = 0.0021$, **; SynTKO-SynTKO+IDR $p = 0.030$, *. (C) Actin enrichment in synaptic vesicle cohorts is defined as a normalized intensity signal of actin channel within regions positive for synaptophysin (wild-type and synapsin triple knockout) or both synaptophysin and EGFP (in rescue experiments). Data from three independent neuronal preparations (for WT: $N = 3$; $n = 1592$; for SynTKO: $N = 3$; $n = 7096$; for SynTKO+FL: $N = 4$; $n = 209$; for SynTKO +IDR: $N = 3$; $n = 460$); data points represent maximum actin intensity per region positive for synaptophysin. The box represents mean ± 1.0 SE, and whiskers represent mean ± 1.0 SD, and the central line represents median. Mann–Whitney test; WT-SynTKO, $p = 3.06 \times 10^{-251}$; SynTKO-SynTKO+FL, $p = 2.12 \times 10^{-79}$; SynTKO-SynTKO +IDR, $p = 1.96 \times 10^{-135}$; ****$p < 0.0001$. (D) Area of synaptophysin-positive regions in different conditions. Note in SynTKO neurons the presence of regions of varying size, particularly the smaller ones, presumably a consequence of dispersed SVs into smaller cohorts and/or individual vesicles. The red square represents mean values, and the central line shows the median. Data from at least three independent replicates (for WT: $N = 3$; $n = 1592$; for SynTKO: $N = 3$; $n = 7096$; for SynTKO+FL: $N = 4$; $n = 209$; for SynTKO+IDR: $N = 3$; $n = 460$). Source data are available online for this figure.

