## [Peer Review File · The EMBO Journal]

Condensates of synaptic vesicles and synapsin-1 mediate actin polymerization

Akshita Chhabra, Christian Hoffmann, Gerard Aguilar Pérez, Aleksandr Korobeinikov, Jakob Rentsch, Nadja Hümpfer, Linda Kokwaro, Luka Gnidovec, Arsen Petrovic, Jaquin Wallace, Johannes Tromm, Cristina Román-Vendrell, Emma Johnson, Branislava Rankovic, Eleonora Perego, Tommaso Volpi, Rubén Fernández-Busnadiego, Sarah Köster, Silvio Rizzoli, Helge Ewers, Jennifer Morgan, and Dragomir Milovanovic

Corresponding author(s): Dragomir Milovanovic (Dragomir.Milovanovic@dzne.de)

Review Timeline:

Submission Date:	27th May 24
Editorial Decision:	3rd Jul 24
Appeal Received:	1st Apr 25
Editorial Decision:	11th Jun 25
Revision Received:	3rd Jul 25
Accepted:	8th Jul 25

Editor: Kelly M Anderson / Ioannis Papaioannou

Transaction Report:

Dear Dr. Milovanovic,

Thank you for submitting your manuscript for consideration by the EMBO Journal. It has now been seen by three referees whose comments are shown below. Given their opinions and the fact that the EMBO Journal can only afford to accept papers which receive enthusiastic support from a majority of referees, I am afraid we cannot offer to publish it here.

Thank you in any case for the opportunity to consider this manuscript. I am sorry we cannot be more positive on this occasion, but we hope nevertheless that you will find our referees' comments helpful.

Yours sincerely,

Kelly M Anderson, PhD
Editor, The EMBO Journal
k.anderson@embojournal.org

Referee #1:

This paper addresses an important question, namely the role of synapsins in organizing vesicle clusters. However there are very few data in this paper and critical issues are not addressed.

1. What form of Synapsin 1 is examined here and do other synapsin isoforms (other splice variants of synapsin 1 and the similarly abundant synapsin 2) perform similar biochemical activities?
2. What exactly is the concept of the relation of actin and synapsin in the paper? It is well established that there is no actin in the vesicle cluster. Is the idea that actin links vesicles clusters together? That seems nonsensical since vesicle clusters are rarely close to each other. De Camilli has shown nicely that actin is involved in endocytosis perisynaptically not in the vesicle cluster. Is that what the authors propose, namely that synapsin 1 is involved in endocytosis? what is the evidence?
3. There is no evidence that any of the paper's observations are physiologically relevant. Figure 5 looks at the synapsin mutant mice but the phenotype is miniscule (10% change) only significant because pseudoreplicates not real replicates are used for statistics.

The role of phase separation in vesicle cluster organization is really important question but this paper is far too premature in addressing this question.

Referee #2:

In "CONDENSATES OF SYNAPTIC VESICLES AND SYNAPSIN ARE MOLECULAR BEACONS FOR ACTIN SEQUESTERING AND POLYMERIZATION", Akshita et al build upon their previous results that synapsin undergoes a phase separation to sequester synaptic vesicles, to demonstrate that condensation and actin binding properties of synapsin result in actin polymerization that forms actin arrays linking synaptic vesicles in pre-synaptic boutons. These results are supported by both in vitro and in vivo reconstitution assays. These assays also demonstrate that the IDR domain of synapsin is necessary for actin bundling.

For the most part, the data is clearly presented, and experiments are performed with the appropriate controls. I have only a few minor concerns/questions of clarifications that would help to strengthen the current submission.

The authors nicely demonstrate that the stress granule protein G3BP2 which can undergo phase separation and create an actin meshwork, lacks actin binding properties and therefore cannot polymerize actin like synapsin. However, it is unclear to me, where the actin binding sites of synapsin are located. From what I can find, it appears that the actin binding domain is primary located in the C domain, not the IDR. Therefore, it is a bit unclear to me why the IDR is necessary and sufficient to recruit and promote the polymerization of actin. Could the authors please indicate where the actin binding sites are in synapsin and indicate these in the diagram of synapsin in S. Fig. 1. Could the authors also comment in the discussion about why the IDR alone leads to linear rather than aster-like actin formations.

In Fig. 5, actin appears to be more clustered with the synapsin FL or IDR rescues than in the WT. Both rescues also appear to disrupt the regularly spaced cortical actin rings of the axon. Based on these results, could the authors comment on synapsin's role in regulating the reserve and readily releasable pools of synaptic vesicles as well as potential regulation of actin exchange between SV-associated asters and cortical actin rings?

Referee #3:

The manuscript by Chhabra, Hoffmann et al. investigates the specific role of synapsins in the organisation of synaptic vesicles (SVs) in neurons by supporting the architecture of actin asters around these SVs, as well as their intercommunication. The authors discuss the relevance of these structures for neuronal communication based on SV clusters that assemble by liquid-liquid phase separation (LLPS) thanks to synapsins. The authors have previously investigated the role of synapsin-1 in the organisation of SV clusters by analysing condensates with liquid-like properties and in cellular studies of motility, and found that the full-length and short intrinsically disordered fragment of synapsin 1 restored the SV motility pattern in synapsin triple knock-out animals. Indeed, casein kinase II appears to be a key kinase in regulating the formation of aggregates with SVs, suggesting a regulated disassembly similar to that observed in synaptic activity. Synapsins are known to bind actin, and here the reorganisation of these three elements was monitored by *in vitro* reconstitution approaches and super-resolution imaging in lamprey and mouse neurons. The data presented show that sites of LLPS of synapsin correspond to SVs and actin polymerisation foci. The organisation of SV clusters in neurons appears to be synapsin dependent, as the triple knock-out mouse (all three synapsins) shows poorly organised clusters. This is rescued by reconstitution with full length and short intrinsically disordered fragment of synapsin 1.

The manuscript is well written and the experiments are generally well described and documented. There are some relevant questions that are required at this stage to support the statements in the main text and the claims of authors:

Do the authors think that the use of proteins and models from different species (rabbit actin, human synapsin, mouse and lamprey experimental models) might influence their results and therefore make this study less general than described in principle in the text? Are all the proteins involved conserved between species and regulated by similar processes/proteins?

Are the actin asters resistant to actin-targeted drugs, such as latrunculin A and cytochalasin B or D, or to myosin-targeted drugs? SVs also appear to form asters in the images provided (e.g. Fig. 3; Supplementary Fig. 6). Can the authors ensure that there are no other elements in isolated SVs that could interfere or regulate the reaction? The composition of the SV should be shown as an additional control.

Is the order of addition of the elements important for the formation of aggregates? Within the cell, all the elements will be together; as actin can bind to SVs in the absence of synapsin-1, have the authors tried adding the elements in different order and at different times of the reaction?

Although imaging of the lamprey axon shows co-localisation of actin assemblies, synapsin and SV, these do not appear to form asters (Fig. 3f does not demonstrate aster presence). Figure 5 requires quantification. The MFI of insets in

Fig 5a-d is different in single channels (grey images) and the merge images (magenta/green). Synaptophysin aggregates of smaller size seem to co-localize with actin structures, even if synapsins are absent. This is in contrast with the claim of authors. Are microtubules or other cytoskeletal components relevant here?

Figure 5: Do these STED images correspond to single planes? Where is located the post-synaptic density in these pre-synaptic boutons? Authors should visualize the complete structure here to help differentiate single neurons (there can be also auto-synapses)? The asters are not observed there, either in wt or Ko mice; main structure of actin seems to be conserved, as pointed-out by authors (Fig. 5i). Is there other cytoskeletal element involved? Molecular motors helping the movement of the SVs?

The triple knock-out mouse model is reconstituted with one of the synapsins. Are the other members of the family involved in the organisation and regulation of the LLPS described so that these experiments lack the complexity required for the *in vivo* process? The authors should perform some *in vitro* reconstitution with all the synapsins.

Minor:

Minor:

Fig Panels Fig. 5e, f, g, i j should be properly quoted in the text under Results, page 10.

Suppl. Fig. 8 describes the recognition of synapsin-1 in lampreys by the antibody used; please, make it clear in the main text.

** As a service to authors, EMBO Press provides authors with the possibility to transfer a manuscript that one journal cannot offer to publish to another EMBO publication or the open access journal Life Science Alliance launched in partnership between EMBO Press, Rockefeller University Press and Cold Spring Harbor Laboratory Press. The full manuscript and if applicable, reviewers' reports, are automatically sent to the receiving journal to allow for fast handling and a prompt decision on your manuscript. For more details of this service, and to transfer your manuscript please click on Link Not Available. **

POINT-BY-POINT RESPONSE TO:**CONDENSATES OF SYNAPTIC VESICLES AND SYNAPSNIN ARE MOLECULAR BEACONS FOR ACTIN SEQUESTERING AND POLYMERIZATION**

We thank all three Referees for the insightful and constructive comments. We have done a comprehensive revision of the manuscript to address the issues they raised. Specifically, we:

1. cloned, expressed and purified WT and 3 phosphomimetics of synapsin-1 (PKA, CaMKII, and double PKA-CaMKII phosphomimetic). We see that actin polymerization from synapsin-1 condensates is activity/phosphorylation dependent, with CaMKII having a dominant role.

These data are now included in the form of two new figures, Figure 3 and EV Fig. 6.

2. determined the effects of latrunculin A on synapsin-driven actin polymerization both in vitro and in living lamprey synapse.

These comprehensive data are now included as the new Figure 5.

3. employed network analyses and provided a quantitative assessment of network properties for synapsin/actin assemblies.

This is included as new panels in Figure 6 D-E.

4. performed cryo-EM tomography of murine synapses, and visualized pre-synaptic actin filaments associated with both the SV cluster and at the periaxial zone.

These EM images are included as the new Figure 7 A-E.

5. performed expansion microscopy experiments on wild-type and synapsin triple knock-out neurons. The resulting quantitative assessment led to the finding that deletion of synapsins specifically leads to the loss of pre-synaptic actin, while post-synaptic actin remains unaltered.

This new data are included in the new Figure 7 F-I.

Several additional experimental requests of the referees were addressed - i.e., *new Figures EV3, EV7, EV8, EV12.*

Specific point-by-point responses are included below.

Referee #1:

This paper addresses an important question, namely the role of synapsins in organizing vesicle clusters. However there are very few data in this paper and critical issues are not addressed.

We thank the Referee for pointing out to the importance of the question, and for highlighting the critical points of the manuscript, which needed improvement. Inspired by these comments, we performed new experiments, which confirmed our initial hypotheses and strengthened all of our claims. We also performed a major rewriting of the manuscript, and amended the figures according to the suggestions of the Referee. We hope that our new submission clarifies all of the issues raised, and that the Referee agrees that our manuscript has improved extensively.

1. What form of Synapsin 1 is examined here and do other synapsin isoforms (other splice variants of synapsin 1 and the similarly abundant synapsin 2) perform similar biochemical activities?

We have focused on synapsin 1a, the most highly expressed variant (Südhof *et al*, 1989). Our biochemical analysis of the phase separation properties of synapsin 2 suggested a similar phase behavior (Milovanovic *et al*, 2018). Similarly, synapsins 1 and 2 share the similar sequence characteristics and biophysical properties, such as ‘charge blockiness’, or the enrichment with proline and polar residues (Hoffmann *et al*, 2025).

Importantly, the basic phase biochemical activities of synapsin 2 have been reported in our prior studies. We now emphasize this throughout the revised manuscript with new text (**lines 417-429**).

Inspired by this comment, in conjunction with the comments of Referee #2, we decided to characterize other variants of synapsin 1 as well, specifically several key phosphomimetics of synapsin 1a. To achieve this, we purified synapsin 1 phosphorylation mutants that mimic synaptic activity and affect synapsin 1-actin assemblies. We cloned PKA (Ser 9), CaMKII (Ser-568 and Ser-605), and double PKA/CaMKII (Ser 9, Ser-568, and Ser-605) by mutating the target serines to glutamate. We were able to observe that synapsin 1 phosphomimetics had on synapsin 1-actin assemblies. Sholl analyses of actin networks formed by these condensates of synapsin 1 indicate that PKA significantly reduces the connectivity and network properties of synapsin 1-actin assemblies. On the other hand, the CaMKII phosphomimetic, either alone or in combination with the PKA phosphomimetic, led to a dramatic decrease in actin polymerization, practically abolishing the ability of synapsin to trigger actin polymerization. These new results are included **in the main text (lines 205-230)** and as a **new main Figure 3 and new EV6**.

2. What exactly is the concept of the relation of actin and synapsin in the paper? It is well established that there is no actin in the vesicle cluster. Is the idea that actin links vesicles clusters together? That seems nonsensical since vesicle clusters are rarely close to each other. De Camilli has shown nicely that actin is involved in endocytosis perisynaptically not in the vesicle cluster. Is that what the authors propose, namely that synapsin 1 is involved in endocytosis? what is the evidence?

It is well established that actin and synapsin functionally cooperate during vesicle trafficking at synapses (Bähler and Greengard, 1987; Hirokawa *et al*, 1989; Benfentati *et al*, 1992; Shupliakov *et al*, 2002; Bloom *et al*, 2003, Sankaranarayanan *et al*, 2003; Fernandez-Busnadiago *et al*, 2010). However, the underlying mechanisms that coordinate the synapsin-actin networks are unknown. Our study demonstrates a novel finding, namely that synapsin condensates seed actin polymerization, forming structures *in vitro* that recapitulate the actin-synapsin networks in living vertebrate synapses. We have emphasized this concept in the revised Abstract, Introduction and Discussion, supported by extensive *in vitro* and *in vivo* data.

We agree with the Referee that actin is involved in endocytosis in the periaxial zone of synapses, and indeed one of the co-authors (Morgan) was involved in this discovery when she was a postdoc in the De Camilli lab. Our findings showing that actin eventually forms toroids around synapsin condensates, which is in agreement with the published literature.

However, a second pool of actin is also found within the vesicle clusters. Several studies over the years suggested the presence of actin both at SV cluster and at the periaxial zone. Early electron microscopy studies (Hirokawa *et al*, 1989), EM tomography (Fernández-Busnadiago *et al*, 2010) and super-resolution imaging all indicate the presence of substantial levels of polymerized actin in the SV cluster (Wilhelm *et al.*, 2014 and Bingham *et al*, 2023). To validate these existing claims, we performed cryo-EM tomography of hippocampal neurons in culture.

Indeed, we could observe actin filaments associated with SV clusters, as well as the filaments in the periaxial zone. These results are now included as a **new main Figure 7 A-E**.

3. There is no evidence that any of the paper's observations are physiologically relevant. Figure 5 looks at the synapsin mutant mice but the phenotype is miniscule (10% change) only significant because pseudoreplicates not real replicates are used for statistics.

We have performed two experimental lines to dissect the actin/SV organization in living synapses upon (i) acute perturbations (experiments in lamprey) and (ii) chronic perturbations (experiments in synapsin triple knockout mice). In lamprey experiments (Figure 4 C-F and new Figure 5 E-I), we visualize actin toruses at the interface of SV condensates, which can be locally stabilized with phalloidin or disrupted by upon sequestering with latrunculin A. These data were collected from N=3-4 biological replicates, as reported in the Methods and figure legends.

In synapsin triple knock-out mice, we performed expansion microscopy (**new Figure 7 F-I**) and super-resolution STED imaging (**Figure 8 - former Figure 5**). We analyzed neurons from three independent biological replicates (i.e., each individual replicate comes from a neuronal prep of 4 newborn pups performed at p0/p1), with hundreds of synapses analyzed. All the details of the experimental procedure and statistics are in the Methods and indicated in the figure legend.

From the expansion microscopy data, we detected actin in VGLUT1-positive regions. The abundance of actin significantly decreases in neurons lacking synapsins. whereas the actin signal overlapping with the postsynaptic marker PSD95 remains unchanged. From the two-color super-resolution STED imaging, we see a similar decrease of actin signal, which can be rescued by overexpressing the full-length synapsin or synapsin IDR, which is known to be able to undergo phase separation. We have now included the representative images as heatmaps of individual channels, which makes the effect more clearly visible (**new EV 12A**).

We would also like to note that we specifically developed a script (deposited and publicly available on Github) to unbiasedly segment both pre- and post- synaptic channels and quantifies the intensity of the actin channel in the positive structures. Finally, to ensure rigorous statistical measurements for actin enrichment in SV clusters, we have now a) compared the means between the replicates (averaging all synapses within one biological replicate); b) compared a normalized intensity signal of actin channel with regions positive for (wild-type and synapsin triple knockout) or both synaptophysin and EGFP (in rescue experiments) (**EV 12C**).

These perturbations of actin and synapsin in two vertebrate synapse models, and the downstream impacts on presynaptic organization, demonstrate the physiological relevance of our findings.

The role of phase separation in vesicle cluster organization is really important question but this paper is far too premature in addressing this question.

We hope that the new series of experiments have strengthened our claims, thereby rendering this manuscript ready for publication.

Referee #2:

In "CONDENSATES OF SYNAPTIC VESICLES AND SYNAPSIN ARE MOLECULAR BEACONS FOR ACTIN SEQUESTERING AND POLYMERIZATION", Akshita et al build upon their previous results that synapsin undergoes a phase separation to sequester synaptic vesicles, to demonstrate that condensation and actin binding properties of synapsin result in actin polymerization that forms actin arrays linking synaptic vesicles in pre-synaptic boutons. These results are supported by both in vitro and in vivo reconstitution assays. These assays also demonstrate that the IDR domain of synapsin is necessary for actin bundling. For the most part, the data is clearly presented, and experiments are performed with the appropriate controls. I have only a few minor concerns/questions of clarifications that would help to strengthen the current submission.

We thank the Referee for the supportive comments and the valuable suggestions on improving the manuscript, which we implemented in the revised version.

1. The authors nicely demonstrate that the stress granule protein G3BP2 which can undergo phase separation and create an actin meshwork, lacks actin binding properties and therefore cannot polymerize actin like synapsin. However, it is unclear to me, where the actin binding sites of synapsin are located. From what I can find, it appears that the actin binding domain is primarily located in the C domain, not the IDR. Therefore, it is a bit unclear to me why the IDR is necessary and sufficient to recruit and promote the polymerization of actin. Could the authors please indicate where the actin binding sites are in synapsin and indicate these in the diagram of synapsin in S. Fig. 1.

We thank the Referee for this point. Synapsin-1 contains several actin-binding regions within both domain C and the IDR region. Since domain C is also responsible for dimerization of synapsin-1, we wanted to tease apart whether actin-binding region from a part of protein that can phase separate (i.e., IDR region) versus dimerization region (domain C), which does not phase separate, could trigger actin polymerization. Indeed, the actin binding property alone is not sufficient to trigger phase separation (**EV 4**). We have now indicated the actin binding regions of synapsin 1 (**EV 1A**).

2. Could the authors also comment in the discussion about why the IDR alone leads to linear rather than aster-like actin formations.

The distinct shape of asters is extremely interesting, as it strongly points to role of interfacial tension of condensates in regulating the morphologies of actin network. IDR alone forms more linear actin formations because the actin seeding is more discrete, while full-length may offer more seeding sites due to the actin-binding regions. Inspired by this comment and in line of investigating various variants of synapsin 1, we examined whether phosphomimetics of synapsin 1 would form networks of distinct morphologies. Our analyses (**new Figure 3**) strongly support this observation by indicating that CaMKII-containing phosphomimetics of full-length protein, which are precisely within the IDR, results in the linear-like actin assemblies.

Furthermore, we performed a heatmap analysis (Pearson's correlation) of co-dependencies for the node properties of the synapsin 1:actin assemblies, followed by a principal component (PC) analysis of synapsin1:actin assemblies. Analyzing the similarities between nodes of the actin network and their relationships, we observed that larger nodes exhibit the highest range of intensities and are the most connected, whereas smaller nodes show the opposite trend (**new Fig. 6D, E**).

3. In Fig. 5, actin appears to be more clustered with the synapsin FL or IDR rescues than in the WT. Both rescues also appear to disrupt the regularly spaced cortical actin rings of the axon. Based on these results, could the authors comment on synapsin's role in regulating the reserve and readily releasable pools of synaptic vesicles as well as potential regulation of actin exchange between SV-associated asters and cortical actin rings?

Thank you for this valuable comment. In synapsin TKO neurons, actin that is not recruited enriched in the synapses seems recruited and enriched in the actin rings, making the spacing undisrupted but the intensity of actin signal. In neurons ectopically expressing synapsin-1 constructs, less actin seems to be in the processes in comparison to the neurons non-transfected by synapsin-1 with more actin accumulating with synapsin-1 in the soma from the very beginning of protein translation. These data suggest the roles of several pools of presynaptic actin: within the synapsin-SV condensate; in the torus around the SVs (periaxonal zone); in the asters that extend from the synapsin-SV condensate, some of which extend to other condensates/synapses. This is in line with prior reports showing that there are several functional pools of presynaptic actin regulating exo-/endo- cytoskeleton.

Given the long distance and varying topology of the actin network within synaptic boutons (Bingham *et al*, 2023) (Ogunmowo *et al*, 2023), our model of synapsin:actin assemblies explains that SV trafficking can occur independently of the concentration gradient of SVs. Indeed, our superresolution STED imaging in murine hippocampal neurons indicates that there is an absence of actin enrichment in the vicinity of SVs in the absence of synapsins. In the revised version of the manuscript, we have now included images of neurons with heatmaps emphasizing the distribution actin signal along the axons (new EV12) and amended the Discussion to emphasize on the potential roles of distinct actin pools in axons (lines 403-415).

Referee #3:

The manuscript by Chhabra, Hoffmann *et al*. investigates the specific role of synapsins in the organisation of synaptic vesicles (SVs) in neurons by supporting the architecture of actin asters around these SVs, as well as their intercommunication. The authors discuss the relevance of these structures for neuronal communication based on SV clusters that assemble by liquid-liquid phase separation (LLPS) thanks to synapsins. The authors have previously investigated the role of synapsin-1 in the organisation of SV clusters by analysing condensates with liquid-like properties and in cellular studies of motility, and found that the full-length and short intrinsically disordered fragment of synapsin 1 restored the SV motility pattern in synapsin triple knock-out animals. Indeed, casein kinase II appears to be a key kinase in regulating the formation of aggregates with SVs, suggesting a regulated disassembly similar to that observed in synaptic activity. Synapsins are known to bind actin, and here the reorganisation of these three elements was monitored by *in vitro* reconstitution approaches and super-resolution imaging in lamprey and mouse neurons. The data presented show that sites of LLPS of synapsin correspond to SVs and actin polymerisation foci. The organisation of SV clusters in neurons appears to be synapsin dependent, as the triple knock-out mouse (all three synapsins) shows poorly organised clusters. This is rescued by reconstitution with full length and short intrinsically disordered fragment of synapsin 1.

The manuscript is well written and the experiments are generally well described and documented. There are some relevant questions that are required at this stage to support the statements in the main text and the claims of authors:

We thank the Referee for the positive assessment of our work, the careful reading and the constructive suggestions, which we integrated in the revised manuscript.

1. Do the authors think that the use of proteins and models from different species (rabbit actin, human synapsin, mouse and lamprey experimental models) might influence their results and therefore make this study less general than described in principle in the text? Are all the proteins involved conserved between species and regulated by similar processes/proteins?

Thank you for this important point. The amino acid sequences of synapsin-1, the chemical signatures of synapsin-1 IDRs and their physiological functions at synapses are all highly conserved across both vertebrates and invertebrates (*Kao et al, 1999; Pieribone et al, 1995; Bloom et al, 2003; Hoffmann et al, 2025*), suggesting that these findings are likely to be generalizable across species. We have now included this point in the Discussion (**lines 419-429**).

2. Are the actin asters resistant to actin-targeted drugs, such as latrunculin A and cytochalasin B or D, or to myosin-targeted drugs? SVs also appear to form asters in the images provided (e.g. Fig. 3; Supplementary Fig. 6). Can the authors ensure that there are no other elements in isolated SVs that could interfere or regulate the reaction? The composition of the SV should be shown as an additional control.

This is an excellent suggestion, which we followed in detail. Specifically, we investigated the stability of polymerized actin within synapsin 1 condensates performed pharmacological manipulations in a parallel series of experiments, with in-vitro reconstituted condensates and the acute perturbations in the lamprey synapse.

In the reconstituted experiments, we first preincubated G-actin with Latrunculin A (Lat A). Subsequently, we added Syn1 and triggered condensation with 3% (w/v) PEG 8000. Indeed, the addition of actin monomers into synapsin condensates supplemented with Lat A prevented actin polymerization. On contrary, pre-formed synapsin-actin assemblies remained stable upon the addition of Lat A, clearly indicating that synapsin-driven actin polymerization results in the formation of stable actin fibrils. These results are now included as **the new main Fig. 5 A-D**.

We also employed acute pharmacological perturbations of the actin cytoskeleton at a lamprey synapse. We treated the reticulospinal synapses with 2.5 μ M Lat A either before or after acute microinjections of Alexa488-phalloidin to label any F-actin present. Along the lines of our *in vitro* experiments, pre-treating the synapses with Lat A prevented actin polymerization, as shown by a significant decrease in the number of phalloidin positive presynaptic actin structures as compared to control. In contrast, Lat A was unable to disrupt F-actin structures that were already stabilized with phalloidin, including the actin asters emanating from the actin toroids at synapses. Thus, preformed actin asters are resistant to disruption by Lat A. These results are now included as **the new main Fig. 5 E-I**.

We also reconstituted actin with SVs as a control to test if any SV proteins, including the residual synapsins that are peripherally-associated with the lipid bilayer of SVs, could promote actin polymerization. Yet, we did not observe actin polymerization in the presence of SVs alone, indicating the central role of synapsins. Note that SV preparation is of high purity and SVs contain a small fraction of synapsin 1 attached to it (\sim 50 copies/SV, *Wilhelm et al, 2014*), although this amount of synapsin alone was not sufficient to trigger actin polymerization. In line with previous studies (*Benfenati et al, 1989; Benfenati et al, 1992*), the presence of low concentrations of synapsin 1 enhanced the binding of actin to SVs, as indicated by

fluorescence correlation spectroscopy. This is all presented in **the main text (lines 242-251)** and presented as a **revised figure EV8**, including the composition of our SV preparation.

3. Is the order of addition of the elements important for the formation of aggregates? Within the cell, all the elements will be together; as actin can bind to SVs in the absence of synapsin-1, have the authors tried adding the elements in different order and at different times of the reaction?

In a series of control reconstitutions, we varied the reaction order by either adding G-actin monomers to Syn1-FL condensates or vice versa. Our data indicate that irrespectively of the reaction order, synapsin condensation results in the sequestering of G-actin monomers and triggers the polymerization of actin in the absence of any additional nucleation factor. This is all presented in **the main text (lines 153-156)** and included as a **new figure EV3**.

4. Although imaging of the lamprey axon shows co-localisation of actin assemblies, synapsin and SV, these do not appear to form asters (Fig. 3f does not demonstrate aster presence).

We now employed a newly available Leica Stellaris microscope, to visualize the actin structures in the lamprey presynapse. The imaging of freshly isolated spinal cords bathed in ringer solution followed by injection of Alexa Fluor™ 488-phalloidin indicated the individual synapses connected together by phalloidin labeled “aster-like” structures. This is now included as a **new Figure 4F**.

5. Figure 5 requires quantification. The MFI of insets in Fig 5a-d is different in single channels (grey images) and the merge images (magenta/green). Synaptophysin aggregates of smaller size seem to co-localize with actin structures, even if synapsins are absent. This is in contrast with the claim of authors. Are microtubules or other cytoskeletal components relevant here?

Thank you for pointing out this unintentional mistake – we have now corrected this the revised figure 8. The colocalization of dispersed SVs and small SV cohorts with actin can be due to other actin-binding proteins present at the surface of SVs, which is also supported by the FCS analysis (*Perego et al 2020*; please also see our EV12).

6. Figure 5: Do these STED images correspond to single planes? Where is located the post-synaptic density in these pre-synaptic boutons? Authors should visualize the complete structure here to help differentiate single neurons (there can be also auto-synapses)? The asters are not observed there, either in wt or Ko mice; main structure of actin seems to be conserved, as pointed-out by authors (Fig. 5i). Is there other cytoskeletal element involved? Molecular motors helping the movement of the SVs?

Thank you for all of these remarks. Yes, in 2 color-STED the images correspond to single planes. We did two color STED with SVs in red and actin in far-red; to visualize post-synaptic marker with super-resolution was not possible, so we had to opted for another approaches. Specifically, we performed two lines of experiments:

- (i) To visualize presynaptic actin, we turned to the advanced cryo-EM tomography of hippocampal neurons in culture. Indeed, we could observe actin filaments associated with SV clusters as well as the filaments in the periaxonal zone. These data are now included as a **new main Figure 7 A-E**.
- (ii) We performed ultrastructure expansion microscopy of murine hippocampal neurons and quantified that the presynaptic actin is significantly reduced at the

presynapses from synapsin TKO neurons. These data are now included as a **new main Figure 7 F-I**.

7. The triple knock-out mouse model is reconstituted with one of the synapsins. Are the other members of the family involved in the organisation and regulation of the LLPS described so that these experiments lack the complexity required for the in vivo process? The authors should perform some in vitro reconstitution with all the synapsins.

In the current study, our results were obtained from glutamatergic synapses where synapsin-1 is the main variant. We fully agree that it will be important to explore how the synapsin-SV-actin assemblies work at GABAergic synapses where synapsin-2, another family member of synapsins that undergoes phase separation, plays an important role (*Song & Augustine, 2023*) (*Longfield et al, 2024*). However, inspired by this comment and in conjunction with the suggestions from the other Referees, we invested our efforts in assessing how different activity states (i.e., phosphomimetics of synapsin-1) affect the actin morphology.

To do that we purified synapsin 1 phosphorylation mutants that mimics synaptic activity affects synapsin 1-actin assemblies, we cloned PKA (Ser 9), CaMKII (Ser-568 and Ser-605), and double PKA/CaMKII (Ser 9, Ser-568, and Ser-605) by mutating the target serines to glutamate. We were able to observe that synapsin 1 phosphomimetics had on synapsin 1-actin assemblies. Sholl analyses of actin networks formed by these condensates of synapsin 1 indicate that PKA significantly reduces the connectivity and network properties of synapsin 1-actin assemblies. Yet, the CaMKII phosphomimetic, either alone or in combination with the PKA phosphomimetic, led to a dramatic decrease in actin polymerization, practically abolishing the ability of synapsin to trigger actin polymerization. These data suggest that phospho-regulation of synapsin 1, by enzymes triggered during synaptic activation, releases actin from synapsin condensates, presumably allowing for actin-driven remodeling processes at the plasma membrane and at the periaxonal zone. These new results are included **in the main text (lines 205-230)** and as **a new main Figure 3 and new EV6**.

Minor:

9. Fig Panels Fig. 5e, f, g, i j should be properly quoted in the text under Results, page 10.

This is now corrected and adjusted for new figure numbering (i.e., the former Fig. 5 is Fig. 8 in the revised manuscript).

10. Suppl. Fig. 8 describes the recognition of synapsin-1 in lampreys by the antibody used; please, make it clear in the main text.

This is now directly stated in the main text (line 274).

References

Bähler M & Greengard P (1987) Synapsin I bundles F-actin in a phosphorylation-dependent manner. *Nature* 326: 704–707

Benfenati F, Valtorta F, Bähler M & Greengard P (1989) Synapsin I, a neuron-specific phosphoprotein interacting with small synaptic vesicles and F-actin. *Cell Biol Intl Rep* 13: 1007–1021

Benfenati F, Valtorta F, Chiergatti E & Greengard P (1992) Interaction of free and synaptic vesicle-bound synapsin I with F-actin. *Neuron* 8: 377–386

Bingham D, Jakobs CE, Wernert F, Boroni-Rueda F, Jullien N, Schentarra E-M, Friedl K, Moura JDC, Bommel DM van, Caillol G, *et al* (2023) Presynapses contain distinct actin nanostructures. *J Cell Biol* 222: e202208110

Bloom O, Evergren E, Tomilin N, Kjaerulff O, Löw P, Brodin L, Pieribone VA, Greengard P & Shupliakov O (2003) Colocalization of synapsin and actin during synaptic vesicle recycling. *J Cell Biol* 161: 737–747

Fernández-Busnadiego R, Zuber B, Maurer UE, Cyrklaff M, Baumeister W & Lucic V (2010) Quantitative analysis of the native presynaptic cytomatrix by cryoelectron tomography. *J Cell Biol* 188: 145–156

Hirokawa N, Sobue K, Kanda K, Harada A, Yorifuji H (1989) The cytoskeletal architecture of the presynaptic terminal and molecular structure of synapsin 1. *J Cell Biol*: 108:111–126

Hoffmann C, Ruff KM, Edu IA, Shinn MK, Tromm JV, King MR, Pant A, Ausserwöger H, Morgan JR, Knowles TPJ, *et al* (2025) Synapsin condensation is governed by sequence-encoded molecular grammars. *J Mol Biol*: 168987

Kao H, Porton B, Hilfiker S, Stefani G, Pieribone VA, DeSalle R & Greengard P (1999) Molecular evolution of the synapsin gene family. *J Exp Zool* 285: 360–377

Longfield SF, Gormal RS, Feller M, Parutto P, Reingruber J, Wallis TP, Joensuu M, Augustine GJ, Martínez-Mármol R, Holcman D, *et al* (2024) Synapsin 2a tetramerisation selectively controls the presynaptic nanoscale organisation of reserve synaptic vesicles. *Nat Commun* 15: 2217

Milovanovic D, Wu Y, Bian X & De Camilli P (2018) A liquid phase of synapsin and lipid vesicles. *Science* 361: 604–607

Ogunmowo TH, Jing H, Raychaudhuri S, Kusick GF, Imoto Y, Li S, Itoh K, Ma Y, Jafri H, Dalva MB, *et al* (2023) Membrane compression by synaptic vesicle exocytosis triggers ultrafast endocytosis. *Nat Commun* 14: 2888

Perego E, Reshetniak S, Lorenz C, Hoffmann C, Milovanović D, Rizzoli SO & Köster S (2020) A minimalist model to measure interactions between proteins and synaptic vesicles. *Sci Rep* 10: 21086

Pieribone VA, Shupliakov O, Brodin L, Hilfiker-Rothenfluh S, Czernik AJ & Greengard P (1995) Distinct pools of synaptic vesicles in neurotransmitter release. *Nature* 375: 493–497

Sankaranarayanan S, Atluri PP & Ryan TA (2003) Actin has a molecular scaffolding, not propulsive, role in presynaptic function. *Nat Neurosci* 6: 127–135

Shupliakov O, Bloom O, Gustafsson JS, Kjaerulff O, Löw P, Tomilin N, Pieribone VA, Greengard P & Brodin L (2002) Impaired recycling of synaptic vesicles after acute perturbation of the presynaptic actin cytoskeleton. *Proc Natl Acad Sci USA* 99: 14476–14481

Song S-H & Augustine GJ (2023) Different mechanisms of synapsin-induced vesicle clustering at inhibitory and excitatory synapses. *Cell Rep* 42: 113004

Südhof TC, Czernik AJ, Kao HT, Takei K, Johnston PA, Horiuchi A, Kanazir SD, Wagner MA, Perin MS & De Camilli P (1989) Synapsins: mosaics of shared and individual domains in a family of synaptic vesicle phosphoproteins. *Science* 245: 1474–1480

Wilhelm BG, Mandad S, Truckenbrodt S, Kröhnert K, Schäfer C, Rammner B, Koo SJ, Claßen GA, Krauss M, Haucke V, *et al* (2014) Composition of isolated synaptic boutons reveals the amounts of vesicle trafficking proteins. *Science* 344: 1023–1028

Dear Drago,

Thank you again for submitting your revised manuscript (EMBOJ-2024-118009R-Q) to The EMBO Journal for our consideration, and for your patience during peer review. As I have already informed you, referee #1 was not available for re-review at this time, but your revision has been seen by the original referees #2 and #3, who had previously assessed the initial version of your study, and who kindly agreed to provide additional feedback on whether they find the initially raised concerns by all three referees adequately addressed. The reports of the referees are included below.

I am pleased to say that both referees find the manuscript properly and adequately revised, and the initially raised criticisms and concerns by all three referees sufficiently addressed. They have only few remaining comments for minor corrections or further improvements, which I would like to invite you to address in a final version of your manuscript. Please submit along with your revised manuscript a point-by-point response addressing all remaining comments and detailing any changes to the manuscript and the figures.

From the editorial side, there are also a few changes and corrections that we need you to make in the final version of the manuscript before we can proceed with its acceptance and publication in The EMBO Journal:

- Figures need to be removed from the main manuscript file, only Figure legends should remain below the list of References.
- "Equal contribution" is permitted only for the first and/or the corresponding authors; please remove the "equal contribution" indication for the "second-position" authors and, instead, make sure that the list of authors reflects accurately their contributions to the study and the manuscript; in addition, please provide detailed descriptions of each co-author's contributions using the CRediT contribution taxonomy system when you re-submit your manuscript to our online manuscript handling system (see below for more information).
- The funding information included in the Comments box could not be extracted by our production team; therefore, all funders should be added to the "More Funders" list: "Christian Hoffmann is supported by a fellowship of the Innovative Minds Program of the German Dementia Association. Johannes V Tromm is supported by the German Student Scholarship Foundation."
- Please provide a list of up to 5 relevant keywords after the Abstract of your revised manuscript to improve search engine discoverability of the article.
- Please include your "Code Availability" statement in the "Data Availability" section of the revised manuscript.
- The "Competing interests" heading should be renamed to "Disclosure and competing interests statement".
- The author contributions statement should be removed from the manuscript file. Instead, we use CRediT to specify the contributions of each author in the journal submission system. Please feel free to use the free text box to provide more detailed descriptions during submission. See also our guide to authors for more information:
<https://www.embopress.org/page/journal/14602075/authorguide#authorshippinguidelines>.
- All Figure callouts should be listed sequentially.
- We noticed that callouts for Figure panel 8E are missing.
- Please move all information currently included in the last column of the Author Checklist to the main manuscript file as appropriate. Only the manuscript sections where the information can be found (e.g. "Methods", "Figure legends" etc.) should be listed in the last column of the Author Checklist. Please include the Github link in the "Data availability" statement of the manuscript rather than in the Author Checklist.
- The primer sequences currently listed in your Supplementary Tables 1 and 2 should be moved to your Reagents and Tools Table.
- We noticed that the number of Expanded View (EV) Figures in your manuscript is unusually high (currently 12). Please note that we recommend authors to select a limited number (typically 5, maximum 7-8 if necessary) of supplementary figures for inclusion in the article as Expanded View Figures in order to improve their accessibility, visibility, and utility. Any extra Figures should be included in the Appendix file (along with supplementary text and tables).
- The Appendix file needs to be uploaded as a single PDF file.
- The Movie files should be renamed to "Movie EV1-EV3" and the corresponding callouts corrected accordingly. Please remove their legends from the main manuscript file and, instead, zip each one together with the corresponding Movie file.

- Please note that EMBO press papers are accompanied online by:
A) a short (2 sentences) summary of the findings and their significance,
B) 2-5 short bullet points highlighting the key results, and
C) a synopsis image in .jpg or .png format that is exactly 550 pixels wide and 300-600 pixels high (the height is variable). Please note that the text needs to be legible at the final size.
Please upload this information along with your revised manuscript (the text for A and B should be provided in a separate Word file).

- During our standard Figure checks we detected cell reuse between Figure 8A-H And Figure EV12A. Please check these Figures carefully and correct them if necessary, or clarify and detail this reuse in the respective Figure legends if it is intentional/unavoidable and justified.

- During our routine data checks, our data editors have raised the following queries regarding figures, data, and legends. Please make sure that all requests below are completely addressed in the final version of your manuscript:

1. Please provide the exact p values in the legends of Figures 5H, I; 6I, 8J, EV5 A, EV11 C (or in the Figures).
2. Please indicate the statistical test used for data analysis in the legend of Figure 8J.
3. Please note that the box plots need to be defined in terms of minima, maxima, centre, bounds of box and whiskers, and percentile in the legends of Figures 6I, 8J; EV8 C, EV12 B-D.
4. Please note that information related to "n" is missing in the legends of Figures 1D, 8J, EV8 C, EV12 B-D.
5. Please note that the measure of center for the error bars needs to be defined in the legends of Figures 1C, E.
6. Please note that the scale bar needs to be defined in the legends of Figures 7B, G.
7. Please note that the blue and red arrows are not defined in the legend of Figure 2D. This needs to be rectified.

- The order of the manuscript sections must be corrected as follows: Title page - Abstract and Keywords - Introduction - Results - Discussion - Methods - Data Availability - Acknowledgements - Disclosure and Competing Interests Statement - References - Figure Legends - main Tables (if there are any) - Expanded View Figure Legends.

Please also note that as part of the EMBO publications' Transparent Editorial Process, The EMBO Journal publishes online a Peer Review File along with each accepted manuscript. This File will be published in conjunction with your paper and will include the referee reports, your point-by-point response and all pertinent correspondence relating to the manuscript. You can opt out of this by letting the editorial office know (contact@embojournal.org). If you do opt out, the Peer Review File link will point to the following statement: "No Peer Review File is available with this article, as the authors have chosen not to make the review process public in this case."

We look forward to seeing a final version of your manuscript as soon as possible. Please let us know if you have any questions and use this link to submit your revision: xxxxxxxxxxxxxxxxxxxxxxxxxxxxxx.

Best regards,

Ioannis

Referee #2:

The authors have addressed my comments and the concerns of the referee #1 and have included new phosphomimetic experiments to assess how phosphorylation of different actin domains regulate the different actin assemblies. However the data from mouse neurons particular in Fig 7 is still less than convincing but is strengthened by later rescue experiments in mouse neurons. Specifically, the authors should address why there is so little actin in the cryo-EM image and how statistics were performed to avoid significance from pseudoreplication in Fig. 7 H and I - H and I would benefit from averaging actin intensity from each experiment and performing statistics.

Referee #3:

The authors have properly revised the manuscript and addressed the significant concerns raised by Reviewers 3 and 1. They have demonstrated that synapsin condensates initiate actin polymerization in vitro and recapitulate the observed networks in living vertebrate synapses.

The correct identification of the actin filaments in proximity to vesicle clusters has been facilitated by cryoelectron tomography in murine hippocampal neurons, enabling 3D reconstruction at a high resolution. Indeed, the supplementary experiments performed have consistently demonstrated that pre-synaptic actin is regulated by synapsin1, in contrast to post-synaptic densities. Indeed, a substantial body of data from at least three pertinent physiological systems has been measured and included in the manuscript to support these claims. The in vitro experiments demonstrated that the mechanism of synapsin 1 phosphorylation is supported, and that it is associated with relevant neurophysiological processes, namely CaMKII and PKA. The use of phosphomimetic mutants constitutes a highly efficacious experimental approach. Authors have enhanced the coherence of their claims by employing a range of compounds that target actin polymerisation and stability. In addition, they have utilized diverse in vitro datasets to assess the potential significance of the order in which components of the networks are added, with the aim of observing the resulting congregates.

Minor comments

- It is unclear why there is an absence of an image in Fig. 5B for Syn FL 0 min. This image must be included.
 - It would be interesting for readers if the cryoET tomography is included as a Suppl. Movie.
- The manuscript may be considered for publication upon the resolution of these minor issues.

Responses to the Editorial and Referees' Comments

Responses to the Editorial comments:

1. Figures need to be removed from the main manuscript file, only Figure legends should remain below the list of References.

This is adjusted in the revised version.

2. "Equal contribution" is permitted only for the first and/or the corresponding authors; please remove the "equal contribution" indication for the "second-position" authors and, instead, make sure that the list of authors reflects accurately their contributions to the study and the manuscript; in addition, please provide detailed descriptions of each co-author's contributions using the CRediT contribution taxonomy system when you re-submit your manuscript to our online manuscript handling system (see below for more information).

These updates are now incorporated.

3. The funding information included in the Comments box could not be extracted by our production team; therefore, all funders should be added to the "More Funders" list: "Christian Hoffmann is supported by a fellowship of the Innovative Minds Program of the German Dementia Association. Johannes V Tromm is supported by the German Student Scholarship Foundation."

This is now done in the online platform.

4. Please provide a list of up to 5 relevant keywords after the Abstract of your revised manuscript to improve search engine discoverability of the article.

The keywords are now provided.

5. Please include your "Code Availability" statement in the "Data Availability" section of the revised manuscript.

This is updated in the revised version.

6. The "Competing interests" heading should be renamed to "Disclosure and competing interests statement".

This is adjusted in the revised version.

7. The author contributions statement should be removed from the manuscript file. Instead, we use CRediT to specify the contributions of each author in the journal submission system. Please feel free to use the free text box to provide more detailed descriptions during submission. See also our guide to authors for more information: <https://www.embopress.org/page/journal/14602075/authorguide#authorshipguidelines>.

This is now provided as a separate word file.

8. All Figure callouts should be listed sequentially.

This is adjusted in the revised version.

9. We noticed that callouts for Figure panel 8E are missing.

This is corrected.

10. Please move all information currently included in the last column of the Author Checklist to the main manuscript file as appropriate. Only the manuscript sections where the information can be found (e.g. "Methods", "Figure legends" etc.) should be listed in the last column of the

Author Checklist. Please include the Github link in the "Data availability" statement of the manuscript rather than in the Author Checklist.

This is updated in the revised version.

11. The primer sequences currently listed in your Supplementary Tables 1 and 2 should be moved to your Reagents and Tools Table.

This is updated in the revised version.

12. We noticed that the number of Expanded View (EV) Figures in your manuscript is unusually high (currently 12). Please note that we recommend authors to select a limited number (typically 5, maximum 7-8 if necessary) of supplementary figures for inclusion in the article as Expanded View Figures in order to improve their accessibility, visibility, and utility. Any extra Figures should be included in the Appendix file (along with supplementary text and tables).

We revised all figures and selected 7 EV Figures with the remaining figures prepared as an Appendix File. The callouts in the main text have been adjusted to this new flow.

13. The Appendix file needs to be uploaded as a single PDF file.

This is now provided as a separate file.

14. The Movie files should be renamed to "Movie EV1-EV3" and the corresponding callouts corrected accordingly. Please remove their legends from the main manuscript file and, instead, zip each one together with the corresponding Movie file.

This is updated in the revised version.

15. Please note that EMBO press papers are accompanied online by:
A) a short (2 sentences) summary of the findings and their significance,
B) 2-5 short bullet points highlighting the key results, and
C) a synopsis image in .jpg or .png format that is exactly 550 pixels wide and 300-600 pixels high (the height is variable). Please note that the text needs to be legible at the final size. Please upload this information along with your revised manuscript (the text for A and B should be provided in a separate Word file).

These files (A&B as MS Word and C as PNG) are now provided as separate files.

16. During our standard Figure checks we detected cell reuse between Figure 8A-H And Figure EV12A. Please check these Figures carefully and correct them if necessary, or clarify and detail this reuse in the respective Figure legends if it is intentional/unavoidable and justified.

This has been explicitly stated in the figure legend.

17. During our routine data checks, our data editors have raised the following queries regarding figures, data, and legends. Please make sure that all requests below are completely addressed in the final version of your manuscript:

17.1. Please provide the exact p values in the legends of Figures 5H, I; 6I, 8J, EV5 A, EV11 C (or in the Figures).

This is updated in the revised version.

17.2. Please indicate the statistical test used for data analysis in the legend of Figure 8J.

This is updated in the revised version.

17.3. Please note that the box plots need to be defined in terms of minima, maxima, centre, bounds of box and whiskers, and percentile in the legends of Figures 6I, 8J; EV8 C, EV12 B-D.

This is updated in the revised version.

17. 4. Please note that information related to "n" is missing in the legends of Figures 1D, 8J, EV8 C, EV12 B-D.

This is updated in the revised version.

5. Please note that the measure of center for the error bars needs to be defined in the legends of Figures 1C, E.

This is updated in the revised version.

17.6. Please note that the scale bar needs to be defined in the legends of Figures 7B, G.

This is updated in the revised version.

17.7. Please note that the blue and red arrows are not defined in the legend of Figure 2D. This needs to be rectified.

This is updated in the revised version.

18. The order of the manuscript sections must be corrected as follows: Title page - Abstract and Keywords - Introduction - Results - Discussion - Methods - Data Availability - Acknowledgements - Disclosure and Competing Interests Statement - References - Figure Legends - main Tables (if there are any) - Expanded View Figure Legends.

The flow has been adjusted.

Responses to the comments of Referee #2:

The authors have addressed my comments and the concerns of the referee #1 and have included new phosphomimetic experiments to assess how phosphorylation of different actin domains regulate the different actin assemblies. However the data from mouse neurons particular in Fig 7 is still less than convincing but is strengthened by later rescue experiments in mouse neurons.

We thank the Referee for the positive assessment of our work and the constructive suggestions for further discussion, which we all integrated into the revised manuscript.

1. Specifically, the authors should address why there is so little actin in the cryo-EM image.

We have now explicitly discussed this (ln. 363-368): “*Only a few filaments were detected at the presynapse, likely because the dense presynaptic environment and generally short actin filaments are challenging to visualize. Furthermore, at rest, presynaptic actin is largely monomeric, making it elusive in cryo-electron tomograms. Future development of machine-learning approaches that will require larger datasets and enable more robust quantitative analysis will be needed to address these challenges.*”

2. How statistics were performed to avoid significance from pseudoreplication in Fig. 7 H and I - H and I would benefit from averaging actin intensity from each experiment and performing statistics.

We use the expansion microscopy to resolve and analyze each synaptic structure separately rather than averaging all synapses in a dish. To ensure the robustness, omit oversampling and account for synapse heterogeneity, we only analyzed synapse where both pre- and post- synaptic structures were clearly resolved upon expansion. This has now been explicitly stated in the main text (ln. 370-372).

Responses to the comments of Referee #3:

The authors have properly revised the manuscript and addressed the significant concerns raised by Reviewers 3 and 1. They have demonstrated that synapsin condensates initiate actin polymerization in vitro and recapitulate the observed networks in living vertebrate synapses.

The correct identification of the actin filaments in proximity to vesicle clusters has been facilitated by cryoelectron tomography in murine hippocampal neurons, enabling 3D reconstruction at a high resolution. Indeed, the supplementary experiments performed have consistently demonstrated that pre-synaptic actin is regulated by synapsin1, in contrast to post-synaptic densities. Indeed, a substantial body of data from at least three pertinent physiological systems has been measured and included in the manuscript to support these claims. The in vitro experiments demonstrated that the mechanism of synapsin 1 phosphorylation is supported, and that it is associated with relevant neurophysiological processes, namely CaMKII and PKA. The use of phosphomimetic mutants constitutes a highly efficacious experimental approach. Authors have enhanced the coherence of their claims by employing a range of compounds that target actin polymerisation and stability. In addition, they have utilized diverse in vitro datasets to assess the potential significance of the order in which components of the networks are added, with the aim of observing the resulting congregates.

We thank the Referee for the shared enthusiasm about our work and, especially, for the constructive and fruitful revision process.

Minor comments

1. It is unclear why there is an absence of an image in Fig. 5B for Syn FL 0 min. This image must be included.

These images are now included in the revised version.

2. It would be interesting for readers if the cryoET tomography is included as a Suppl. Movie. **This is a great suggestion. We have now prepared a cryoET tomogram with the annotations and incorporated it as a Movie EV4.**

The manuscript may be considered for publication upon the resolution of these minor issues.

Dear Drago,

Congratulations on an excellent work! I am very pleased to inform you that your manuscript has been accepted for publication in The EMBO Journal. Thank you for comprehensively addressing the initially raised referees' concerns and our editorial requests for corrections and changes.

Please remember to send me your edited video synopsis for your article within the next few days.

Your manuscript will then be processed for publication by EMBO Press. It will be copy edited and you will receive page proofs prior to publication. Please note that you will be contacted by Springer Nature Author Services to complete licensing and payment information.

If you have any questions, please do not hesitate to contact the Editorial Office. Thank you for your contribution to The EMBO Journal. Working with you has been a pleasure!

Best regards,

Ioannis
